# A genome-wide CRISPR screen identifies *CALCOCO2* as a regulator of beta cell function influencing type 2 diabetes risk

Antje K. Rottner ●[1], Yingying Ye ●[2,9], Elena Navarro-Guerrero[3,9], Varsha Rajesh[2], Alina Pollner[2], Romina J. Bevacqua[4,5], Jing Yang ●[2], Aliya F. Spigelman[6], Roberta Baronio[3], Austin Bautista[6], Soren K. Thomsen[1], James Lyon[6], Sameena Nawaz[1], Nancy Smith[6], Agata Wesolowska-Andersen[7], Jocelyn E. Manning Fox[6], Han Sun ●[2], Seung K. Kim ●[4,5], Daniel Ebner[2], Patrick E. MacDonald[6] & Anna L. Gloyn ●[1,2,5,7,8] ✉

Identification of the genes and processes mediating genetic association signals for complex diseases represents a major challenge. As many of the genetic signals for type 2 diabetes (T2D) exert their effects through pancreatic islet-cell dysfunction, we performed a genome-wide pooled CRISPR loss-of-function screen in a human pancreatic beta cell line. We assessed the regulation of insulin content as a disease-relevant readout of beta cell function and identified 580 genes influencing this phenotype. Integration with genetic and genomic data provided experimental support for 20 candidate T2D effector transcripts including the autophagy receptor *CALCOCO2*. Loss of *CALCOCO2* was associated with distorted mitochondria, less proinsulin-containing immature granules and accumulation of autophagosomes upon inhibition of late-stage autophagy. Carriers of T2D-associated variants at the *CALCOCO2* locus further displayed altered insulin secretion. Our study highlights how cellular screens can augment existing multi-omic efforts to support mechanistic understanding and provide evidence for causal effects at genome-wide association studies loci.

Genome-wide association studies (GWAS) have delivered thousands of robust associations for type 2 diabetes (T2D) and related traits but most map to noncoding regions with a likely regulatory function[1]. Incomplete fine mapping means most GWAS loci are not mapped to a single causal variant but rather multiple variants in a credible set, each of which could potentially influence gene expression in a different cellular context. The typical step after fine mapping involves connecting the putatively causal variants and the regulatory elements to the genes they regulate, using methods such as *cis*-expression quantitative trait loci (eQTL) colocalization, single-cell chromatin co-accessibility and DNA proximity assays[2–5]. The limitations of these approaches are their cell type- and context dependency (assays conducted in inappropriate cell

[1]Oxford Centre for Diabetes, Endocrinology and Metabolism, Radcliffe Department of Medicine, University of Oxford, Oxford, UK. [2]Department of Pediatrics, Division of Endocrinology, Stanford School of Medicine, Stanford University, Stanford, CA, USA. [3]Target Discovery Institute, Nuffield Department of Medicine, University of Oxford, Oxford, UK. [4]Department of Developmental Biology, Stanford University School of Medicine, Stanford, CA, USA. [5]Stanford Diabetes Research Centre, Stanford School of Medicine, Stanford University, Stanford, CA, USA. [6]Department of Pharmacology and Alberta Diabetes Institute, University of Alberta, Edmonton, Alberta, Canada. [7]Oxford NIHR Biomedical Research Centre, Oxford University Hospitals Trust, Oxford, UK. [8]Wellcome Centre for Human Genetics, Nuffield Department of Medicine, University of Oxford, Oxford, UK. [9]These authors contributed equally: Yingying Ye, Elena Navarro-Guerrero. ✉e-mail: agloyn@stanford.edu

types or states may reveal variant-to-gene connections not related to disease pathogenesis) and molecular pleiotropy (variants of interest may regulate transcription of several genes in *cis*, obscuring the identity of the causal transcript). These approaches can generate hypotheses about candidate effectors but typically fall short of providing definitive evidence.

Perturbation studies can provide more compelling evidence of causation, but only if using authentic models and disease-relevant phenotypes[6]. The strongest evidence arises from disease-associated coding variants that provide a readout of the consequences of perturbations of gene and protein function in humans, but the low frequency of most such variants limits this approach[2]. Human cellular models and CRISPR-based technologies provide an attractive alternative for generating genome-wide profiles of the phenotypic consequences of gene perturbation and understanding disease biology[6]. Central to this aspiration is confidence in the disease relevance of a cell type. For T2D, both physiological and epigenomic evidence highlight the central role of pancreatic islets, and consequently the insulin-producing beta cells, in mediating disease risk[3–7]. Substantial differences between rodent and human islets or beta cells argue for the use of human tissue and cell lines[8–13]. We and others have generated large transcriptomic and epigenomic resources in this key human tissue enabling genome-wide integration of genetic and genomic data to identify candidate effector transcripts at T2D GWAS loci[4,7,14,15]. We now complement these resources with a genome-wide CRISPR loss-of-function (LoF) screen in the well-characterized human pancreatic beta cell line EndoC-βH1 to identify genes that regulate insulin content[16–18]. The immortalized EndoC-βH1 cell line displays a multi-omic signature similar to primary human beta cells albeit with distinct characteristics highlighting the fetal and transformed origin of the cell line[18,19]. While insulin content is lower compared to primary islets, EndoC-βH1 cells demonstrate similar electrophysiological and secretory properties, making it a physiologically relevant model to study beta cell function in vitro[17,20–22].

This genome-wide CRISPR screen in a human beta cell line provides cellular evidence for 580 genes as regulators of beta cell function, supports a disease-relevant and likely causal role of 20 genes at T2D loci and identifies the autophagy receptor *CALCOCO2* as a regulator of human beta cell function. We demonstrate that carriers of diabetes risk alleles in *CALCOCO2* have altered insulin secretion and that loss of *CALCOCO2* in human beta cells leads to autophagy-mediated altered insulin granule homeostasis.

## Results

### A pooled CRISPR assay for human beta cell function

Glucose-stimulated insulin secretion is the primary measure of beta cell function but an unsuitable phenotypic readout for pooled high-throughput screening as secreted insulin cannot be linked to its cellular source. Unlike insulin secretion, intracellular insulin content, however, can be quantified using fluorescence-activated cell sorting (FACS). Changes in insulin content have also been associated with T2D-associated risk genes, such as at the *peptidylglycine α-amidating monooxygenase* (*PAM*) locus where T2D risk alleles cause reduced expression and/or function and reduced *PAM* expression is associated with reduced insulin content or for *TCF7L2*, the locus mediating the strongest risk for T2D and a master regulator of insulin production with decreased insulin content in carriers of the risk allele[23–25].

Using a FACS readout, we designed a pooled CRISPR LoF screening pipeline in the human pancreatic beta cell line EndoC-βH1 based on introducing single-gene knockouts (KOs) through a lentiviral genome-wide CRISPR library (Fig. 1a). Cells were sorted into populations containing either low or high levels of intracellular insulin, and stably integrated sgRNAs were identified through next-generation sequencing.

To ensure that the FACS-based insulin content readout in EndoC-βH1 was highly specific and sensitive, we chose an antibody

and protocol based on comparing several complementary strategies (Extended Data Fig. 1). Near complete siRNA-based knockdown of *INS* in EndoC-βH1 resulted in two populations with distinct average fluorescence intensities, validating the sensitivity (Fig. 1b–d). The level of separation was in concordance with an independent alphaLISA quantification of insulin content demonstrating around 46% of residual protein (Fig. 1e). The specificity of the antibody was confirmed in the non-*INS* expressing human cell line HEK293T (Fig. 1f,g). Finally, the screening protocol was validated by targeting three genes in a small-scale screen including the positive control genes *INS* and *PAM*, known to induce a reduction in insulin content upon deletion, and *NMS*, a negative control not expressed in EndoC-βH1 (ref. 23) (Fig. 1h). Compared to an empty vector (EV) control coding for Cas9 only, the cells demonstrated a reduced INS FACS signal and enrichment of *INS* and *PAM* sgRNA within the INS$^{low}$ population, whereas sgRNAs targeting *NMS* showed no difference (Fig. 1i–k). Consistent with its previously shown weaker effect on insulin content, sgRNAs targeting *PAM* demonstrated a highly replicable but lower enrichment compared to sgRNAs targeting *INS*, confirming the suitability of the pipeline for genome-wide screening[23].

### A CRISPR screen identifies regulators of insulin content

We performed two independent genome-wide screen replicates in the human beta cell line EndoC-βH1, using the Toronto KnockOut version 3.0 (TKOv3) CRISPR library targeting 18053 protein-coding human genes with four sgRNAs per gene[26]. Per replicate, a minimum of 670 million cells were lentivirally transduced at a low multiplicity of infection (MOI) with a coverage of around 500 cells per sgRNA (Fig. 1a). Compared to controls, the screening cells demonstrated a wider insulin signal distribution in the FACS, particularly toward lower insulin, indicative of altered insulin content due to CRISPR KO-mediated effects (Fig. 2a,b and Extended Data Fig. 2). Two cell populations were collected, those with low (INS$^{low}$) and those with high insulin content (INS$^{high}$), followed by sgRNA amplification and sequencing (Extended Data Fig. 3). We used the MAGeCK algorithm to compare sgRNA abundance and identify sgRNAs enriched or depleted in INS$^{low}$ cells relative to INS$^{high}$ cells[27]. Enriched sgRNAs identify positive regulator genes leading to a reduction in insulin content upon gene KO, and depleted sgRNAs identify negative regulators associated with an increase in insulin content upon gene KO. To reduce the number of false-positive hits, genes were only classified as screening hits if sgRNAs demonstrated consistent effects across replicates and met the threshold (FDR < 0.1) for at least two of the four sgRNAs in both independent screening replicates. In total, 580 genes fulfilled these stringent criteria, we considered these as robust, reproducible screening hits and took them forward for further evaluation (Supplementary Table 1).

To evaluate the biological relevance of these hits, we asked if our screen was able to identify genes known to be involved in insulin regulation and beta cell function. sgRNAs targeting *INS* were enriched in the INS$^{low}$ population, confirming the sensitivity of the screen (Fig. 2c,e,f). The overall enrichment of *INS* was lower than that of other known regulators of insulin secretion as only three of four sgRNAs induced an effect (Fig. 2c,f). Other established regulators of beta cell function and genes involved in monogenic diabetes or T2D risk were identified among the screening hits, such as *NKX2.2*, *SLC2A2*, *PPP1R15B* and *IGF2* (Fig. 2c,e,f)[28–35]. Non-targeting control sgRNAs showed no effect on insulin content (Fig. 2d). In addition to direct regulators of insulin content, the screen also identified multiple general regulators of transcription and translation with indirect effects on the phenotype (Fig. 2f), all of them classified as common essential genes in CRISPR screens with longer culturing durations[36–38].

### Network analysis of CRISPR screening hits

We next considered whether the screening hits were enriched for functional classifications involved in the regulation of

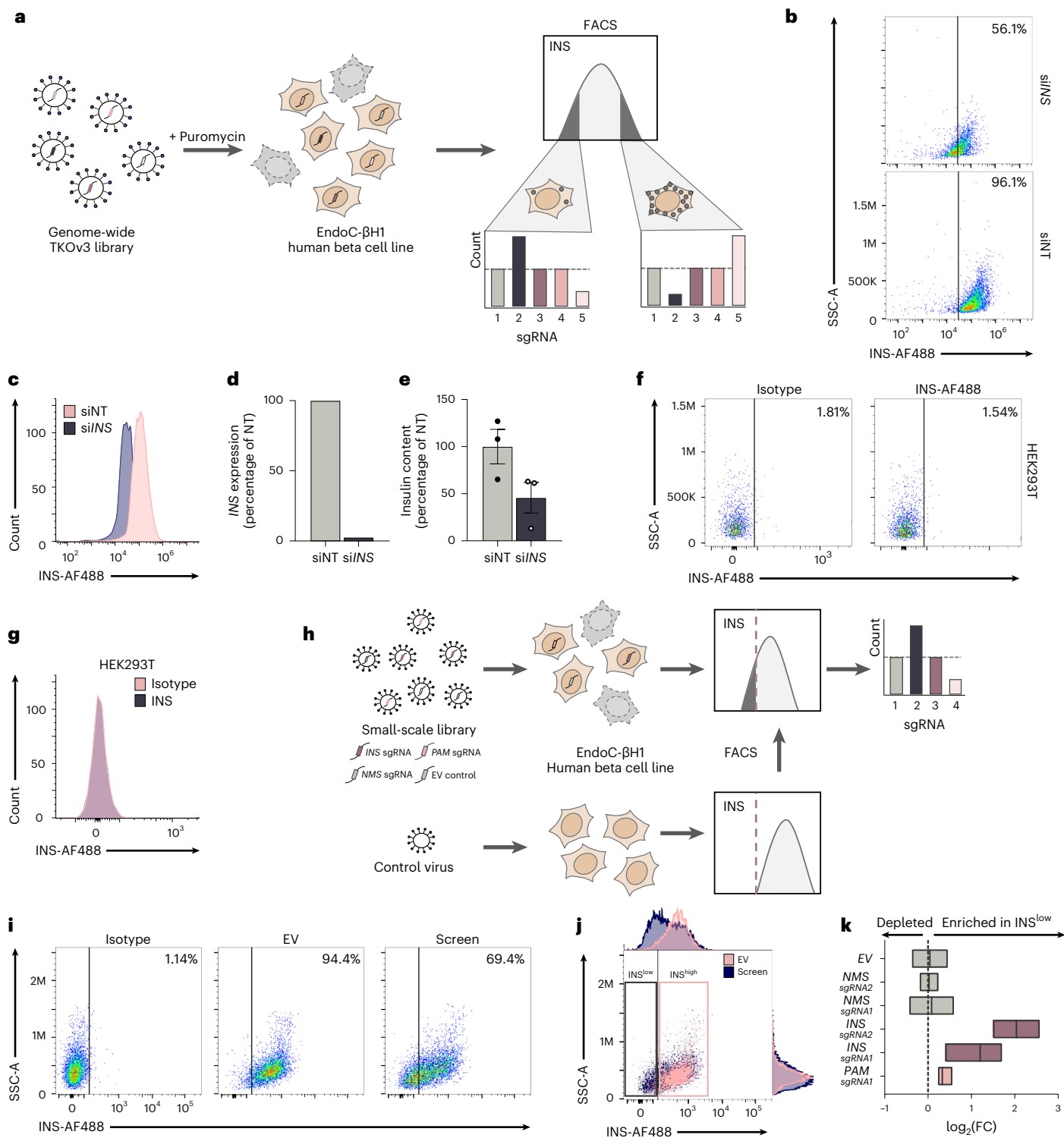

**Fig. 1 | Optimization of a CRISPR screen for insulin content in EndoC-βH1.**
**a**, Pipeline for a genome-wide CRISPR LoF screen in EndoC-βH1 from viral transduction (left) to antibiotic selection (middle) and a final FACS selection followed by sequencing and enrichment analysis of integrated sgRNAs (right). **b**,**c**, FACS staining for intracellular insulin in *INS*-silenced EndoC-βH1 (si*INS*, blue) or their respective non-targeting controls (siNT, pink) as individual plots (**b**) or histogram overlay (**c**). **d**,**e**, The associated mRNA expression of *INS* (**d**) and corresponding levels of intracellular insulin as measured by alphaLISA (**e**). **f**,**g**, FACS staining using the insulin antibody (blue) or isotype control (pink) in the human embryonic kidney cell line HEK293T as individual plots (**f**) or histogram overlay (**g**). **h**, Pipeline for a small-scale CRISPR screen in EndoC-βH1 targeting only three genes *(NMS, INS* and *PAM)* alongside EV control cells. FACS sorting gates for low insulin (dashed line) were determined based on control cells

transduced in parallel. sgRNA enrichment was assessed in INS$^{low}$ compared to INS$^{high}$. **i**,**j**, FACS staining in EndoC-βH1 comparing insulin staining of EV control cells (pink) and cells from the small-scale CRISPR screen (blue) as individual plots (**i**) or overlay (**j**). Respective boxes indicate sorting gates INS$^{low}$ and INS$^{high}$. **k**, sgRNA enrichment for all individual sgRNAs in INS$^{low}$ compared to INS$^{high}$ from control (gray) samples and positive controls (*INS* and *PAM*, purple and pink). The boxplot spans the minimal to maximal values, while the center line depicts the median. All data are mean ± s.e.m. from two (**i**–**k**) or three independent experiments and representative FACS plots are shown with their respective silencing efficiency. Data were analyzed by two-sample *t* test (**e**). No graphical depiction of significance indicates nonsignificant results. NT, non-targeting. log$_2$(FC), log$_2$(fold change); EV, empty vector; SSC-A, side scatter area.

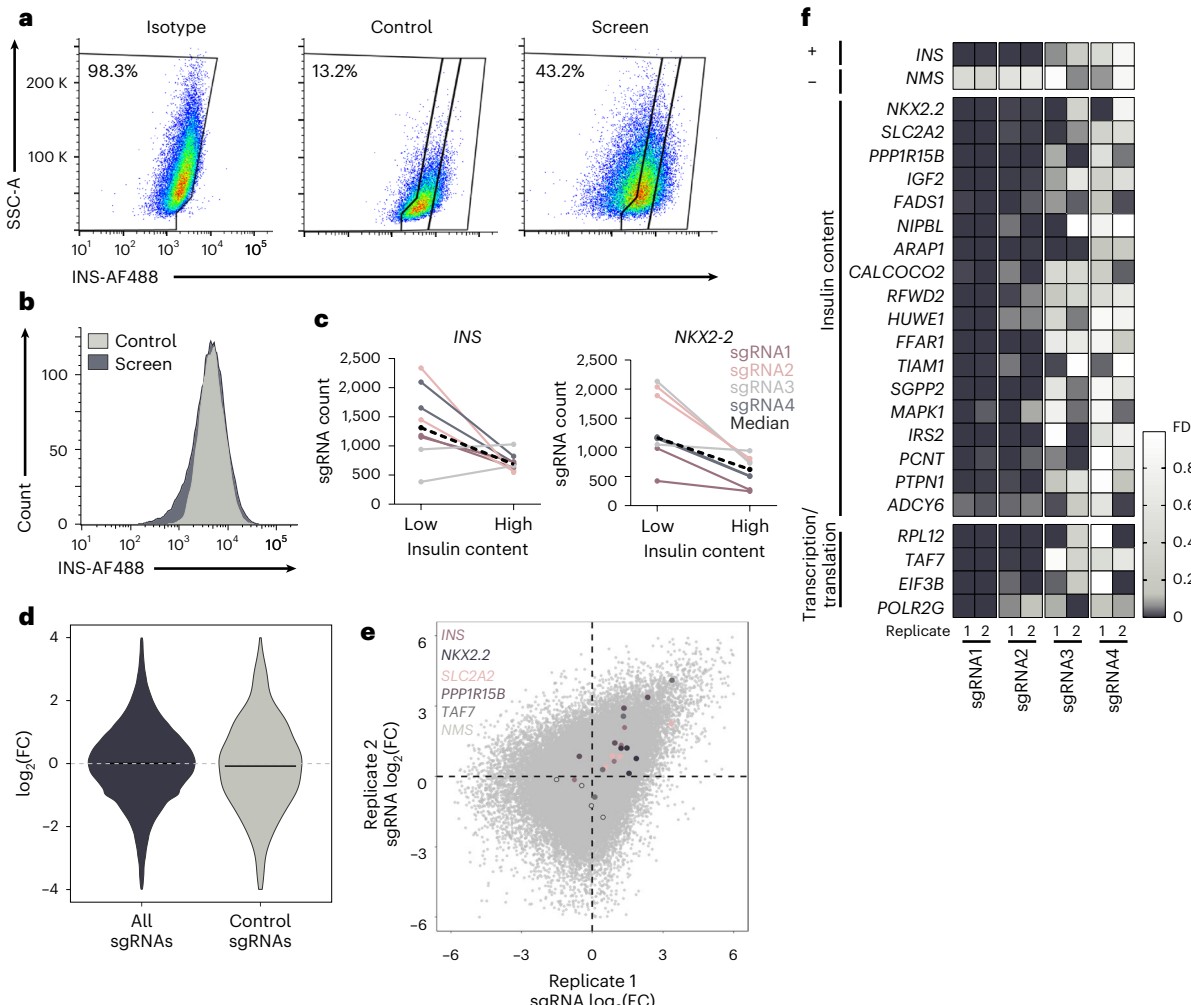

**Fig. 2 | A CRISPR screen for insulin content in EndoC-βH1. a,b**, FACS staining for intracellular insulin in EndoC-βH1 transduced with the CRISPR screening library compared to cells transduced with EV virus or cells stained with isotype matching antibodies shown as individual plots (**a**) and histogram overlay (**b**). **c**, Changes in sgRNA count from low to high insulin content screening sample with each color representing the same sgRNA across the two screen replicates for *INS* and *NKX2-2* (log$_2$(FC) 1.06 and 1.21, respectively). The black dashed line represents the median sgRNA count for this gene. **d**, sgRNA distribution of log$_2$(FC) for all sgRNAs (blue) compared to control sgRNAs targeting LacZ, EGFP and luciferase (gray). The black line indicates the median values for each group.

**e**, sgRNA distribution of log$_2$(FC) within each screening replicate with specific sgRNAs of interest (four per gene) highlighted in individual colors per gene, while overall sgRNA distribution is depicted in light gray. **f**, Individual genes of interest with their respective sgRNAs per replicate, the color ranges from not significant (white) to highly significant (dark blue) based on the FDR thus representing a significant sgRNA enrichment or depletion between INS$^{low}$ and INS$^{high}$. *INS* (+) and *NMS* (−) on the top panel represent positive and negative control genes, respectively. Data are from two independent genome-wide CRISPR screen replicates, and screen FDR values were determined using MAGeCK (**e**, **f**) or through a two-sample *t* test (**d**). log$_2$(FC), log$_2$(fold change); EV, empty vector.

insulin content. Gene Ontology (GO) term and Kyoto Encyclopedia of Genes and Genomes (KEGG) pathway analysis highlighted insulin and beta cell function-associated categories and transcription- and translation-related categories consistent with modifying total intracellular protein levels (Fig. 3a)[39,40]. STRING protein network analysis further emphasized the fundamental role of INS within the screen as a central node with several independent connections (Fig. 3b)[41]. The screening hits were significantly enriched for shared protein networks, providing additional confidence in the sensitivity to identify interrelated complexes (Fig. 3c and Extended Data Fig. 4). Many hits mapped to the functional categories of ubiquitin-mediated proteolysis and autophagy, mitochondrial ATP synthesis, vesicle trafficking and exocytosis, GPCR signaling, lipid metabolism and the MAPK signaling pathway, all of them containing both previously unknown regulators of insulin content and those with known roles in insulin secretion or beta cell function.

## Integration of screening hits with effector transcripts

Next, we sought to use our cellular screen as complementary perturbation evidence to support the assignment of effector genes at T2D loci. We applied three complementary approaches (https://t2d.hugeamp.org) that have been developed to combine genetic association results for T2D with diverse sources of genetic and genomic data to generate lists of the 'effector' gene(s) most likely to mediate the genetic associations[42–45]. The merged effector gene assignments from the three methods generated a list of 336 candidate effector transcripts at T2D loci, with no candidates identified by all three methods. The intersection of this list with the set of screen hits yielded 20 genes (Supplementary Table 2) for which our screen provided biological evidence for a role in a disease-relevant phenotype. Of these 20 genes, 5 (25%) genes are assigned as 'causal' by the curated heuristic effector gene prediction method with a further 4 (20%) genes assigned between strong, moderate and possible.

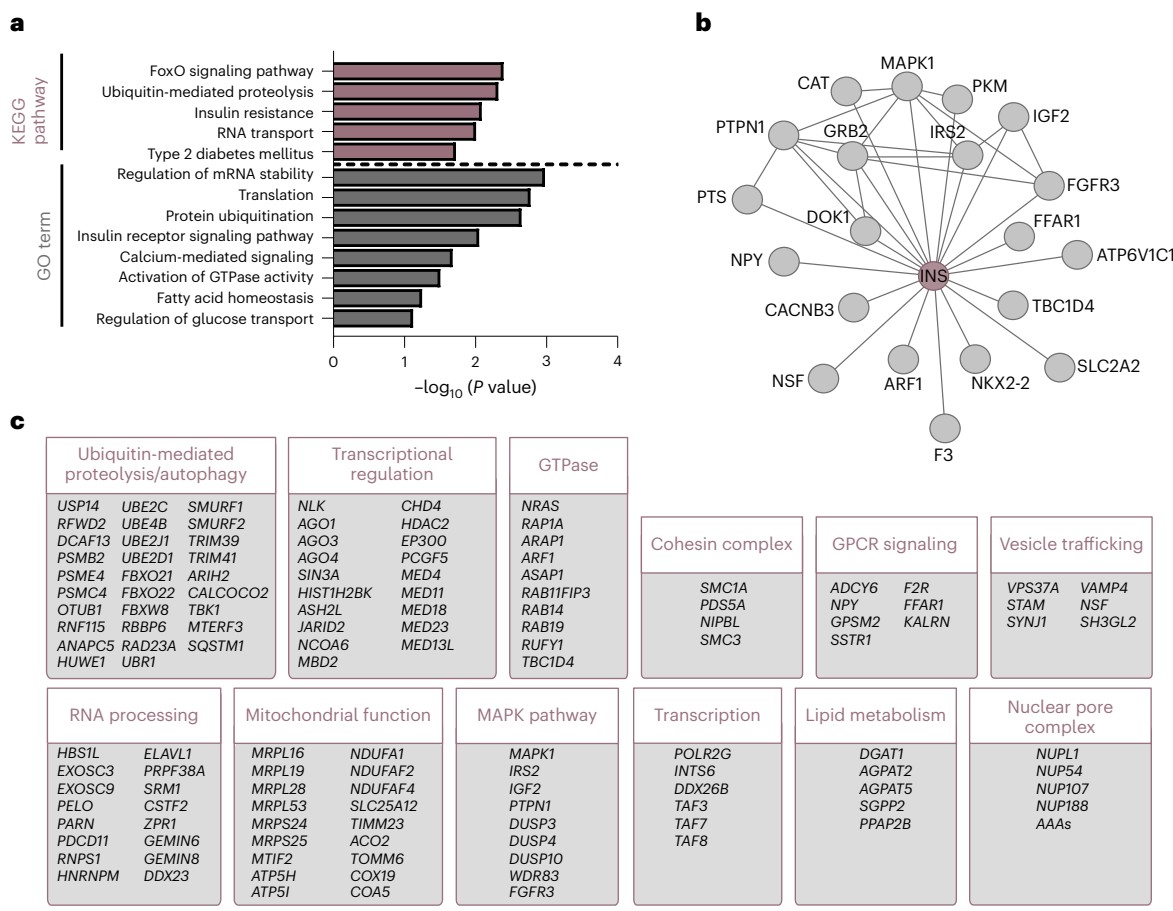

**Fig. 3 | Integration of CRISPR screening hits. a**, Pathway enrichment analysis assessing KEGG pathways or GO terms for biological processes. Selected pathways are shown, ranked by *P* value. **b,c**, STRING pathway analysis showing protein–protein associations including physical and functional interactions between INS and other screening hits (**b**) and for screening hits clustered into functionally associated groups (**c**).

To compare how our perturbation screen performed against other commonly applied approaches for effector transcript prioritization, we explored a relatively modest albeit the largest to date, eQTL study of 420 human islet donors[15]. In this study, colocalization between T2D GWAS and islet *cis*-eQTL signals was supported by two methods at 20 loci, rising to 39 when statistical support for colocalization was required from only one of the approaches. While both approaches (CRISPR screen and eQTL) identified similar numbers of genes, no gene was common among them, suggesting that islet eQTLs are unlikely to cause reduced insulin content and additional cellular phenotypes (for example insulin secretion) may underlly their effect. These observations support the inclusion of multiple approaches for optimal effector gene prioritization and encourage both larger eQTL studies to detect further signals and an expansion of targeted and/or genome-wide screens for additional cellular phenotypes across relevant cell types.

### Loss of *CALCOCO2* reduces insulin content

We focused our follow-up studies on one of the 20 predicted effector genes and the screening hit *CALCOCO2* as its role in human beta cells has not been explored (Fig. 4a)[14,46–49]. *CALCOCO2* has a crucial role in selective autophagy by linking the degradation target to the autophagic machinery[50,51]. So far, it has been shown to initiate autophagy for invading pathogens (xenophagy) and damaged mitochondria (mitophagy)[50–52]. *CALCOCO2* maps to a T2D GWAS locus usually named for the gene *TTLL6*, where fine mapping has resolved the signal to an ~177-kb region containing multiple genes and 118 variants in the credible set[5]. An independent coding variant signal within

*CALCOCO2* (rs10278, p.Pro347Ala) suggests that *CALCOCO2* merits close attention as the effector transcript[5]. Physiological clustering has revealed that this locus is likely to exert its primary effect on disease risk through the modulation of beta cell function[5]. In this CRISPR screen, KO of *CALCOCO2* resulted in a decrease in insulin content (Fig. 4b). None of the other genes at this locus showed an effect on insulin content in the screen, further supporting *CALCOCO2* as the likely effector transcript (Extended Data Fig. 5).

We obtained independent confirmation of the effect of *CALCOCO2* loss through siRNA-based knockdown in EndoC-βH1. Silencing of *CALCOCO2* was highly efficient and induced a mean mRNA and protein reduction of 78.1% and 80.0% (Fig. 4c–e and Extended Data Fig. 7). Cell viability was not affected (Fig. 4f). Intracellular insulin content was assessed using two independent antibody-based detection methods. Highlighting the sensitivity of the CRISPR screen, insulin content was significantly reduced in *CALCOCO2*-silenced cells by 24.0% and 38.0%, measured by alphaLISA and western blot, respectively (Fig. 4g–i). This effect could be rescued upon overexpression of *CALCOCO2* harboring silent mutations to prevent siRNA silencing (Extended Data Fig. 5). The reduction in insulin content was not due to decreased insulin gene expression (Fig. 4j).

To confirm that this screen correctly identified T2D-associated genes modulating beta cell function, we examined additional genes predicted to be causal for T2D which were not identified as screening hits (Extended Data Fig. 6)[43]. *QSER1* and *PLCB3* contain coding variants consistent with a causal role in T2D but with unresolved mechanism of effect or tissue of action[5,53]. Consistent with our negative screening

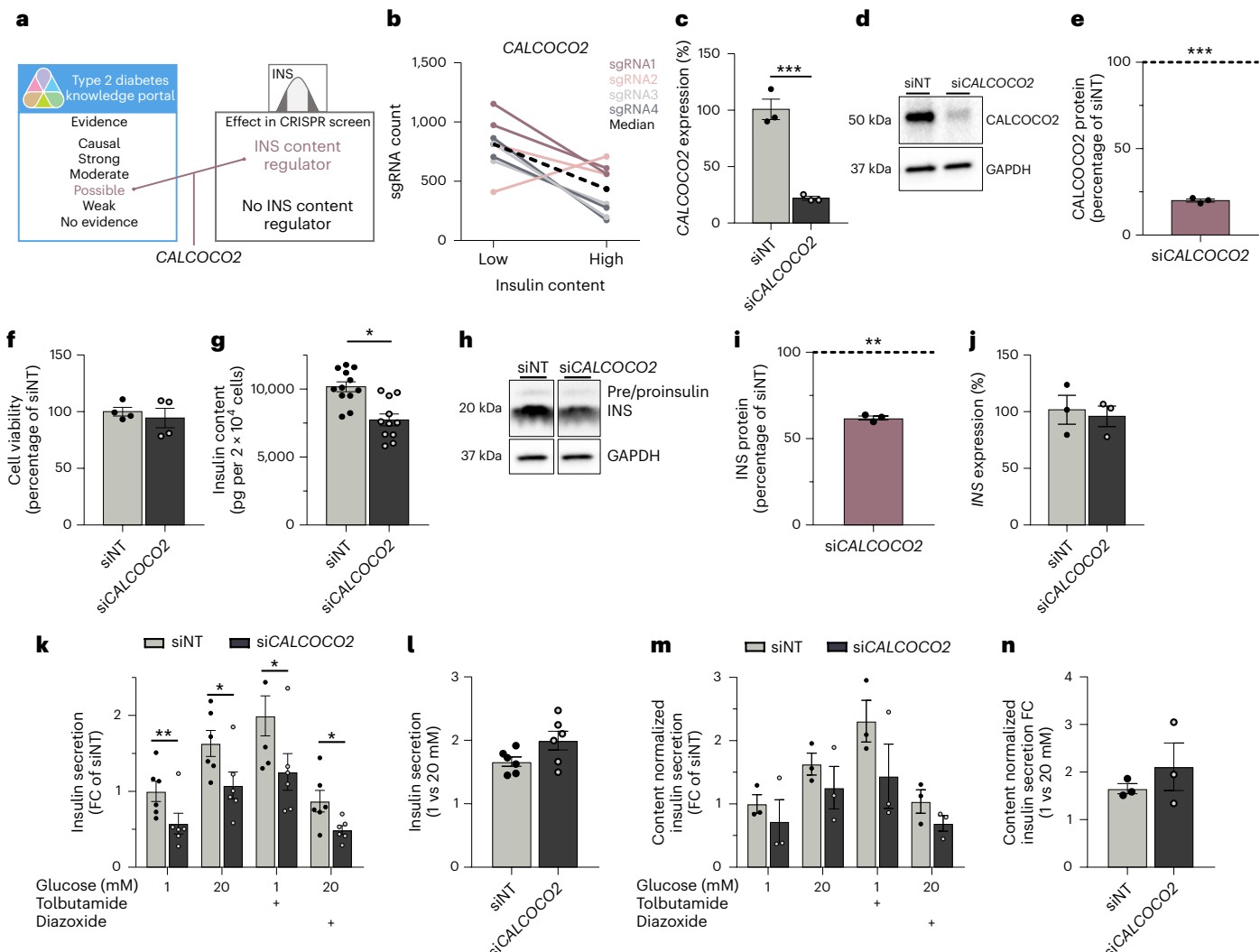

**Fig. 4 | *CALCOCO2* knockdown reduces insulin content in EndoC-βH1.**
**a**, Gene prioritization approach assessing genes with low evidence as T2D effector transcripts (as outlined in the integration approach) that have also been identified as a screening hit, highlighted *CALCOCO2*. **b**, Changes in sgRNA count from low to high insulin content sample with each color representing the same sgRNA across the two screen replicate. The black dashed line represents the median sgRNA count for this gene (log₂(FC): 1.08). **c–n**, All data are from si*CALCOCO2*-treated EndoC-βH1 compared to non-targeting control cells. **c**, mRNA expression of *CALCOCO2* (*P* = 0.0002). **d**, Protein level of CALCOCO2 and its loading control GAPDH. **e**, Quantification of *CALCOCO2* western blot data, normalized to GAPDH and siNT control cells (*P* = 0.0008). **f**, Cell count measurements. **g**, Insulin content in pg per 20,000 cells, measured by alphaLISA (*P* = 0.0166). **h**, Protein level of insulin and its loading control GAPDH. The antibody also detects insulin precursors but only gives a

weak signal compared to mature insulin. **i**, Quantification of insulin and western blot data normalized to GAPDH and siNT control cells (*P* = 0.0015). **j**, mRNA expression of *INS*. **k,m**, Insulin secretion normalized to siNT (**k**; *P* = 0.0051, 0.0336, 0.0282 and 0.0152) or to insulin content and siNT (**m**) in 1 mM, 20 mM, 1 mM + 100 μM tolbutamide or 20 mM glucose + 100 μM diazoxide. **l,n**, Insulin secretion fold change from 1 to 20 mM glucose normalized to siNT (**l**) or to insulin content and siNT (**n**). The protein level is displayed as a percentage of siNT, which is highlighted as a dotted line at 100%. All data are mean ± s.e.m. from three (**c,e,i,j,m,n**), four (**f**), six (**k,l**) or 12 (**g**) independent experiments. Data were analyzed using two-sample *t* test (**c**), one-sample *t* test (**e,i,**), one-way (**f,g,j,l,n**) or two-way (**k,m**) ANOVA with Sidak's multiple comparison test. No graphical depiction of significance indicates nonsignificant results. **P* < 0.05, ***P* < 0.01, ****P* < 0.001. FC, fold change; NT, non-targeting.

results, siRNA knockdown of either gene did not affect intracellular insulin content or insulin secretion. Together, this genome-wide CRISPR screen was able to identify an insulin regulatory function of *CALCOCO2* while distinguishing other phenotypes likely related to islet-cell development (*QSER1*) or insulin action (*PLCB3*).

**CALCOCO2 is required for human beta cell function**
While we have shown that insulin content is reduced upon loss of *CALCOCO2*, altered glucose-stimulated insulin secretion cannot be concluded based on changes in insulin content. We therefore independently assessed the effects of *CALCOCO2* knockdown on insulin secretion. On average, insulin secretion was significantly reduced by

39.3% upon silencing of *CALCOCO2* (Fig. 4k). The glucose stimulation index from low to high glucose was, however, unchanged, indicating an appropriate response to changes in glucose concentrations (Fig. 4l). Normalization to insulin content reduced the difference between *CALCOCO2*-silenced cells and controls to 30.2%. Even though the level of insulin secretion did not reach baseline upon normalization, the effect was not statistically significant and highlighted reduced insulin content as the underlying cause of the reduction in total insulin secretion (Fig. 4m, n).

We extended our studies to assess *CALCOCO2* function in primary human pancreatic islets. Immunofluorescence studies in human pancreas sections confirmed protein expression and localization to the

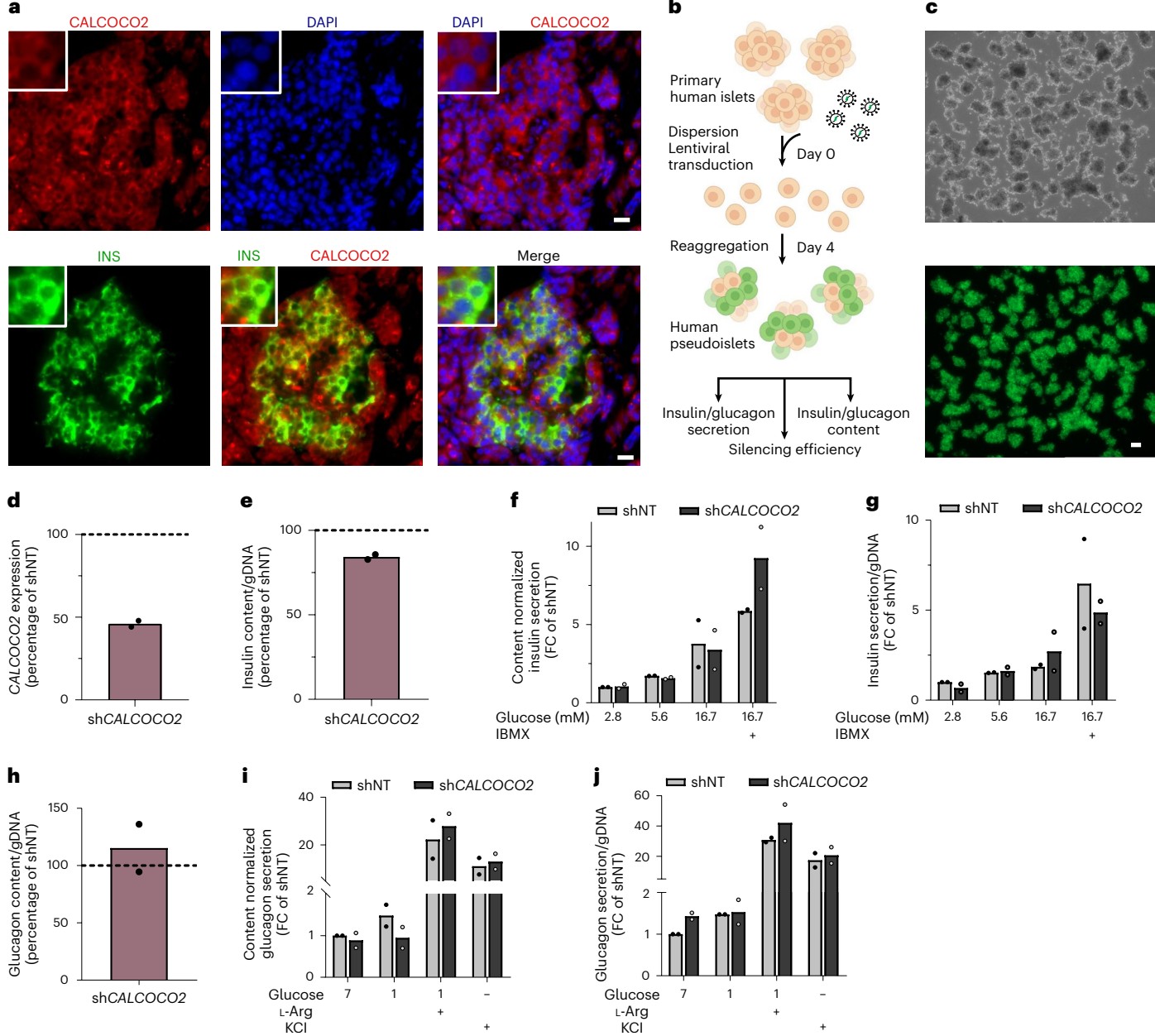

**Fig. 5 | *CALCOCO2* is involved in primary human islet function.**
**a**, Immunofluorescence staining of primary human islets in pancreas sections. Sections were double stained for INS (green) and CALCOCO2 (red). Cell nuclei were counterstained with DAPI (blue). Scale bar is 10 μm. **b**–**j**, All data are from sh*CALCOCO2* pseudoislets compared to non-targeting control pseudoislets (shNT). **b**, Pseudoislet transduction and formation from primary human islets. **c**, sh*CALCOCO2* pseudoislets under bright field (top) and GFP positive pseudoislets (bottom). Scale bar is 100 μm. **d**, mRNA expression of *CALCOCO2*. **e**,**h**, Intracellular insulin (**e**) and glucagon (**h**) content. **f**,**g**,**i**,**j**, Insulin (**f**,**g**) or glucagon (**i**,**j**) secretion normalized to content (**f**,**i**) or to genomic DNA (gDNA) (**g**,**j**) in 2.8, 5.6, 16.7 or 16.7 mM glucose + IBMX for insulin secretion and 7, 1, 1 mM glucose +L-arginine or KCl for glucagon secretion. All data are mean ± s.e.m. from two independent donor and three independent immunofluorescence staining. L-Arg, L-arginine; FC, fold change; NT, non-targeting.

cytoplasm of beta cells (Fig. 5a and Extended Data Fig. 7). To determine whether the effects of *CALCOCO2* loss can be replicated beyond in vitro models and beta cells, we performed shRNA knockdown in primary human islets. Islets were dispersed, transduced with shRNA targeting *CALCOCO2* (sh*CALCOCO2*) and reaggregated to pseudoislets (Fig. 5b). Successful transduction of pseudoislets was confirmed by assessing green fluorescent protein (GFP) coexpression (Fig. 5c). *CALCOCO2* expression was on average reduced by 54.0% compared to non-targeted control shRNA (shNT; Fig. 5d). In line with the results in EndoC-βH1, intracellular insulin demonstrated a modest but consistent reduction of 15.7% (Fig. 5e). Although the decrease in intracellular insulin in

pseudoislets was lower than in EndoC-βH1, it was reduced to the same extent in both the cell line and primary human tissue if normalized to the achieved silencing efficiency, supporting a relationship between *CALCOCO2* levels and function. Pseudoislets, however, did not demonstrate significantly affected insulin secretion (Fig. 5f,g). Stimulation with high glucose and the secretion potentiator IBMX increased secretion of insulin upon normalization to content in sh*CALCOCO2*. This effect was lost upon normalization to genomic DNA and therefore reflects the impact of changes in insulin content on secretion. We similarly assessed glucagon content and secretion to identify potential effects of *CALCOCO2* in alpha cells (Fig. 5h–j). While sh*CALCOCO2*

showed some trend toward blunted glucagon secretion, none of the secretory conditions or glucagon content was significantly changed. Together, our findings show that *CALCOCO2* has an important role in regulating insulin content in human beta cells.

## *CALCOCO2* regulates insulin granule homeostasis

The mechanism underlying *CALCOCO2*'s regulation of insulin content in human beta cells, however, remained to be explored. To perform an unbiased assessment of mRNA expression changes in si*CALCOCO2*-treated EndoC-βH1, we performed RNA-sequencing (RNA-seq) (Supplementary Table 5). Only *CALCOCO2* was identified as being significantly differentially expressed, suggesting a primarily posttranscriptional mechanism on insulin content. Albeit below the significance threshold, the next two genes (*B4GALT5* and *GCNT1*) in the list of those differentially expressed are involved in glycosylation/O-glycan processing (Extended Data Fig. 8). Glycosylation is involved in regulating autophagy and together with *CALCOCO2s* established role as an autophagy receptor, we hypothesized that *CALCOCO2*'s effect on insulin content was autophagy mediated[50,51,54–57].

Immunofluorescence was used to establish the cellular localization of CALCOCO2 in EndoC-βH1 and showed no colocalization with insulin vesicles (Extended Data Fig. 7). We next performed electron microscopy (EM) to evaluate potential cellular ultrastructural changes upon *CALCOCO2* silencing. In line with the reduced total insulin content in pooled measurements, insulin vesicle density was significantly reduced upon *CALCOCO2* silencing (Fig. 6a,b). The proportion of mature insulin granules was increased compared to control cells, indicating a reduction in immature and transitioning granules was driving the effect on total insulin content (Fig. 6a,c). This was further confirmed as mature granule density was unaltered (Fig. 6d). Pooled measurements independently corroborated the structural EM data as intracellular pre/proinsulin (referred to as proinsulin) was significantly reduced upon silencing, an effect that was not lost upon normalization to total insulin content (Fig. 6e,f). This reduction in proinsulin was likely not mediated by altered insulin maturation as expression of genes involved in insulin processing (*PCSK1*, *CPE* and *PCSK2*) was unchanged although enzyme activity was not assessed (Extended Data Fig. 8).

EM analysis revealed that the mitochondrial structure was dramatically altered in si*CALCOCO2* cells with distorted cristae and shape, consistent with increased mitochondrial separation or fission before targeting by autophagy (Fig. 6g,h)[58]. Unexpectedly, mitochondrial ATP production, however, was unaffected upon stimulation with glucose (Fig. 6i). Induction of FCCP-mediated mitochondrial depolarization to induce mitophagy reduced ATP production equally in si*CALCOCO2* and control cells, indicating functional mitophagy (Fig. 6i). The expression levels of the mitophagy mediators *PINK1* and *PARKIN (PARK2)* were not affected (Extended Data Fig. 8). Taken together, there was no indication

that mitochondrial function or externally induced mitophagy was affected upon *CALCOCO2* silencing. Additionally, si*CALCOCO2*-silenced EndoC-βH1 demonstrated an accumulation of vacuolated structures with many containing remnants of cellular components, highlighting alterations in degradation-based mechanisms (Fig. 6j).

We next assessed if this reduction in immature granules and proinsulin was mediated through an effect of *CALCOCO2* on autophagy. Autophagy is highly dynamic and needs to be artificially blocked to accurately measure the amplitude of autophagic flux[59]. We incubated EndoC-βH1 with the lysosomal inhibitor Bafilomycin A1 (BafA1), which prevents autophagosome-lysosome fusion and lysosomal degradation, and assessed LC3-positive puncta which are representative of autophagosomes. While the formation of LC3-positive structures markedly increased in both conditions, the accumulation was more prominent in *CALCOCO2*-silenced cells (Fig. 6k, l). Colocalization of LC3-positive autophagosomes with insulin containing granules further supported an autophagy-mediated reduction of insulin granules upon loss of *CALCOCO2* (Fig. 6k). We further confirmed that the reduction in insulin granules and proinsulin was not mediated by the ubiquitin-proteasome system as the proteasome inhibitor MG132 did not restore proinsulin content (Extended Data Fig. 8).

Our results indicate that *CALCOCO2* is a regulator of insulin granule homeostasis and mediates its observed effect on total insulin content through autophagy-based reduction in proinsulin and immature granules.

## T2D risk alleles in *CALCOCO2* alter primary islet function

Having identified CALCOCO2 as a regulator of insulin content, we next investigated whether carriers of T2D risk alleles at the *CALCOCO2* locus exhibited similar secretory phenotypes to support *CALCOCO2* as the effector transcript at this locus. We assessed insulin content and secretion in intact primary islets from nondiabetic organ donors stratified for their genotypes at two independent T2D-association signals at the *CALCOCO2* locus (Supplementary Table 6). We selected a likely causal coding variant (rs10278) identified in an exome-array study and the lead GWAS variant from the large 99% credible set (118 variants) at the T2D-associated locus (rs35895680)[5]. The rs10278 variant is predicted to result in a substitution of proline with alanine at codon 347 in most isoforms, including the major isoform in human islets (NM_005831)[5]. Its effect allele G, which is associated with reduced risk of T2D, demonstrated altered glucose-stimulated insulin secretion but not altered insulin content in primary islets (Fig. 6m,n). The GWAS lead variant rs35895680 (allele C reduces risk) which is in strong linkage disequilibrium with a variant in the credible set located in the 3′-UTR of *CALCOCO2* did not affect insulin content. Islets from individuals homozygous for the T2D-protective allele, however, demonstrated increased insulin secretion under high-glucose conditions compared to those with one

**Fig. 6 | *CALCOCO2* regulates insulin granule homeostasis. a–l**, si*CALCOCO2*-treated EndoC-βH1 compared to siNT. **a**, Representative images with immature insulin granules (short white arrowheads), mature granules (long white arrowheads) and vacuolic structures (black/white arrowheads). **b**, Quantification of insulin granules per square micrometer (*P* = 0.0137). **c**, Quantification of mature insulin granules normalized to total number of granules per cell (*P* < 0.0001). **d**, Quantification of mature insulin granules normalized to square micrometer. **e**, Proinsulin (ELISA), normalized to siNT (*P* = 0.0033). **f**, Proinsulin normalized to total insulin (ELISA) and siNT (*P* = 0.0081). **g**, Representative EM images with enlargements (top right), mitochondria (white arrowheads) and vacuolic structures (black/white arrowheads). **h**, Quantification of altered mitochondria, normalized to total number of mitochondria (*P* = <0.0001). **i**, ATP measurements after incubation with low/high glucose, with/without the mitophagy inducer FCCP. **j**, Representative images of enlarged vacuolated structures in si*CALCOCO2*, highlighted in **a** and **g** with black/white arrowheads. **k**, Representative staining for LC3 (left) and INS (middle) treated with Bafilomycin A1 (BafA1) or DMSO. **l**, Higher

magnification of the indicated LC3 regions (white box). **m–p**, Intact nondiabetic primary human islets. **m,n**, Carrier (GC) and control (CC) islets for the coding variant rs10278 (protective allele G) in insulin content (**m**) and insulin secretion (**n**). *P* = 0.0029. **o,p**, Islets with indicated genotype at the GWAS lead variant rs35895680 (protective allele C) in insulin content (**o**) and insulin secretion (**p**). *P* = 0.0004. Boxes show the interquartile range between first and third quartiles, the median (horizontal line) and whiskers from the 10th to the 90th percentile (**m–p**). All data are mean ± s.e.m. from three (**a,g,i–l**) or four (**e,f**) independent experiments; 16 (**b,d**), 13 (**c**) and 14 (**h**) siNT cells and 27 (**c**), 40 (**b,d,h**) si*CALCOCO2* cells over three independent experiments and at least 21 donors (Supplementary Table 6; **m–p**). Homozygous individuals for rs10278 (GG) were not analyzed due to low sample size (**m,n**). Scale bar is 1 µm (**a,g**), 0.25 µm (**j**), 20 µm (**k**) and 10 µm (**l**). Data were analyzed using two-sample *t* test (**b–d,h**), two-way ANOVA Sidak's multiple comparison test (**i,m–p**) or one-sample *t* test (**e,f**). No graphical depiction of significance indicates nonsignificant results. \**P* < 0.05, \*\**P* < 0.01, \*\*\**P* < 0.001, \*\*\*\**P* < 0.0001. NT, non-targeting.

copy (Fig. 6o,p). Collectively, we demonstrate that T2D-associated risk variants associated with *CALCOCO2* affect insulin secretion in primary human islets although the relationships of the variant with insulin content and *CALCOCO2* function remain to be explored.

## Discussion

Recent efforts to identify effector transcripts at GWAS association signals have concentrated on integrating disease-relevant transcriptomic and epigenomic datasets or on detailed mechanistic studies at a single locus[4,7,14,15,23,47,60–64]. Genome-wide perturbation datasets in disease-relevant cell types could close the gap between effector transcript and disease biology and enable focused translational efforts. Here, we present a genome-wide pooled CRISPR LoF screen to perform a comprehensive characterization of regulators of insulin content in the human beta cell line EndoC-βH1.

Our CRISPR screening approach was successful in identifying robust modulators of insulin content, as shown through the detection of not only the insulin gene itself but also genes involved in monogenic

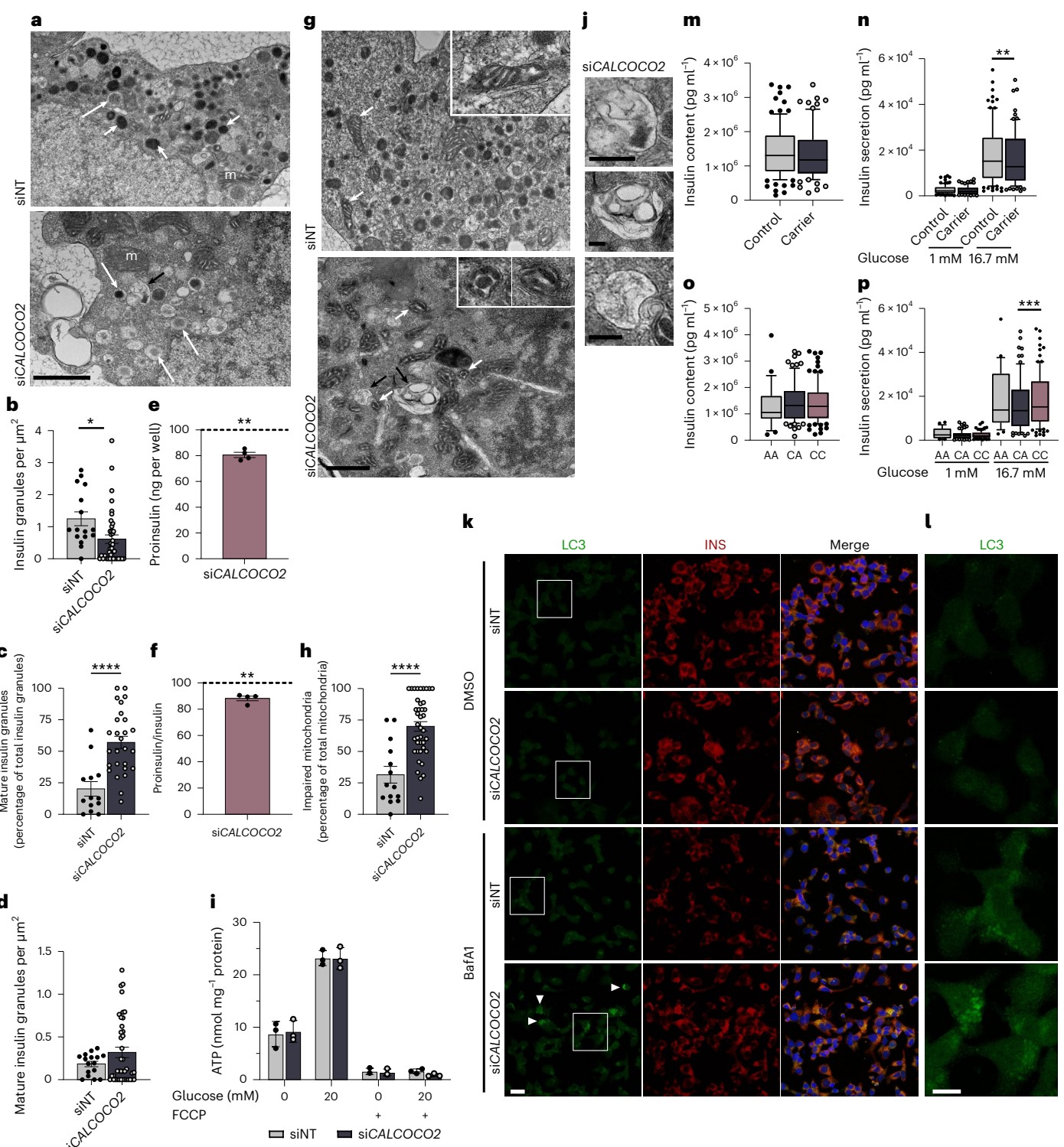

types of diabetes or known regulators of insulin transcription and secretion. Through the integration of our screening hits with prioritization tools, 20 genes were identified as effector transcripts working through beta cell dysfunction. Our data add to a growing number of genome-wide multi-omic datasets which can be harnessed to connect genetic discovery and biological mechanisms and serve as a model for future cellular studies not only in human beta cells. Furthermore, our unbiased genome-wide data will be relevant for other traits/diseases where the pancreatic beta cell has a role, including type 1 diabetes.

We initially identified the T2D risk-associated gene *CALCOCO2* as a CRISPR screening hit and a positive regulator of insulin content. We selected this gene for further study due to its potential to underlie a T2D GWAS signal and a lack of knowledge on its role in beta cell biology. Our focused functional studies confirmed its effect on insulin content and showed an indirect, insulin-content-mediated reduction in insulin secretion. We therefore provide functional data connecting *CALCOCO2* to beta cell function, a likely effect on disease pathogenesis and tissue of action, expanding the evidence for a potential causal role in T2D risk (Supplementary Discussion)[5,53,65].

While *CALCOCO2* has been reported as an essential receptor in Parkin-mediated mitophagy and even though we observed altered mitochondrial ultrastructure, its loss did not result in impaired mitochondrial function in EndoC-βH1 or reduced clearance of mitochondria upon artificial induction of mitophagy, pointing toward a mitochondria-independent mechanism in *CALCOCO2*'s regulation of insulin content[52,66–68]. Instead, our results support an effect on insulin content through degradation of proinsulin-containing immature insulin granules. While the effects of the coding T2D risk variant in *CALCOCO2* on expression and function remain to be established, the observed reduced insulin secretion might represent the long-term effects of altered insulin granule homeostasis through adapted insulin expression and granule biogenesis to compensate for changes in granule degradation, resulting in an affected pool of mature and secretion-ready granules.

While this genome-wide CRISPR screen has identified genes involved in beta cell function and provides a resource for future integration and in-depth studies, it only captures one disease-relevant cellular phenotype in one condition. The screen assessed changes in insulin content under basal glucose conditions, and consequently, genes affecting insulin secretion without modulating insulin content or changes in intracellular content that only occur upon glucose starvation or stimulation were not captured.

This is an important step to accelerate efforts on genome-wide screens, both loss and gain of function in disease-relevant human cell lines for T2D and represents a proof of concept that cellular screens can augment genomic efforts linking variants to regulatory elements and transcripts to bridge the gap between gene expression and disease-relevant cellular biology. In summary, we have developed and performed a genome-wide pooled CRISPR screen in a model of human beta cells, providing a comprehensive perturbation dataset to associate genes of interest with a direction of effect, tissue of action and functional mechanism. We have successfully demonstrated how a genome-wide perturbation set can be used as a prioritization tool for causal genes at T2D GWAS loci and highlight *CALCOCO2* as a modulator of insulin granule homeostasis and beta cell function with a likely causal role in T2D.

## Online content

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

## Methods

### Samples and ethical approval

Human pancreatic islets and pancreas tissue were isolated from deceased donors under ethical approval from the Human Research Ethics Board of the University of Alberta (Pro00013094, Pro00001754) and obtained from the NIDDK-funded Integrated Islet Distribution Program (IIDP) (RRID:SCR_014387) and the National Diabetes Research Institute (NDRI; https://ndriresource.org/). All donors' families gave informed consent for the use of pancreatic tissue in research and were not financially compensated. HEK293T and EndoC-βH1 were obtained from Sigma (12022001) and EndoCells, now known as https://www.humancelldesign.com, respectively[19].

### Cell culture

HEK293T cells were routinely passaged in DMEM 6429 (Sigma-Aldrich) containing 10% fetal calf serum (FCS), 100 U ml$^{-1}$ penicillin and 100 µg ml$^{-1}$ streptomycin. EndoC-βH1 were routinely passaged as described in Supplementary Methods. All cells were mycoplasma-free (Lonza) and grown at 37 °C and 5% $CO_2$. If indicated, EndoC-βH1 cells were treated with 10 nM BafA1, 1 µM MG132 or DMSO for 7 h (all Sigma-Aldrich).

### Human tissue procurement

Deidentified human islets and pancreas samples for immunostaining and functional shRNA experiments were obtained from four non-diabetic organ donors procured through the IIDP and the Alberta Diabetes Institute Islet Core[69]. Human islets for T2D risk allele studies were obtained from 194 nondiabetic organ donors (hemoglobin A1C (HbA1C) < 6) isolated and assessed at the Alberta Diabetes Institute IsletCore as previously described and cultured in DMEM with 10% FCS and 100 U ml$^{-1}$ penicillin/streptomycin (all Thermo Fisher Scientific) at 37 °C and 5% $CO_2$ (refs. 69,70). Human pancreatic tissue was procured through the NDRI. Details of donors used in the study are shown in Supplementary Tables 3 and 6. There was no significant difference in age, sex, BMI or HbA1c between groups. DNA was extracted from digested pancreatic exocrine tissue using the DNeasy Blood & Tissue kit (Qiagen), and genotyping was performed on an Infinium Omni2.5Exome-8 BeadChip array (Illumina).

### Cloning of individual sgRNAs

Single CRISPR sgRNAs were cloned into plentiCRISPRv2 (Addgene, 52961) as described in Supplementary Methods. Briefly, BsmBI compatible tails were added to complementary sgRNA oligonucleotides and annealed (Supplementary Table 4). PlentiCRISPRv2 was digested and ligated with annealed sgRNA oligonucleotides followed by transformation and amplification in competent cells. Correct sgRNA integration was validated through Sanger Sequencing, and plasmids were further used to produce lentivirus.

### Pooled sgRNA library amplification

Toronto human knockout pooled library (TKOv3) containing 71,090 sgRNAs based on a lentiCRISPRv2 backbone was a gift from Jason Moffat, University of Toronto (Addgene, 90294)[26]. The library was transformed and amplified in Endura Competent Cells (Lucigen) as described in Supplementary Methods[26]. Transformation efficiency was determined through serial dilution to ensure sgRNA representation, and plasmids were subsequently extracted using Plasmid Mega kits (Qiagen). A sequencing library was prepared as described below, and library representation was confirmed through sequencing on a NextSeq500 (Illumina) using 75 bp single end reads.

### Lentiviral production and transduction

Lentivirus for individually cloned sgRNAs and pooled libraries was produced and titered as described in Supplementary Methods. Briefly, CRISPR plasmids were cotransfected in 60% confluent HEK293T with packaging vectors pMD2.G (Addgene, 12259) and psPAX2 (Addgene, 12260) using jetPRIME transfection reagents (Polyplus transfection). Viral supernatant was collected at 48 h and 72 h post-transfection and concentrated using ultracentrifugation. The functional viral titer was determined in EndoC-βH1 by measuring the percentage of survival after transduction with different viral dilutions and antibiotic selection. Cell viability was assessed using the CyQUANT Direct Cell Proliferation assay (Invitrogen). The required amount of lentivirus to achieve 26% survival which is representative of an MOI of 0.3 was calculated and used in subsequent small-scale or genome-wide screen transductions. EndoC-βH1 cells were transduced 48 h after seeding for 6 h and selected for 7 d in 4 µg ml$^{-1}$ puromycin with media changes as required.

### Genome-wide CRISPR screen in EndoC-βH1

Two independent genome-wide CRISPR screens were performed consecutively in EndoC-βH1 with independent lentiviral CRISPR library transductions, FACS and sgRNA-seq. Each screen was performed at a coverage of 500 cells per sgRNA and MOI of 0.3. A total of 670 million and 744 million cells were transduced in replicate one and two, respectively. The cells were transduced as described and incubated in 4 µg ml$^{-1}$ puromycin for 7 d with media changes as required followed by collecting, staining and FACS analysis.

### INS intracellular staining and FACS

EndoC-βH1 cells were collected and incubated with LIVE/DEAD Fixable Far Red Dead Cell Stain (Thermo Fisher Scientific) for 30 min at room temperature and washed in 1% BSA in PBS. The cells were fixed and permeabilized using the BD Cytofix/Cytoperm kit (BD Biosciences) for 20 min at 4 °C and washed using Perm/Wash Buffer (BD Biosciences). Staining with primary antibodies was performed overnight at 4 °C followed by incubation with suitable secondary antibodies diluted in Perm/Wash Buffer (Supplementary Table 4). Upon testing several INS targeting antibodies, a rabbit monoclonal anti-INS antibody from Cell Signaling (3014) was used in the genome-wide screen. The samples were filtered through a 70-µm cell strainer and sorted on a FACSAria III (BD Biosciences) using a 100-µm nozzle. Isotype and transduction controls stained with each antibody alone were analyzed alongside the samples. Samples were gated based on live and single cells before INS level assessment. Gates for cells with reduced INS level (INS$^{low}$) were set based on the isotype control and the highest 10% of INS expressing cells were sorted as INS$^{high}$. Sorted cell samples were stored at −80 °C until DNA extraction. Flow cytometry data were analyzed using Flowjo 10.6 (BD Biosciences).

### Preparation of genomic DNA for next-generation sequencing

DNA was extracted from frozen FACS-sorted samples using QIAamp Blood Maxi/Mini Kit (Qiagen) and processed for sequencing in a two-step PCR approach. Integrated sgRNAs were amplified using Q5 polymerase (NEB) with 2.5 µg DNA input per reaction and an optimized number of cycles per sample. Specific Illumina TruSeq adapters were attached to each sample using Q5 polymerase (NEB) with an optimized number of cycles (Supplementary Table 4). PCR products were run on a 2% gel and purified using QIAquick Gel extraction kit (Qiagen). Each Illumina library sample was qPCR based quantified using the KAPA Library Quantification Kit (Roche) before pooling, and multiplexed sequencing on a NEXTSeq500 (Illumina) was performed as 75 bp reads with standard Illumina sequencing primer and PhiX to approximately 20% spike-in.

### Analysis of pooled CRISPR screen

Raw fastq sequencing read files were merged for each sample using the 'cat' command. To identify enriched and depleted sgRNAs in this CRISPR screen, we used the MAGeCK (v0.5.9.2) algorithm[27]. Briefly, the 'count' command was used to extract and count sgRNA sequences from the raw fastq files based on the TKOv3 library input yielding

a mean read count of 981 aligned reads per sgRNA (Extended Data Fig. 3). sgRNAs with low read counts of less than 10 and sgRNAs mapping to genes that were not expressed in EndoC-βH1 were excluded. The 'test' command was used with the paired module to assess and analyze sgRNA enrichment or depletion between INS[low] and INS[high] populations. The analysis median normalizes read counts across samples to account for varying read count depth and applies a mean-variance model to identify significant sgRNA enrichment or depletion. MAGeCK multiple-testing adjusted sgRNA-level enrichment scores were the basis for gene-level hit selection. We applied additional stringent criteria to only prioritize hits with highly consistent effects across both replicates requiring genes to have an FDR < 0.1 for at least two of four sgRNAs in both independent replicates. CRISPR screen analysis was performed in Python 3.8 and R 3.5.

### Gene set enrichment and pathway analysis
Enriched GO and KEGG pathways within the screening hits were determined using the Database for Annotation, Visualization and Integrated Discovery 6.8 with homo sapiens as list and background input[39]. Protein connectivity networks based on physical and functional interactions were identified among screening hits using STRING v11 (ref. [41]). Only interactions with a high confidence score of ≥0.9 were selected.

### Gene silencing in EndoC-βH1
Forward siRNA-based silencing in EndoC-βH1 was performed at 24 h after plating using ON-TARGET plus SMARTpool siRNAs or a non-targeting control (Dharmacon; Supplementary Table 4) at 15-nM final concentration. siRNAs were pre-incubated with 0.4% Lipofectamine RNAiMAX (Invitrogen) in Opti-MEM reduced serum-free medium (Gibco) for 15 min at room temperature before dropwise addition to the culture. Cells were assayed or collected 72 h post-transfection. *CALCOCO2* silencing in experiments presented in Fig. 6 was performed at a final concentration of 50 nM siRNA and assessed after 96 h.

### *CALCOCO2* silencing in primary human islets
Lentiviral constructs coding for shRNAs targeting human *CALCOCO2* were obtained from Dharmacon (Supplementary Table 4), and virus was produced as described above. Primary islets were dispersed into single cells by enzymatic digestion (Accumax, Invitrogen), and $1 \times 10^6$ cells were transduced with $1 \times 10^9$ viral units per 1 mL. Transduced islet cells were cultured in ultralow attachment well plates for 5 d before further analysis.

### *CALCOCO2* overexpression in EndoC-βH1
Silent mutations were introduced into the CDS of human *CALCOCO2* at four target sites based on the target regions of the ON-TARGET plus SMARTpool *CALCOCO2* siRNA. Restriction sites containing EcoRI and BamHI motives and Kozak sequences were added to the 5′ and 3′ terminus of the CDS. The gene fragment was synthesized (IDT) and cloned into the plasmid pCDH-CMV-MCS-EF1-copGFP. Empty plasmid served as a control in subsequent experiments. Lentivirus was produced as described above, and EndoC-βH1 cells were transduced at an MOI of 0.5.

### qPCR gene expression analysis
RNA for gene expression analysis from EndoC-βH1 was extracted using TRIzol reagent (Invitrogen) and synthesized into complementary DNA using the GoScript Reverse Transcriptase System (Promega). RNA for gene expression analysis from primary human pseudoislets was extracted using the PicoPure RNA isolation kit (Life Technologies), and cDNA was synthesized using the Maxima first strand cDNA synthesis kit (Thermo Fisher Scientific), according to the manufacturer's instructions. Quantitative qPCR (qPCR) was performed using TaqMan real-time PCR assays on a 7900HT Fast Real-Time PCR System (all Applied Biosystems, Supplementary Table 4). Ct-values were analyzed using the ΔΔCt method, and target genes were normalized to the combined average of the housekeeping *PPIA*, *GAPDH* and *TBP*. *CALCOCO2* expression in EndoC-βH1 and primary islets was extracted from previously published and analyzed RNA-seq data[71,72].

### RNA-seq
RNA-seq was performed using PolyA capture with an average of >20 million paired-end reads on a NEXTSeq500 (Illumina). Fastq files were aligned to human genome reference (GRCh38) using STAR v2.7.9a (Spliced Transcripts Alignment to Reference) with ENSEMBL gene annotations (v101). Gene expression levels were counted using feature-Counts (v2.0.1) on exonic reads. Differential expression was compared using the Wald test in DESeq2 (v1.26.0). *P* values were adjusted using the Benjamini and Hochberg method.

### SYBR Green-based qPCR
sgRNA integrations in the small-scale screen were quantified using SYBR Green-based quantitative PCR and primer targeting the respective sgRNA sequences (Supplementary Table 4). DNA from FACS-sorted EndoC-βH1 was extracted as described. Each sample was prepared using 20 ng of total DNA and SYBR Green PCR Master mix (Bio-Rad), following the manufacturer's instructions. The samples were amplified and analyzed as described for gene expression experiments.

### Insulin secretion assay in EndoC-βH1
Silenced EndoC-βH1 cells were starved overnight at 48 h post-transfection in culture medium containing 2.8 M glucose followed by 30 min incubation in 0 mM glucose. Insulin secretion was initiated through static incubations in the indicated glucose or secretagogues conditions for 1 h. Insulin containing supernatant was collected, and cells were lysed in ice-cold acid ethanol to release intracellular insulin. Insulin was quantified using the Insulin (human) AlphaLISA Detection kit and the EnSpire Alpha Plate Reader (both Perkin Elmer) based on 1:10 and 1:200 dilutions for supernatant and insulin content, respectively. Intracellular proinsulin was quantified using the Proinsulin ELISA kit (Mercodia). Secreted insulin was normalized to the level of intracellular insulin content or cell count, which was measured before cell lysis using the CyQUANT direct cell proliferation assay (Invitrogen).

### Insulin secretion assays in primary human islets
Groups of 15 islets in triplicate were pre-incubated for 1 h at 37 °C in Krebs Ringer Buffer (KRB) (115 mM NaCl; 5 mM KCl; 24 mM NaHCO₃; 2.5 mM CaCl₂; 1 mM MgCl₂; 10 mM HEPES (pH 7.4); 0.1% BSA) with 1 mM glucose. Islets were subsequently incubated for 1 h in 1 mM glucose KRB followed by a 1 h stimulation in 16.7 mM glucose KRB. Supernatants were removed and total insulin content was extracted from the islet pellet using acid ethanol. Samples were stored at −20 °C and assayed for insulin via chemiluminescence using the STELLUX Chemi Human Insulin ELISA (Alpco).

### Insulin and glucagon secretion assays in primary pseudoislets
Batches of 25 pseudoislets were used per donor for in vitro secretion assays in RPMI 1640 (Gibco) supplemented with different levels of glucose. For insulin secretion assays, pseudoislets were pre-incubated for 1 h at 2.8 mM glucose followed by incubation at 2.8, 5.6, 16.7 and 16.7 mM + IBMX glucose concentrations for 60 min each. For glucagon secretion assays, pseudoislets were incubated at 7, 1, 1 mM + 10 mM arginine glucose concentrations for 60 min each. Supernatants were collected, and cells were lysed in acid ethanol to extract insulin and glucagon content. Human insulin in the supernatants and pseudoislet lysates was quantified using a human insulin ELISA kit (Mercodia). Human glucagon in the supernatants and pseudoislet lysates was quantified using a human glucagon ELISA kit (Mercodia). Secreted insulin and glucagon levels are presented as a percentage of gDNA, quantified from the pseudoislet lysates.

## ATP measurement

EndoC-βH1 were incubated for 1 h at 37 °C in glucose secretion assay buffer (SAB) (114 mM NaCl, 4.7 mM KCl, 1.2 mM KH$_2$PO$_4$, 1.16 mM MgSO$_4$, 25.5 mM NaHCO$_3$, 2.6 mM CaCl$_2$, 20 mM HEPES (pH 7.3), 0.2% BSA). Following starvation, the cells were stimulated for 40 min with 0 mM or 20 mM glucose-containing SAB supplemented with either DMSO or 10 μM FCCP and lysed in 100 μl ATP assay buffer (Biovision). ATP content in the lysates was measured using a luminometric assay following the manufacturer's instructions (Biovision) and normalized to total protein contents as determined by BCA protein assay (Pierce).

## Western Blot analysis

Whole-cell protein extracts were obtained from cell pellets through lysis in RIPA buffer (50 mM Tris pH 7.4, 150 mM NaCl, 1% Triton X-100, 0.5% sodium deoxycholate, 0.1% SDS) containing 1X protease inhibitor cocktail (Roche). Protein concentrations were quantified using the DC protein assay (Bio-Rad), 10 μg of total protein were mixed with sample buffer (4× Laemmli Buffer (Bio-Rad), 10% β-mercaptoethanol (Sigma-Aldrich), and boiled for 10 min at 80 °C. Denatured samples were run on a 4–20% Criterion TGX Stain-Free Precast Gel for 15 min at 300 V in Tris-glycine-SDS buffer and transferred to a 0.2 μm polyvinylidene difluoride using the Trans-Blot Turbo Transfer System (all Bio-Rad). Unspecific antibody binding was blocked through incubation with 3% BSA in Tris-buffered saline containing 0.1% Tween-20 (TBST) for 1 h at room temperature. Primary antibody incubations were performed at 4 °C overnight followed by incubation with HRP-conjugated secondary antibodies for 1 h at room temperature (Supplementary Table 4). Blots were imaged using the ChemiDoc MP Imaging System (Bio-Rad) and reprobed as described using loading control antibodies of appropriate size. Image Lab 6.0 software (Bio-Rad) was used to quantify protein bands and proteins of interest were normalized to their respective loading control on the same blot.

Immunofluorescence analysis in human pancreatic tissue. Pancreata were fixed in 4% paraformaldehyde overnight, cryoembedded, and sections of 4 μm were prepared. Tissue sections were dipped in distilled water, boiled for 20 min in target retrieval solution (Dako), permeabilized for 10 min at room temperature using 1% PBS-Triton X-100 (Sigma-Aldrich) and unspecific antibody binding was blocked using 1% BSA (Roche), 0.2% nonfat milk, 0.5% Triton X-100, 1% DMSO in PBS (all Sigma-Aldrich). Primary antibody incubations were performed overnight at 4 °C followed by incubation with secondary Alexa Fluor-conjugated antibodies at room temperature for 1 h (Supplementary Table 4). The slides were mounted using VectaShield mounting media containing DAPI (Vector Laboratories) and imaged on a Zeiss AxioM1 (Zeiss) microscope using an ×20 and ×40 objective. ImageJ 1.52b software was used to prepare immunofluorescence images.

## Immunofluorescence analysis in EndoC-βH1

EndoC-βH1 cells were plated in Nunc Lab-Tek II Chamber Slides (Thermo Fisher Scientific) and fixed with 3% PFA-K-PIPES and 3% PFA-Na$_2$BO$_4$ for 5 and 10 min, respectively, followed by permeabilization with 0.1% Triton X-100 for 30 min at room temperature (all Sigma-Aldrich). The cells were incubated in blocking solution containing 5% donkey serum (Jackson ImmunoResearch) in PBS for 30 min. Primary antibodies were diluted in blocking solution and slides incubated overnight at 4 °C followed by incubation with secondary Alexa Fluor-conjugated antibodies and DAPI (MP Biomedicals) at room temperature for 1 h (Supplementary Table 4). Imaging was performed on a STELLARIS 8 Confocal Microscope (Leica Microsystems).

## Transmission EM

Control and silenced EndoC-βH1 cells were resuspended in 10% gelatin in 0.1 M sodium cacodylate buffer (pH 7.4) at 37 °C and allowed to equilibrate for 5 min. Cells were pelleted, excess gelatin removed, chilled to 4 °C and incubated with cold 1% osmium tetroxide rotating for 2 h at 4 °C. They were washed three times with cold ultrafiltered water, en-bloc stained overnight in 1% uranyl acetate at 4 °C while rotating. The samples were dehydrated in a series of ethanol washes (30%, 50%, 70% and 95%) for 20 min each at 4 °C, equilibrated to room temperature, washed in 100% ethanol twice and incubated in propylene oxide (PO) for 15 min. The samples were infiltrated with EMbed-812 resin mixed with PO at a 1:2, 1:1 and 2:1 ratio for 2 h each. They were incubated in 2:1 resin to PO overnight rotating at room temperature, placed into EMbed-812 for 4 h and subsequently into molds with fresh resin followed by incubation at 65 °C overnight. Sections of 80 nm were picked up on formvar/carbon-coated 100 mesh Cu grids and stained for 30 s in 3.5% uranyl acetate in 50% acetone followed by 0.2% lead citrate for 3 min. Imaging was performed using JEM-1400 120 kV (JEOL) and images were taken using a Gatan Orius 4k × 4k digital camera. Image quantification was independently performed by three blinded assessors.

## Statistics and reproducibility

Statistical analyses were performed in Prism 8.1 (GraphPad Software). A number of biological independent replicates are shown as individual data points with exact numbers indicated in the respective figure legends. The error bars represent the standard error of the mean. No statistical method was used to predetermine sample size. Significant outlier in primary islet assessments from T2D risk carriers was excluded based on the pre-established ROUT outlier test ($Q$ = 1%). If appropriate, fold changes were plotted but statistical analysis was performed on log-transformed values. If required, such as for imaging analysis, assessors were blinded to sample allocation. Normally distributed variables were compared between two or more groups using two-sample Student's $t$ test or one-way ANOVA followed by Sidak's multiple comparison test. Samples that were normalized to their respective control within a single replicate were analyzed using a one-sample Student's $t$ test.

## Reporting summary

Further information on research design is available in the Nature Portfolio Reporting Summary linked to this article.

## Data availability

Fastq sequencing files from the CRISPR screen have been deposited in the European Nucleotide Archive (ENA) at EMBL-EBI under accession number PRJEB44712. Fastq sequencing files from RNA-Sequencing experiments for si*CALCOCO2* samples have been deposited in the European Genome-phenome Archive (EGA) under study number EGAS00001006127 and EndoC-βh1 expression data from a previously published study can be accessed under PRJEB15283 (ENA)[71]. The data are freely available to download while the processed counts can be found in Supplementary Dataset 1. RNA-seq data were aligned to the human genome reference GRCh38 (ftp://ftp.ensembl.org/pub/release-101/fasta/homo_sapiens/dna/Homo_sapiens.GRCh38.dna.primary_assembly.fa.gz) and counted with the gene annotation (ftp://ftp.ensembl.org/pub/release-101/gtf/homo_sapiens/Homo_sapiens.GRCh38.101.gtf.gz) downloaded from Ensembl database. Source data of unprocessed blots and Extended Data can be accessed online. Source data are provided with this paper.

## Code availability

The source code of MAGeCK is freely available at http://liulab.dfci.harvard.edu/Mageck and differential gene expression analysis was performed using the DESeq2 R package. Respective code is available at https://doi.org/10.5281/zenodo.7226348.

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

## Acknowledgements

A.L.G. is a Wellcome Senior Fellow in Basic Biomedical Science. A.K.R and S.K.T. are Radcliffe Department of Medicine Scholars and A.K.R received a travel fellowship from the European Foundation for the Study of Diabetes (EFSD). A.L.G. was funded by the Wellcome (200837), the National Institute of Diabetes and Digestive and Kidney Diseases (NIDDK) (U01-DK105535, U01-DK085545, UM1DK126185) and the Stanford Diabetes Research Center (NIDDK award P30DK116074). The project was further supported by the National Center for Research Resources (NCRR) (1S10RR026780-01) and National Institutes of Health (NIH) (grants R01 DK108817 and R01 DK107507 to S.K.K.) and a Juvenile Diabetes Research Foundation (JDRF fellowship) (to R.J.B.). P.E.M. is supported by the Canada Research Chair in Islet Biology and in part by the Canadian Institutes of Health Research (MOP, 148451). We thank past and current members of the Gloyn lab for advice and encouragement, Dylan Jones for library amplification, Ruddy Montandon for FACS sorting, the Oxford Genomics Centre (Wellcome Centre for Human Genetics) for sequencing, Gene Vector and Virus Core (GVVC) for generating the lentivirus with *CALCOCO2* vector plasmids, the EM core in Cell Sciences Imaging Facility (CSIF) for the preparation of EM samples and the project described was supported, in part, by ARRA award 1S10RR026780-01 from the National Center for Research Resources (NCRR). We thank Karla Kirkegaard for advice with the LC3 staining. We thank Yan Hang, Mollie Friedlander and the Islet Core of the Stanford Diabetes Research Centre for support with hormone secretion assays. We gratefully acknowledge organ donors and their families, and islet procurement through the Alberta Diabetes Institute Islet Core with the assistance of the Human Organ Procurement and Exchange (HOPE) program, Trillium Gift of Life Network (TGLN) and other Canadian organ procurement organizations, the Integrated Islet Distribution Program (U.S. NIH UC4 DK098085), the National Disease Research Interchange, and the International Institute for the Advancement of Medicine. All donors' families gave informed consent for the use of pancreatic tissue in research. The funders had no role in study design, data collection and analysis, decision to publish or preparation of the manuscript.

## Author contributions

A.K.R., Y.Y., S.K.T. and A.L.G. conceived the study. A.K.R., E.N-G. and S.K.T. performed CRISPR screening optimization experiments. A.K.R., E.N-G., R.B. and S.N. performed the CRISPR screen. E.N.G. and R.B performed sequencing library preparation. A.K.R. analyzed the CRISPR screening data. A.W-A. contributed to the CRISPR screening analysis. A.K.R. and A.L.G. performed data integration analysis. A.K.R. performed the validation of *CALCOCO2, QSER1* and *PLCB3* in beta cells. A.F.S., A.B., J.L., N.S., J.M.F., P.E.M. and A.L.G. assessed *CALCOCO2* risk alleles in human islets. Y.Y., V.R., A.P. and J.Y. performed *CALCOCO2* follow-up studies in EndoC-βH1. A.K.R. and R.J.B. performed islet immunostainings. R.J.B., A.K.R. and V.R. performed islet shRNA knockdown experiments. H.S. analyzed RNA-seq data. S.K.K., J.M.F., D.E., P.E.M. and A.L.G. supervised the research. A.K.R. and A.L.G. wrote the manuscript. All authors approved the final draft of the manuscript.

## Competing interests

The authors declare the following competing interests: A.K.R. is now an employee of AstraZeneca, S.K.T. is now an employee of Vertex Pharmaceuticals and A.W-A. is now an employee of Genomics plc. D.E. and E.N-G are now also affiliated with XCellomics. All experimental work by A.K.R., S.K.T. and A.W-A. was carried out under employment at the University of Oxford. A.K.R. holds stocks in Astra Zeneca. A.L.G.'s spouse holds stock options in Roche. The other authors declare no competing interests.

## Additional information

**Extended data** is available for this paper at https://doi.org/10.1038/s41588-022-01261-2.

**Correspondence and requests for materials** should be addressed to Anna L. Gloyn.

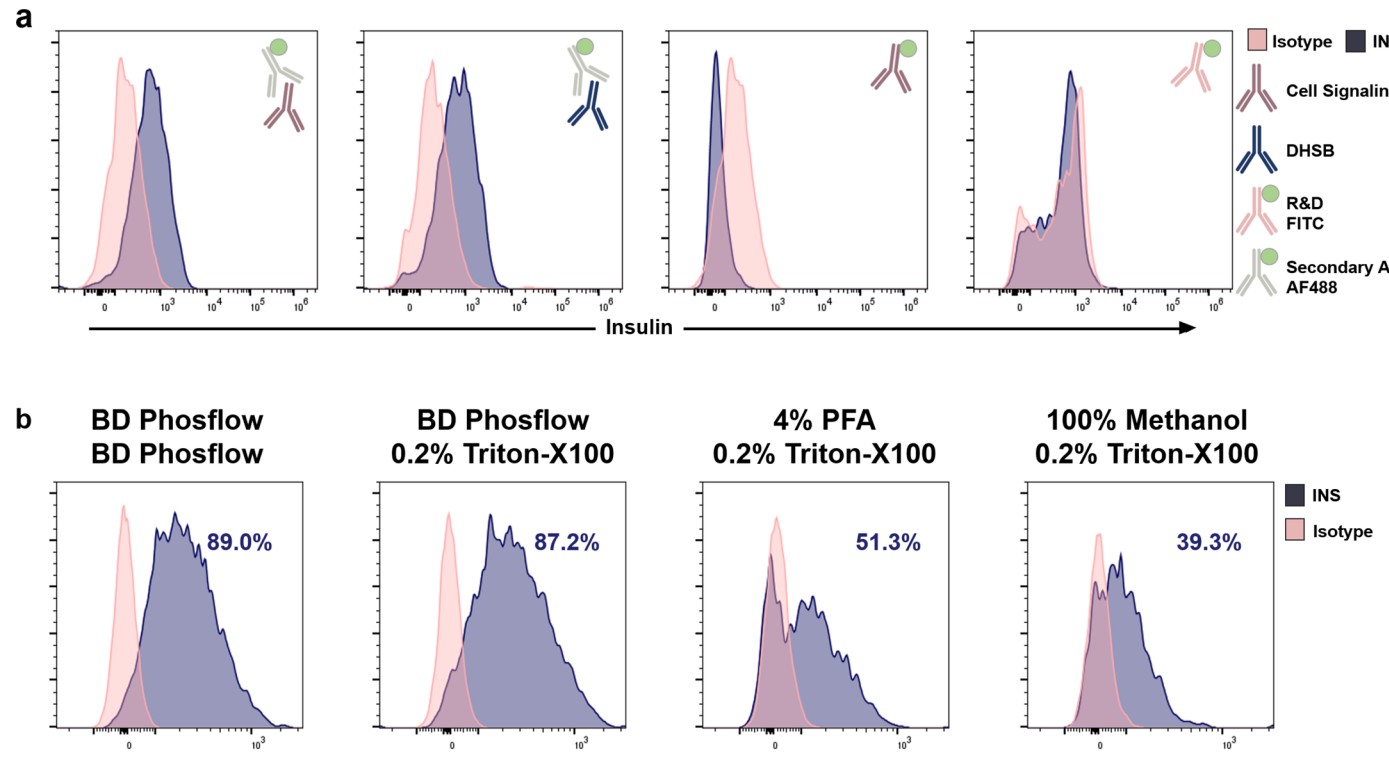

**Extended Data Fig. 1 | FACS optimization for intracellular INS staining.** Histograms showing different FACS staining strategies in EndoC-βH1 comparing intracellular insulin (blue graph) and respective isotype controls (pink graph) using (**a**) different monoclonal primary antibodies as indicated (purple, blue, pink antibody) in combination with a secondary Ab (gray) conjugated to AF488 (green) or direct conjugation to FITC (green) and (**b**) using different staining protocols with various fixation (top row) or permeabilization buffers (bottom row) as indicated. Three independent experiments were performed and representative histograms are shown. The genome-wide CRISPR screen was performed using the anti-INS antibody from Cell Signaling and the commercial fixation and permeabilisation Cytofix/Cytoperm reagents and protocol from BD Sciences.

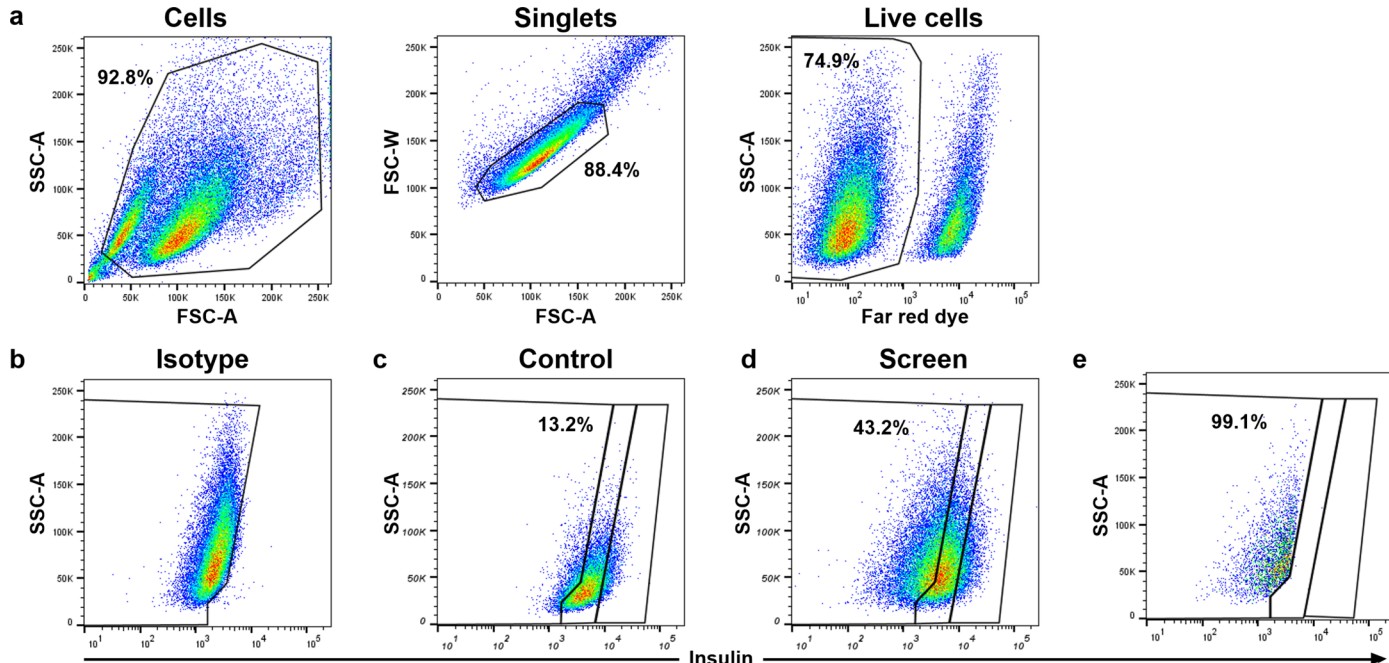

**Extended Data Fig. 2 | FACS sorting of the genome-wide CRISPR screen.** Representative FACS images and sorting gates for stained EndoC-βH1 in the genome-wide CRISPR screens. (**a**) Sorting gates assessing cell size, granularity and viability to exclude debris, define singlets and live cells. (**b**) Resorting of low insulin cell population to assess the sample purity. SSC-A, side scatter area; FSC-A, forward scatter area; FSC-W, forward scatter width.

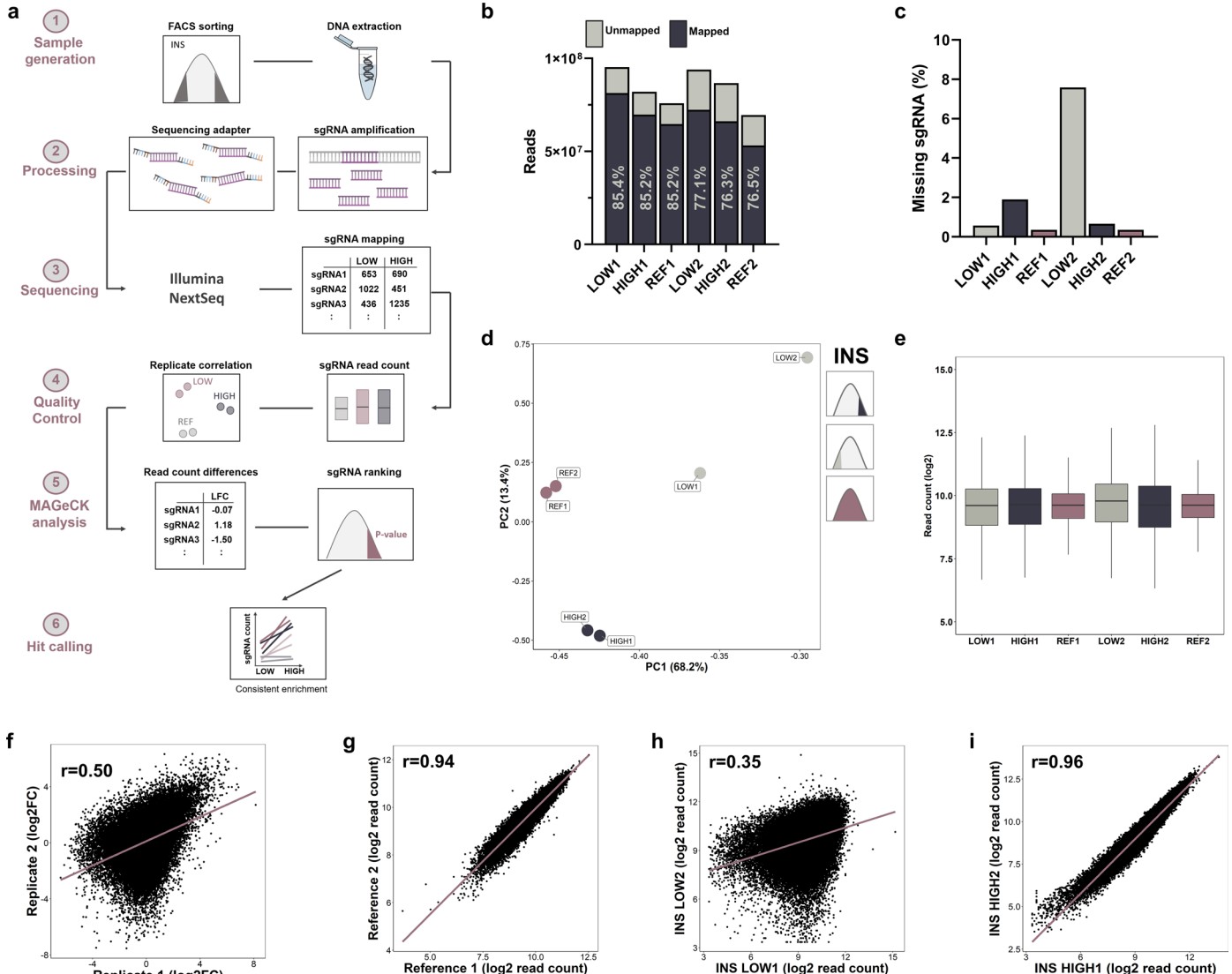

**Extended Data Fig. 3 | CRISPR screen outline and quality control. (a)** CRISPR screen outline from initial FACS sorting of transduced EndoC-βH1 to final sequencing analysis. (**b**, **c**, **e**) Read count and mapped sgRNA quality control for low insulin (LOW1, LOW2), high insulin (HIGH1, HIGH2) and reference samples (REF1, REF2) from both replicates including: (**b**) total read counts and the proportion of mapped reads and (**c**) the number of zero-count sgRNAs as proportion of total sgRNAs and (**e**) normalized read count distributions. Boxes show IQR between first and third quartiles, the horizontal line shows the median and whiskers show Q3 + 1.5 × IQR and Q1 – 1.5 × IQR. (**d**) Principal Component

Analysis on normalized sgRNA read counts for the first two components of low *(gray)* and high *(blue)* insulin content and reference samples *(purple)* from both replicates. (**f–i**) Pairwise Pearson correlation analysis of fold changes between low and high insulin content for each replicate (**f**), reference replicates (**g**), low insulin content replicates (**h**) and high insulin content replicates (**i**). The purple line indicates the linear regression line. Log2FC, log2 (fold change); r, Pearson correlation coefficient. All data are sgRNA reads from two independent genome-wide CRISPR screen replicates.

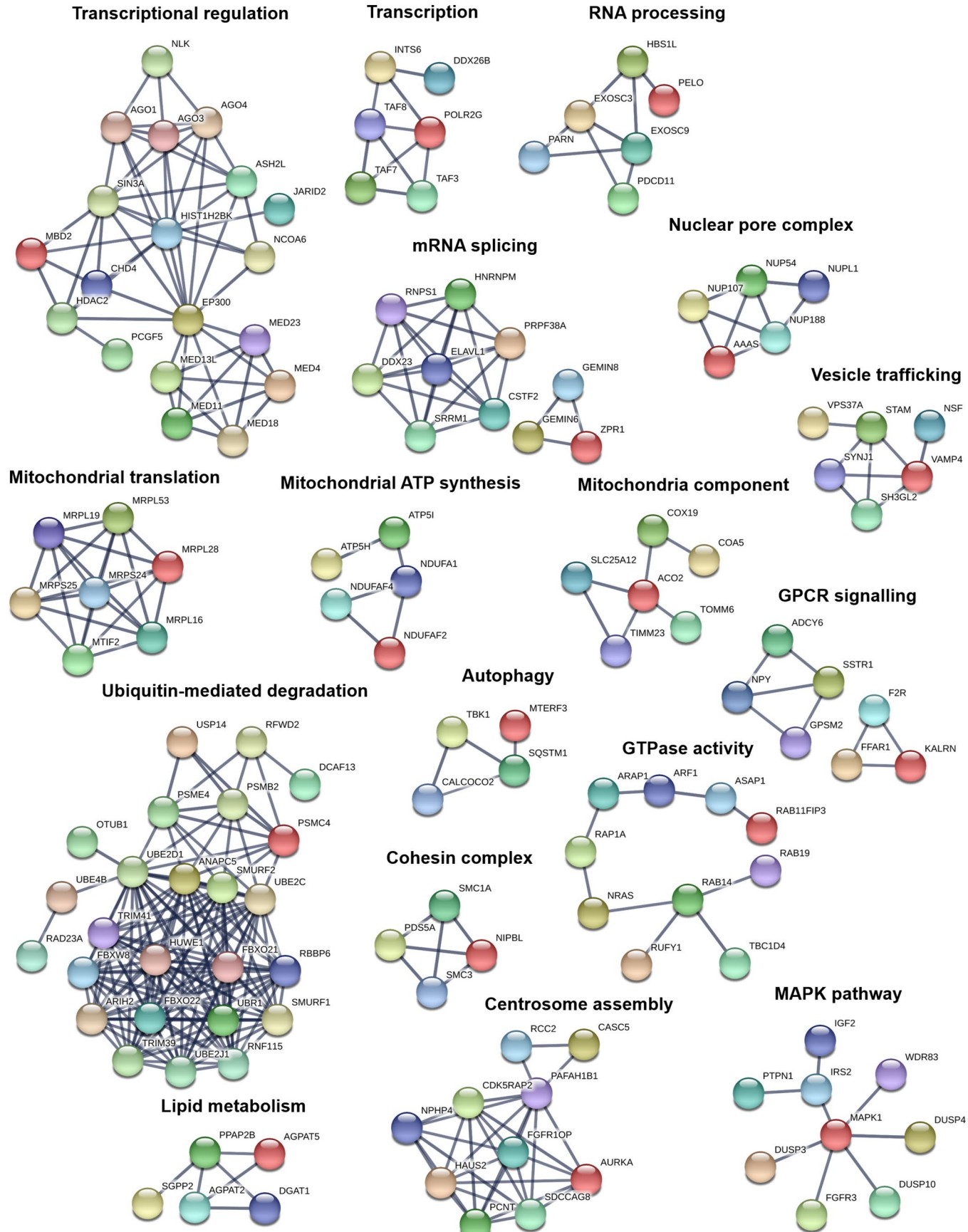

**Extended Data Fig. 4 | Protein association networks of screening hits.** STRING pathway analysis showing protein-protein associations including physical and functional interactions for CRISPR screening hits. Clusters were manually identified, annotated, and individually plotted and selected clusters are shown. Confidence level to determine interactions was set to highest confidence (0.9).

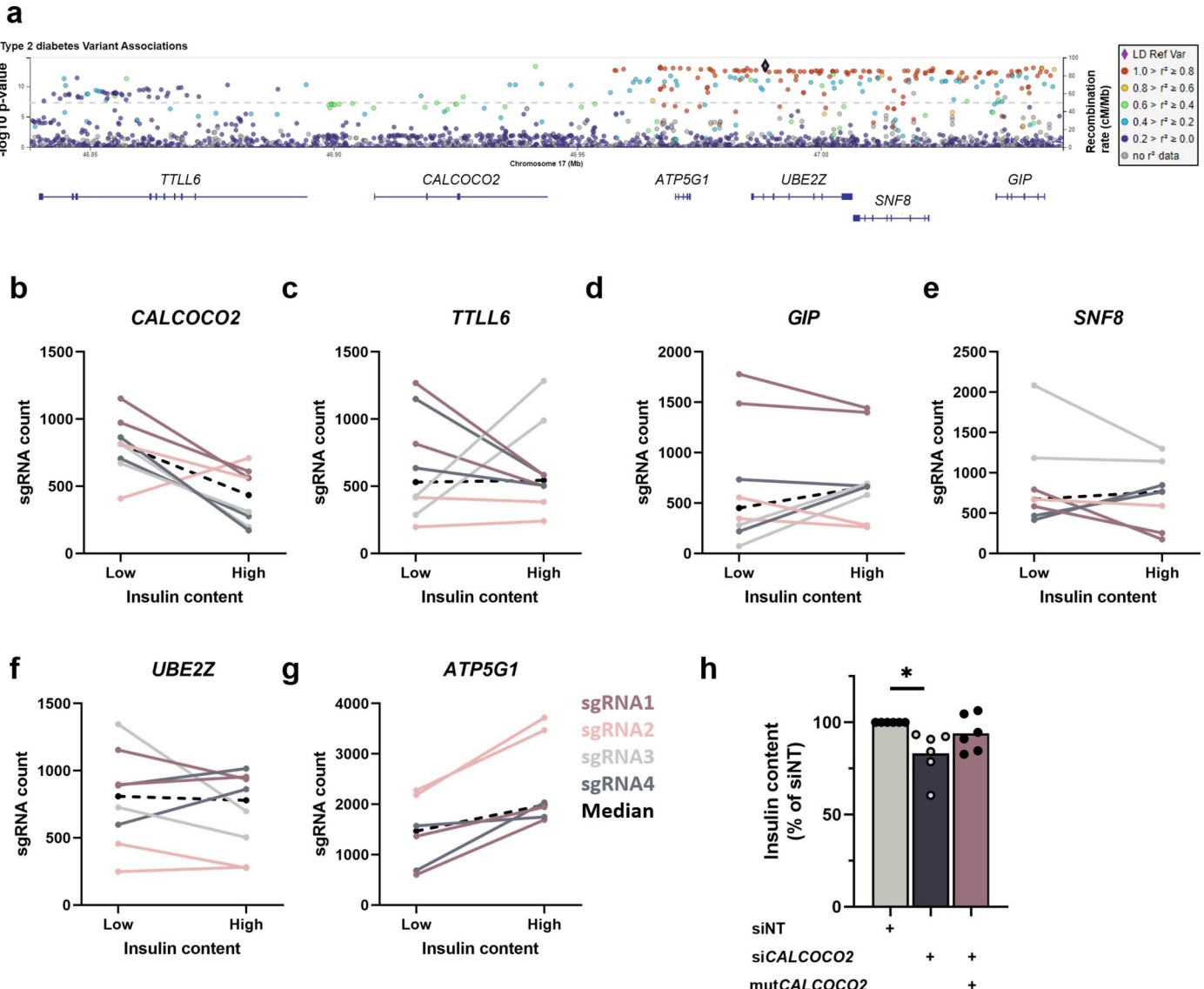

**Extended Data Fig. 5 | CALCOCO2 locus and rescue effect.** (**a**) *CALCOCO2* locus with credible set GWAS association signals *(top)* and neighboring genes *(bottom)*. (**b**–**g**) sgRNA count distribution for all 8 sgRNAs per gene (4 sgRNAs per replicate) in the genome-wide CRISPR screen for genes at the *CALCOCO2* locus. Changes in sgRNA count from low to high insulin content sample with each color representing the same sgRNA across the two screen replicates. The black dashed line represents the median sgRNA count for this gene. (**h**) EndoC-βH1 were stably transduced with either control virus and siRNA (siNT), control

virus and si*CALCOCO2* or lentivirus to express *CALCOCO2* harboring silent mutations (mut*CALCOCO2*) making it resistant to si*CALCOCO2* (*P* = 0.0352). Data were normalized to their respective control (siNT) and analyzed using a one-sample t-test. All data are mean ± SEM from six independent experiments or two independent genome-wide CRISPR screen replicates. P-value * < 0.05. No graphical depiction of significance indicates non-significant results. NT, non-targeting.

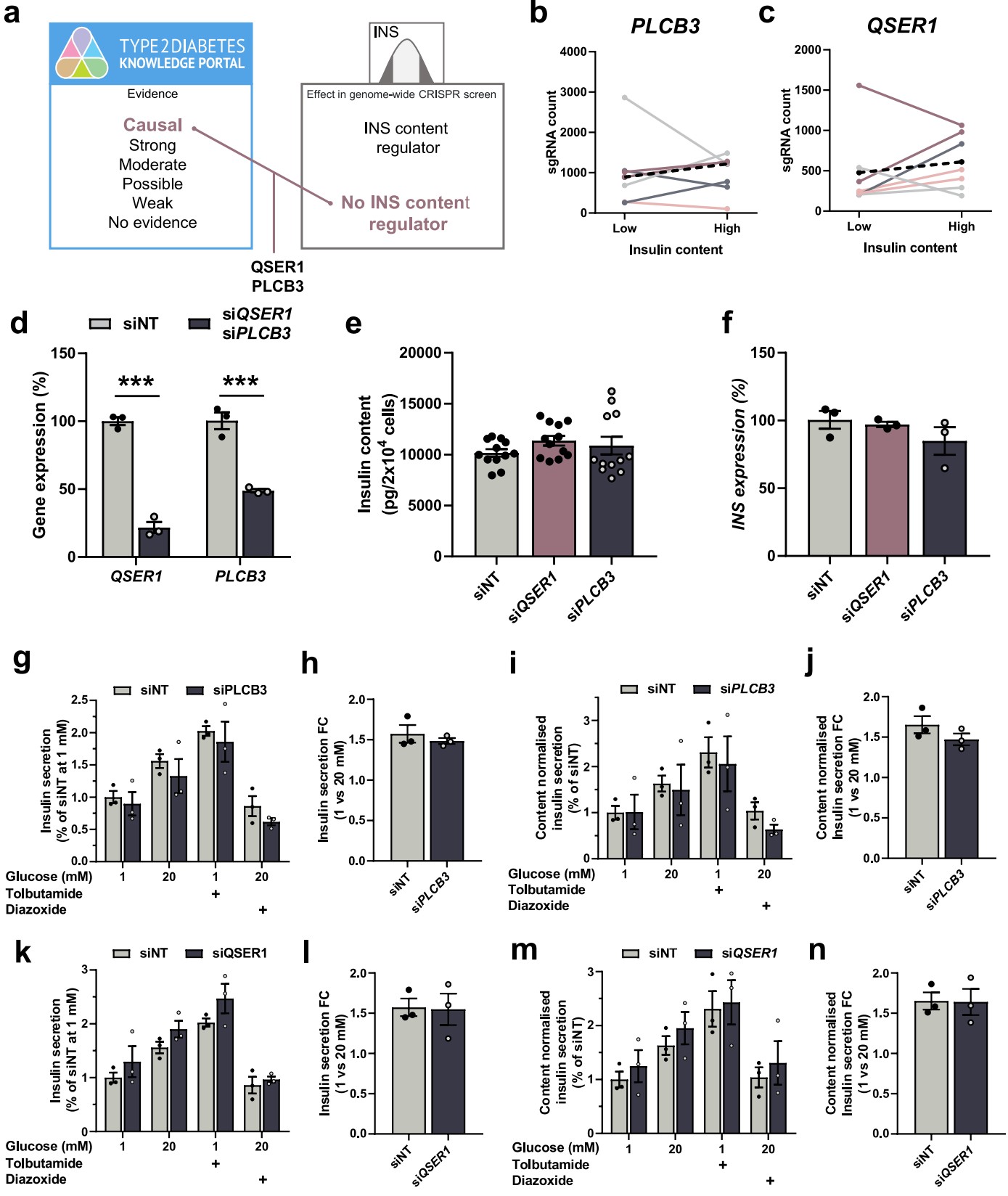

Extended Data Fig. 6 | See next page for caption.

**Extended Data Fig. 6 | *QSER1* or *PLCB3* knockdown does not affect insulin content in EndoC-βH1.** (**a**) Gene prioritization approach combining genes with causal evidence as T2D effector transcripts (as outlined in the integration approach) while also not being a screening hit, highlighted *QSER1* and *PLCB3*. *PLCB3* plays a role in the amplification pathway in the beta cell but has also been associated with insulin action in a multitrait physiological clustering approach. *QSER1* has recently been identified as a DNA methylation regulator associated with pancreatic differentiation defects. (**b**) Changes in sgRNA count from low to high insulin content sample with each color representing the same sgRNA across the two screen replicates. The black dashed line represents the median sgRNA count for this gene. (**d**–**n**) All data are from si*QSER1* or si*PLCB3* treated EndoC-βH1 compared to non-targeting control cells. (**d**) mRNA expression of *QSER1*

and PLCB3 in their respective silenced cells (blue) compared to siNT control cells (gray) ($P$ = 0.0010, 0.0005). (**e**) Insulin content in pg per 20 000 cells. (**f**) mRNA expression of *INS*. (**g**, **k**) Insulin secretion normalized to siNT or (i,m) to insulin content and siNT in 1 mM, 20 mM, 1 mM + 100 μM tolbutamide or 20 mM glucose+100 μM diazoxide. (**h**, **j**, **l**, **n**) Insulin secretion fold change from 1 to 20 mM glucose. All data are mean ± SEM from three (**d**, **f**–**n**) or 12 (**e**) independent experiments or two independent genome-wide CRISPR screens (**b**, **c**). Data were analyzed using two-sample t-test (**d**, **h**, **j**, **l**, **n**) and one-way ANOVA with Sidak's multiple comparison test (**e**, **f**, **g**, **i**, **k**, **m**). No graphical depiction of significance indicates non-significant results. P-values *** < 0.001. FC, fold change; LFC, log2 (fold change); NT, non-targeting.

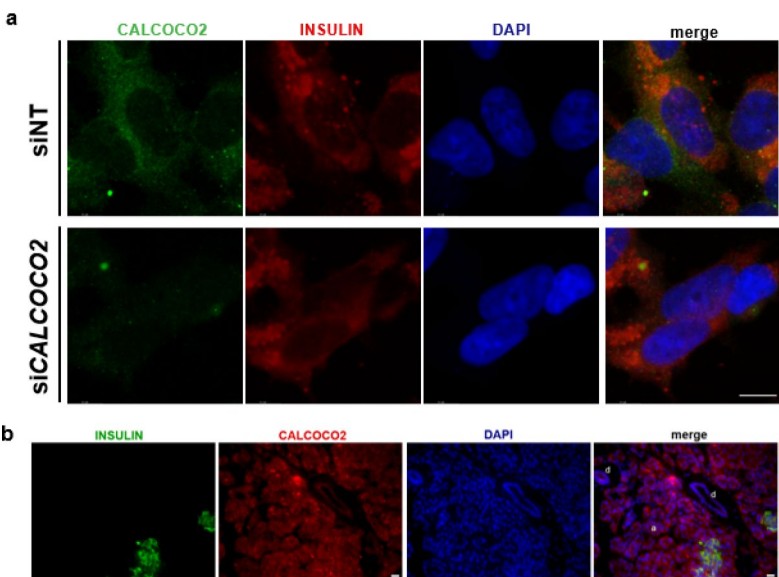

**Extended Data Fig. 7 | CALCOCO2 localization within the cell and the exocrine pancreas.** Immunofluorescence staining of EndoC-βH1 treated with si*CALCOCO2* or siNT (**a**) and exocrine tissue in pancreas sections (**b**). Sections were double immunostained for INS and CALCOCO2 (*red* or *green* as indicated). Cell nuclei were counterstained with DAPI (*blue*). Scale bar is 6 μm (**a**) or 10 μm (**b**). **d**: ductal cells; a: acinar cells; i: islet. NT, non-targeting. All images are representative from three independent immunofluorescence staining.

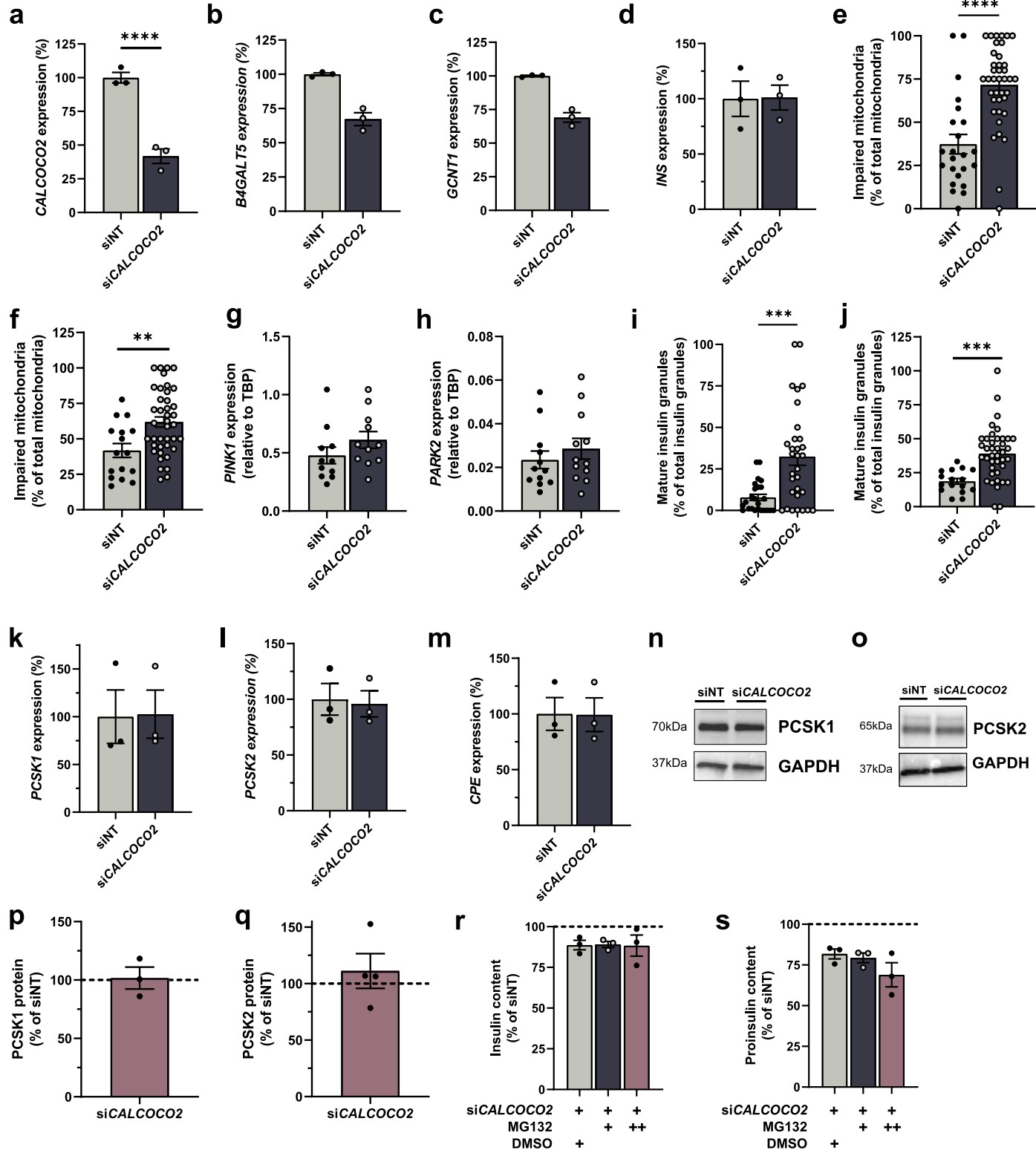

**Extended Data Fig. 8 | *CALCOCO2* RNA-Seq and EM analysis.** All data are from s*iCALCOCO2* treated EndoC-βH1 compared to non-targeting control cells. (**a–d**, **g**, **h**, **k–m**) mRNA expression of indicated target genes measured in RNA-Seq (**a–d**, **k–m**) or qPCR (**g**, **h**) with their respective silenced cells (blue) compared to siNT control cells (gray) (*P* = 1.87 × 10⁻⁶). (**e**, **f**) Quantification of altered mitochondria in EM images based on abnormal shape or cristae, normalized to total number of mitochondria by blinded assessor two and three (*P* < 0.0001, 0.0032). (**i**, **j**) Quantification of mature insulin granules normalized to total number of granules per cell in EM images by blinded assessor two and three (*P* = 0.0002, 0.0002). (**n**, **o**) Protein level of PCSK1 (**n**), PCSK2 (**o**) and its loading control GAPDH. (**p**, **q**) Quantification of PCSK1 (p) and PCSK2 (**q**) western blot protein level normalized to GAPDH and siNT control cells. r,s) Insulin (**r**) or proinsulin content (**s**) after treatment with MG132 at 1 μM (+), 10 μM (++) or DMSO compared to siNT control cells (dotted line). All data are mean ± SEM from three independent experiments or at least sixteen cells measured by three independent assessors (**e**, **f**, **i**, **j**). The protein level is displayed as percentage of siNT which is highlighted as dotted line at 100%. Data were analyzed using two-sample t-test (**e–j**), Wald test in DESeq2 adjusted using the Benjamini and Hochberg method (**a–d**, **k–m**), one-sample t-test (**p**, **q**) and one-way ANOVA with Sidak's multiple comparison test (**r**, **s**). No graphical depiction of significance indicates non-significant results. P-values ** < 0.01, *** < 0.001, **** < 0.0001. NT, non-targeting.

# Reporting Summary

Nature Research wishes to improve the reproducibility of the work that we publish. This form provides structure for consistency and transparency in reporting. For further information on Nature Research policies, see our Editorial Policies and the Editorial Policy Checklist.

## Statistics

For all statistical analyses, confirm that the following items are present in the figure legend, table legend, main text, or Methods section.

| n/a | Confirmed | |
|---|---|---|
| ☐ | ☒ | The exact sample size ($n$) for each experimental group/condition, given as a discrete number and unit of measurement |
| ☐ | ☒ | A statement on whether measurements were taken from distinct samples or whether the same sample was measured repeatedly |
| ☐ | ☒ | The statistical test(s) used AND whether they are one- or two-sided <br> *Only common tests should be described solely by name; describe more complex techniques in the Methods section.* |
| ☐ | ☒ | A description of all covariates tested |
| ☐ | ☒ | A description of any assumptions or corrections, such as tests of normality and adjustment for multiple comparisons |
| ☐ | ☒ | A full description of the statistical parameters including central tendency (e.g. means) or other basic estimates (e.g. regression coefficient) AND variation (e.g. standard deviation) or associated estimates of uncertainty (e.g. confidence intervals) |
| ☐ | ☒ | For null hypothesis testing, the test statistic (e.g. $F$, $t$, $r$) with confidence intervals, effect sizes, degrees of freedom and $P$ value noted <br> *Give P values as exact values whenever suitable.* |
| ☒ | ☐ | For Bayesian analysis, information on the choice of priors and Markov chain Monte Carlo settings |
| ☒ | ☐ | For hierarchical and complex designs, identification of the appropriate level for tests and full reporting of outcomes |
| ☐ | ☒ | Estimates of effect sizes (e.g. Cohen's $d$, Pearson's $r$), indicating how they were calculated |

*Our web collection on statistics for biologists contains articles on many of the points above.*

## Software and code

Policy information about availability of computer code

| | |
|---|---|
| Data collection | Commercial software associated with the following instruments was used to collect data: <br> - 7900HT Fast Real-Time PCR System (Applied Biosystems) <br> - EnSpire Alpha Plate Reader (Perkin Elmer) <br> - ChemiDoc MP Imaging System (Bio-Rad) <br> - Zeiss AxioM1 (Zeiss) <br> - FACSAria III (BD Biosciences) <br> - EM-1400 120kV (JEOL) <br> - STELLARIS 8 Confocal Microscope (Leica Microsystems). <br> - NEXTSeq500 (Illumina) <br> - JEM-1400 120kV (JEOL) |
| Data analysis | The following software/code was used to analyse the data: <br> - Flowjo 10.6 (BD Biosciences) <br> - MAGeCK (v0.5.9.2) algorithm (Li et al. 2014) <br> - Python 3.8 <br> - R 3.5 <br> - STRING v11 (Szklarczyk et al. 2019) <br> - STAR v2.7.9a (Liao et al. 2014) <br> - featureCounts (v2.0.1) <br> - DESeq2 (v1.26.0) (Love et al. 2014) <br> - Database for Annotation, Visualization and Integrated Discovery 6.8 (DAVID) <br> - Image Lab 6.0 software (Bio-Rad) <br> - ImageJ 1.52b <br> - Prism 8.1 (GraphPad Software) |

Respective code is available at https://doi.org/10.5281/zenodo.7226348.

For manuscripts utilizing custom algorithms or software that are central to the research but not yet described in published literature, software must be made available to editors and reviewers. We strongly encourage code deposition in a community repository (e.g. GitHub). See the Nature Research guidelines for submitting code & software for further information.

## Data

Policy information about availability of data

All manuscripts must include a data availability statement. This statement should provide the following information, where applicable:
- Accession codes, unique identifiers, or web links for publicly available datasets
- A list of figures that have associated raw data
- A description of any restrictions on data availability

Fastq sequencing files from the CRISPR screen have been deposited in the European Nucleotide Archive (ENA) at EMBL-EBI under accession number PRJEB44712. Fastq sequencing files from RNA-Sequencing experiments for siCALCOCO2 samples have been deposited in the European Genome-phenome Archive (EGA) under study number EGAS00001006127 and EndoC-βH1 expression data from a previously published study can be accessed under PRJEB15283 (ENA)78. The data is freely available to download while the processed counts can be found in the Supplementary Dataset 1. RNA-Seq data were aligned to the human genome reference GRCh38 (ftp://ftp.ensembl.org/pub/release-101/fasta/homo_sapiens/dna/Homo_sapiens.GRCh38.dna.primary_assembly.fa.gz) and counted with the gene annotation (ftp://ftp.ensembl.org/pub/release-101/gtf/homo_sapiens/Homo_sapiens.GRCh38.101.gtf.gz) downloaded from Ensembl database. Source data of unprocessed blots and Extended Data can be accessed online.

# Field-specific reporting

Please select the one below that is the best fit for your research. If you are not sure, read the appropriate sections before making your selection.

☒ Life sciences          ☐ Behavioural & social sciences          ☐ Ecological, evolutionary & environmental sciences

For a reference copy of the document with all sections, see nature.com/documents/nr-reporting-summary-flat.pdf

# Life sciences study design

All studies must disclose on these points even when the disclosure is negative.

| | |
|---|---|
| Sample size | shRNA knockdown experiments in primary islets and functional assessment in T2D risk allele carriers were carried out in the maximum number of primary human islets preparations made available through the Integrated Islet Distribution Network (IIDP) and the Alberta Diabetes Institute Islet Core at the time.<br>Two CRISPR screening replicates were performed based on previous assessment of the screening performance with varying replicate numbers which determined two replicates as ideal replicate to return ratio (Hart et al. 2017).<br>Effect sizes could not be estimated prospectively (precluding power analysis) for cellular experiments in EndoC-βH1, hence standard scientific convention of performing three independent experiments was followed (three independent passages of cells with three technical replicates each). Based on previous experience with the technical variation that can be assumed in those methods, these sample sizes were deemed sufficient to detect real biological effects. For some experiments that allowed for simultaneous sample collection across multiple independently set up experiments such as insulin content, multiples of three were performed with the analysis only being performed after all data had been collected. |
| Data exclusions | Significant outlier in primary islet assessments from T2D risk carriers were excluded based on the pre-established ROUT outlier test (Q=1%). sgRNA in the CRISPR screen with low read counts of less than 10 and sgRNAs mapping to genes that were not expressed in EndoC-βH1 were excluded from the analysis. Data was also not included if experimental mistakes occurred and were noted while performing the experiment. |
| Replication | All functional cellular experiments were reliably reproduced in at least three independent passages. Of those experiments, effects in insulin content were reproduced in two different laboratories and by several independent researchers. For the human islet data, replication on the same samples could not be attempted due to the difficulties in obtaining primary islets from cadaveric donors. |
| Randomization | The islets from human donor were grouped by genotype, so randomization was not required. For cellular experiments, samples were grouped by treatment such as control and siCALCOCO2 and randomly positioned on microplates to minimize any possibly systematic bias from technical artefacts. Other randomization was not required in this observational study. |
| Blinding | Assessors of subjective EM quantification experiments were blinded to treatments (siNT or siCALCOCO2) in data analysis. Data collection was performed with a blinded investigator present to ensure unbiased collection of images. Blinding was not required in other experiments as the outcomes were not based on subjective measurement. |

# Reporting for specific materials, systems and methods

We require information from authors about some types of materials, experimental systems and methods used in many studies. Here, indicate whether each material, system or method listed is relevant to your study. If you are not sure if a list item applies to your research, read the appropriate section before selecting a response.

## Materials & experimental systems

| n/a | Involved in the study |
|-----|----------------------|
| ☐ | ☒ Antibodies |
| ☐ | ☒ Eukaryotic cell lines |
| ☒ | ☐ Palaeontology and archaeology |
| ☒ | ☐ Animals and other organisms |
| ☐ | ☒ Human research participants |
| ☒ | ☐ Clinical data |
| ☒ | ☐ Dual use research of concern |

## Methods

| n/a | Involved in the study |
|-----|----------------------|
| ☒ | ☐ ChIP-seq |
| ☐ | ☒ Flow cytometry |
| ☒ | ☐ MRI-based neuroimaging |

# Antibodies

**Antibodies used**

Western Blot Antibodies
β-Tubulin 1 in 2000 Mouse monoclonal Santa Cruz, sc-365791 (E10)
GAPDH 1 in 10000 Rabbit polyclonal Abcam, ab37168
INS  1 in 1000 Mouse monoclonal Santa Cruz, sc-393887 (C-12)
C-peptide 1 in 1000 Mouse monoclonal ExBio, 11-247-C100 (C-PEP-01)
CALCOCO2 1 in 1000 Mouse monoclonal Santa Cruz, sc-376540 (F6)
PCSK1 1 in 1000 Rabbit polyclonal Proteintech, 28219-1-AP
PCSK2 1 in 1000 Rabbit polyclonal Proteintech, 10553-1-AP
Anti-mouse IgG HRP 1 in 2500 Rabbit polyclonal Thermo Fisher, 31460
Anti-rabbit IgG HRP 1 in 2500 Goat polyclonal Thermo Fisher, 31450

Immunofluorescence Antibodies (Pancreas sections)
CALCOCO2 1 in 100 Mouse monoclonal Santa Cruz, sc-376540 (F6)
INS  1 in 300 Guinea-pig polyclonal Dako, A0564
Anti-mouse IgG AF555 1 in 300 Donkey polyclonal Invitrogen, A-31570
Anti-guinea-pig IgG AF647 1 in 300 Goat polyclonal Abcam, ab150187

Immunofluorescence Antibodies (EndoC-βH1)
CALCOCO2 1 in 100 Rabbit polyclonal Abcam, ab68588
LC3 1 in 500 Rabbit polyclonal Novus Biologicals, NB100-2220
INS 1 in 400 Guinea-pig polyclonal Progen, 16049
Anti-mouse IgG AF555 1 in 300 Donkey polyclonal Invitrogen, A-31570
Anti-rabbit IgG AF488 1 in 300 Donkey polyclonal Invitrogen, A-21206
Anti-guinea-pig IgG AF647 1 in 300 Donkey polyclonal JacksonImmuno, 2340476

FACS antibodies
INS 1 in 10 Rabbit monoclonal Cell Signaling, #3014 (C27C9)
INS 1 in 10 Rat monoclonal  DSHB, GN-ID4
INS 1 in 10 Rat monoclonal R&D, MAB1417 (# 182410)
IgG 1 in 10 Rabbit monoclonal Cell Signaling, #3900 (DA1E)
IgG 1 in 10 Rat Invitrogen, 02-9602
Anti-rat IgG-AF488 1 in 500 Chicken polyclonal Invitrogen, A-21470
Anti-rabbit IgG-AF488 1 in 200 Goat polyclonal Invitrogen, A-11034

**Validation**

All antibodies are commercially available and were validated by the manufacturer. Specifically, the antibodies used for functional experiments targeting INS and CALCOCO2 were validated using siRNA knockdown, CRISPR KO, INS-negative cell lines and siRNA knockdown, respectively.

β-Tubulin Santa Cruz, sc-365791 (E10): Validated by manufacturer (expected size in WB).
GAPDH Abcam, ab37168: Validated by manufacturer (expected size and localization in WB, IHC-P, ICC/IF)
INS Santa Cruz, sc-393887 (C-12): Validated by manufacturer (expected size and localization in WB, ICC/IF)
C-peptide ExBio, 11-247-C100 (C-PEP-01): Validated by manufacturer (expected size and localization in ELISA, RIA, IHC(P), ICC)
CALCOCO2 Santa Cruz, sc-376540 (F6): Validated through loss of detected protein in WB at expected size upon silencing of CALCOCO2 (Fig 4d)
PCSK1 Proteintech, 28219-1-AP: Validated by manufacturer (expected size and localization in WB, IHC,ELISA)
PCSK2 Proteintech, 10553-1-AP: Validated by manufacturer (expected size in WB)
INS Dako, A0564: Validated through loss of detected protein in IF in tissue not expressing INS (exocrine) (ED Fig 7b)
CALCOCO2 Abcam, ab68588: Validated through loss of detected protein in IF upon silencing of CALCOCO2 (ED Fig 7a)
LC3 Novus Biologicals, NB100-2220: Validated by manufacturer (KO validated in WB, ICC/IF)
INS Progen, 16049: Validated by manufacturer (expected localization in ICC/IF and IHC)
INS Cell Signaling, #3014 (C27C9): Validated through loss of signal for protein in FACS upon silencing of INS and no signal in cell line not expressing INS (Fig 1b-f,ED Fig 1)
INS DSHB, GN-ID4: Validated through loss of signal for protein in FACS upon silencing of INS (ED Fig S1)
INS R&D, MAB1417 (# 182410): Validated through loss of signal for protein in FACS upon silencing of INS (ED Fig S1)
IgG Cell Signaling, #3900 (DA1E): Validated by manufacturer, Isotype control antibodies (expected size in IP)

IgG Invitrogen, 02-9602: Validated by manufacturer, Isotype control antibodies

# Eukaryotic cell lines

Policy information about <u>cell lines</u>

| | |
|---|---|
| Cell line source(s) | - The EndoC-βH1 cell line was acquired from Raphael Scharfmann (EndoCells, Paris) under an MTA.<br>- HEK293T has been purchased from sigma, ref: 12022001. |
| Authentication | The EndoC-βH1 has not been authenticated by formal short tandem repeat (STR) profiling, but the cell line is routinely subjected to rigorous testing in a number of different ways, all of which have confirmed the identity as a bona fide human beta cell line. Relevant tests include microarray-based genotyping, RNA-seq, ATAC-seq, and extensive cellular phenotyping, including insulin secretion and content (see Grotz et al. 10.12688/wellcomeopenres.15447.2.) and electrophysiology (see Hastoy et al, 10.1038/s41598-018-34743-7).<br>HEk293T has been authenticated by the manufacturer through STR profiling. |
| Mycoplasma contamination | Cells were tested for mycoplasma contamination on a quarterly basis, and tested negative on all occasions. |
| Commonly misidentified lines<br>(See <u>ICLAC</u> register) | No commonly misidentified cell lines were used in this study. |

# Human research participants

Policy information about <u>studies involving human research participants</u>

| | |
|---|---|
| Population characteristics | Characteristics of human islet donors have been provided in Supplementary Table S6. |
| Recruitment | All donors' families gave written, informed consent for the use of pancreatic tissue in research and were not financially compensated. Islets were distributed through the Integrated Islet Distribution Network (IIDP) and the Alberta Diabetes Institute Islet Core. |
| Ethics oversight | Human pancreatic islets were isolated from deceased donors under ethical approval obtained from the Human Research Ethics Board of the University of Alberta (Pro00013094, Pro00001754). |

Note that full information on the approval of the study protocol must also be provided in the manuscript.

# Flow Cytometry

## Plots

Confirm that:

☒ The axis labels state the marker and fluorochrome used (e.g. CD4-FITC).

☒ The axis scales are clearly visible. Include numbers along axes only for bottom left plot of group (a 'group' is an analysis of identical markers).

☒ All plots are contour plots with outliers or pseudocolor plots.

☒ A numerical value for number of cells or percentage (with statistics) is provided.

## Methodology

| | |
|---|---|
| Sample preparation | EndoC-βH1 cells were harvested and incubated with LIVE/DEAD Fixable Far Red Dead Cell Stain (Thermo Fisher) for 30 min at room temperature to distinguish live from dead cells and washed in 1% BSA in PBS. The cells were fixed and permeabilized using the BD Cytofix/Cytoperm kit (BD Biosciences) for 20 min at 4°C and washed using Perm/Wash Buffer (BD Biosciences). Staining with primary antibodies was performed overnight at 4°C followed by incubation with suitable secondary antibodies diluted in Perm/Wash Buffer. The samples were filtered through a 70 μm cell strainer and sorted on a FACSAria III (BD Biosciences) using a 100 μm nozzle. Isotype and transduction controls stained with each antibody alone were analyzed alongside the samples. |
| Instrument | Data was collected on a FACSAria III (BD Biosciences). |
| Software | Flow cytometry data was collected on commercial software associated with the FACSAria III (BD Biosciences) and analyzed using Flowjo 10.6 (BD Biosciences). |
| Cell population abundance | Values are described for a representatice replicate. FSC-A and SSC-A gating for cells indicated 92.8% cells, FSC-A and FSC-W gating indicated 88.4% singlets. Live/dead staining further gated for 74.9% cells. Sorted INS samples were reanalyzed on the FACSAria III with consistent parameters and demonstarted a purity of 99.1%. |
| Gating strategy | FSC-A/SSC-A gating was performed for an initial broad gating on cells, only debris with low values were excluded from the gate. FSC-A/FSC-W was applied to gate for single cells with exclusion of cells beyond the linear range. SSC-A/655nm gating was performed based on the live/dead staining and the distinct population of live cells with low 655nm values were sorted. |

Insulin staining was assessed based on SSC-A/488nm and gates were determined based on the isotype control and control cells. The gating strategy is shown in ED Fig 2.

☒ Tick this box to confirm that a figure exemplifying the gating strategy is provided in the Supplementary Information.

