## [Peer Review File · Nature Genetics]

Peer Review Information

Manuscript Title: A genome-wide CRISPR screen identifies CALCOCO2 as a regulator of beta cell function influencing type 2 diabetes risk

Corresponding author name(s): Anna L. Gloyn

Reviewer Comments & Decisions:

Decision Letter, initial version:
--

Dear Anna,

Thank you to you and your co-authors for your patience during this peer review process.

Your Article, "A genome-wide CRISPR screen identifies regulators of beta cell function involved in type 2 diabetes risk" has now been seen by 3 referees. You will see from their comments copied below that while they find your work of considerable potential interest, they have raised quite substantial concerns that must be addressed. In light of these comments, we cannot accept the manuscript for publication, but would be very interested in considering a revised version that addresses these concerns.

In brief, all three reviewers seem to appreciate the advance presented by your CRISPR screen, but differ in their opinion of the impact of this study as currently presented.

Reviewer #1 appreciates the novelty of both the screen and CALCOCO2, but thinks that one or the other of these should be greatly expanded (in terms of the analysis and mechanistic characterisation, respectively).

Reviewer #2 is the most negative; while appreciating the technical tour-de-force, they sound wholly unconvinced that the screen phenotype - intracellular insulin content - is relevant for disease.

Reviewer #3 is, on the other hand, supportive; their comments are relatively minor and mostly regard presentation, rather than requests for substantial further work.

We believe that there is sufficient support in these reviews to consider a future publication at Nature Genetics. However, we agree with Reviewer #1's comment that either the screen analysis or the mechanistic work on CALCOCO2 should be expanded (n.b. this latter point is also raised by Reviewer #2), and believe this would substantially increase the impact of your results. We find Reviewer #2's comments on the relevance of the intracellular insulin phenotype to diabetes concerning, but given the

other two referees did not echo this point, we concluded that this criticism could still be satisfactorily addressed.

We hope you will find the referees' comments useful as you decide how to proceed. If you wish to submit a substantially revised manuscript, please bear in mind that we will be reluctant to approach the referees again in the absence of major revisions.

To guide the scope of the revisions, the editors discuss the referee reports in detail within the team, including with the chief editor, with a view to identifying key priorities that should be addressed in revision and sometimes overruling referee requests that are deemed beyond the scope of the current study. We hope that you will find the prioritised set of referee points to be useful when revising your study. Please do not hesitate to get in touch if you would like to discuss these issues further.

If you choose to revise your manuscript taking into account all reviewer and editor comments, please highlight all changes in the manuscript text file. At this stage we will need you to upload a copy of the manuscript in MS Word .docx or similar editable format.

*2) If you have not done so already please begin to revise your manuscript so that it conforms to our Article format instructions, available [here](http://www.nature.com/ng/authors/article_types/index.html). Refer also to any guidelines provided in this letter.

[redacted]

If you wish to submit a suitably revised manuscript we would hope to receive it within 6 months. If you cannot send it within this time, please let us know. We will be happy to consider your revision so long as nothing similar has been accepted for publication at Nature Genetics or published elsewhere. Should your manuscript be substantially delayed without notifying us in advance and your article is eventually published, the received date would be that of the revised, not the original, version.

Thank you for the opportunity to review your work.

Sincerely,

Michael Fletcher, PhD
Associate Editor, Nature Genetics

ORCID: 0000-0003-1589-7087

Referee expertise:

Referee #1: CRISPR screens, diabetes.

Referee #2: diabetes, islet cell biology.

Referee #3: pancreatic islet cell biology, CRISPR screens.

Reviewers' Comments:

Reviewer #1:
Remarks to the Author:
Major points:

The researchers provide two areas of novel and interesting findings: 1) Combining the results of a CRISPR screen with GWAS analysis of T2D risk genes and 2) the role of CALCOCO2 in regulating insulin production. However, the researchers merely scratch the surface on both of these findings. The paper would be more informative if it focused on one of these two and either (a) discussed more hits

of the screen and how they are incorporated in network analysis or (b) investigate the mechanism by which CALCOCO2 regulates insulin, instead of merely validating the phenotype. In order to better address (b), more work should be done to determine the mechanism of action of CALCOCO2 in insulin translation, degradation, and/or secretion. The experiments proposed in the discussion are a good start.

Specific points:

Line 85 and supplemental Figure 1: Several antibodies and staining protocols were tested for human insulin staining but it is not clear which antibody and staining protocol was used in the CRISPR screen and subsequent validation study.

Line 209: No change in mRNA, a reduction in proinsulin, and no change in proinsulin:insulin with CALCOCO2 siRNA suggests CALCOCO2 affects proinsulin translation or turnover. This should be explored.

Line 220: Authors state that normalizing insulin secretion to insulin content reduced the difference between controls and siCALCOCO2 from 39.3% to 30.2%, but do not state that this normalization causes a loss of significant differences in all conditions (which is what the figures show). With this normalization, it seems that CALCOCO2 has no effect on insulin secretion.

Line 222: Should say Figure 4p,q.

Line 236: "If the lower silencing efficiency in pseudoislets is taken into account..." Not clear as to what this is referring to. Citation needed comparing silencing efficiency in primary cells vs infinite cell lines?

Line 238: Would be nice to see data on raw insulin secretion in pseudoislets before normalization to insulin content to see if that is reduced (as was done with EndoC-BH1 analysis).

Line 240: Why did shCALCOCO2 islets increase secretion with IBMX but not control islets?

Line 248: This section is a weak closing to the paper. Validating non-hits of the screen should be in supplementary information, if included at all.

Line 296: "...but also revealed a role in insulin secretion." No differences were observed when secretion was normalized to content, so this statement is a stretch.

Line 862: I believe "DHSB" (an ETC protein) is a typo and legend should say "DSHB" (antibody company).

Reviewer #2:

Remarks to the Author:

The idea for the experimental approach is remarkable and valuable. However, despite the brief mentions in the downstream validation experiments, this sophisticated screen entirely neglects the secretory branch of beta cell function. The indicated shortcomings of GSIS in cell lines are valid, but as the screen was drawn away from secretion, the relationship of the present findings with the defined

hallmarks of beta cell dysfunction in T2D, namely an impaired first phase of insulin release with perturbed pulsatile secretion and increased proinsulin/ratio remain unclear. In the end, notwithstanding the complexity and the merits of this work, I am not convinced that its outcome substantially advances the identification of causality genes for beta cell deficits in T2D.

- The abstract is misleading since the CRISPR-loss-of function screen was performed in a human pancreatic beta cell line rather than in human pancreatic beta cells.

- In the introduction there should be some information regarding the similarities, but also the differences between EndoC-bH1 cells and primary beta cells. The limitations of a screen which neglects insulin secretion as a read-out should be discussed more thoroughly.

- The introduction states that changes in insulin content are causally linked to T2D. What is the evidence for it? The cited reference is not sufficient to support this claim, as the expression of the T2D risk gene PAM is not significantly changed in most of the analyses of islets of T2D donors. In general, the claims of causality should be toned down, also considering the lack of data in islets of prediabetic subjects.

- Does the DAKO A0564 anti-insulin antibody used in the screen to measure insulin content recognize proinsulin? If so, the interpretation of the data would be more challenging.

- CALCOCO2 expression has not been reported to be changed in islets of T2D donors. Shouldn't this be the case for a causal gene?

- The specificity of the phenotypic changes observed in EndoC-bH1 cells should be corroborated by rescue experiments with siRNA-insensitive CALCOCO2 constructs.

- Validation of CALCOCO2 functional importance in an alpha cell line would be beneficial to corroborate its expression to be crucial only for beta cell function.

- The inclusion of mechanistic data addressing how CALCOCO2 positively regulates insulin content would be highly desirable. For instance, is CALCOCO2 downregulation correlated with an increased detection of insulin in autophagosomes? Are insulin granules reduced in number?

- The fact that downregulation of QSER1 and PLCB3 – two other T2D risk genes also not shown to be altered in T2D donor islets – did not change insulin content is not a sufficient proof for the power of the screen. This figure could be better placed in the supplementary material, also to avoid the manuscript to terminate with some difficult to interpret negative results. It is rather surprising instead that neither PAM nor STARD10, but also none of the other genes required for proinsulin processing, e.g., PCSK1, PCSK2 and CPE, are among the 580 genes found to alter insulin content. Conversely, this list includes ARAP1, which the cited studies (ref. 55-56) confuted being the mediator of the T2D risk allele rs140130268. Further, in human islets the T2D risk allele rs11603334 correlated with increased, not reduced, ARAP1 expression (ref. 57).

Reviewer #3:
Remarks to the Author:

Grotz et al. present here a genome-wide CRISPR screen for insulin content, using the human EndoC-bH1 beta cell line. The authors developed an assay based on insulin immunofluorescence to FACS-sort cells with elevated or diminished insulin protein after individual gene knockout, and found 580 genes influencing this phenotype. 20 of the genes overlap with candidate effector transcripts, based on previous GWAS studies and computational predictions. They subsequently evaluate gene *CALCOCO2*, which encodes an autophagy receptor, and show that insulin protein is decreased in human islets, and evaluate the effects on glucose-stimulated insulin secretion.

This work is significant and novel in being the first genome-wide screen performed in human beta cells. Only a handful of studies have performed such screens in rodent cell lines (e.g., MIN6, RIN) so this work is a strong technical advance. The quality of the data and presentation is strong.

I have a few questions about 1) the clarity surrounding the analytical choices made in the screen, and 2) the significance of the findings with *CALCOCO2*. These issues are enumerated as follows:

1. Some clarification should be made around the criteria for selecting cell populations after FACS sorting. The methods say that the 10% of highest insulin expression were chosen, but there is no similar discussion of the low end. Figure S2 shows the screen results - first, they should probably go in the main text (at least panels c-e), and second, the control gating shows that 13.2% of the cells are on the low side of expression. This value goes up to 43.2% with the sgRNA library. Why was this gating chosen?
2. Figure 1: in panels b and e, the scales should be indicated. It's hard to determine by eye that the HEK293T histogram is much lower than the EndoC histograms.
3. Figure 1: in panel g, the "y-axis" is presumably viability or similar, but it should be indicated.
4. Figure 2b: the traces for individual sgRNAs are shown, but it would be helpful to quantify this effect. Related question: were only the INS-low and INS-high populations sequenced, or was the center of the distribution sequenced as well?
5. Figure 2e: the colors are difficult to distinguish by eye to see where each gene falls in the distribution.
6. Figure 2f (and associated text): are any of these genes known to be involved in insulin secretion?
7. Figure S3d: what data was PCA performed on?
8. Page 9: last line, should be figure 4p,q (listed as fig. 5).
9. Regarding the *CALCOCO2* results, it appears that the gene knockout decreases insulin content, but does not really affect insulin secretion (when normalized for genome DNA). I think the last sentence of page 10 ("Together, our findings show that *CALCOCO2* regulates human islet beta cell function.") overinterprets the results, and should be moderated to reflect the data in these figures.
10. The imaging data in figure 5a are not all that convincing, and would be made much stronger if higher magnification images were shown, even as insets to this panel.

These points to address do not deter from the overall excitement of the approach and results. The diabetes community will welcome this study, and I think that with these edits, the manuscript can be made suitable for publication.

Author Rebuttal to Initial comments

Response to referees

*Grotz, AK et al. - A genome-wide CRISPR screen identifies *CALCOCO2* as a regulator of beta cell function involved in type 2 diabetes risk*

We thank the reviewers, and the editor, for their careful evaluation of our manuscript and their patience whilst we have attended to their requests for additional data. We were pleased to learn that the reviewers appreciated our work. Below we outline our response to their comments and detail new results we have generated to strengthen our conclusions and to provide mechanistic insight into the role of *CALCOCO2* in human beta cell function.

Reviewer 1

Major points

The researchers provide two areas of novel and interesting findings: 1) Combining the results of a CRISPR screen with GWAS analysis of T2D risk genes and 2) the role of *CALCOCO2* in regulating insulin production. However, the researchers merely scratch the surface on both of these findings. The paper would be more informative if it focused on one of these two and either (a) discussed more hits of the screen and how they are incorporated in network analysis or (b) investigate the mechanism by which *CALCOCO2* regulates insulin, instead of merely validating the phenotype. In order to better address (b), more work should be done to determine the mechanism of action of *CALCOCO2* in insulin translation, degradation, and/or secretion. The experiments proposed in the discussion are a good start.

We thank the reviewer for their evaluation of our manuscript and for highlighting our novel and interesting findings. We agree that further investigation of the role of *CALCOCO2* in beta cell function would strengthen our findings. We have now performed a series of additional experiments to deepen our understanding of how loss of *CALCOCO2* influences insulin content and secretion. These include investigating the impact of *CALCOCO2* silencing on gene expression, insulin granule phenotype, mitochondrial structure and function, the cellular location of *CALCOCO2* in human beta-cells and the impact of type 2 diabetes risk alleles at the locus on insulin content and secretion in primary human islets. Through these endeavours we are now able to demonstrate the following:

- Reduced insulin content in *CALCOCO2* silenced cells can be restored through over-expression of a silencing resistant *CALCOCO2* construct demonstrating that reduced insulin content is a direct cause of reduced *CALCOCO2* levels
- *CALCOCO2* is localized to the cytoplasm in human beta cells and does not co-localise with insulin
- *CALCOCO2* silencing alters the structure of mitochondria but does not reduce its ability to generate ATP
- Loss of *CALCOCO2* does not affect the cells' ability to degrade mitochondria (mitophagy) upon artificial induction through FCCP
- *CALCOCO2* silencing reduces the number of immature insulin granules and total proinsulin content in the beta cell

- Insulin processing and proinsulin maturation are unaffected upon *CALCOCO2* silencing
- The lysosomal inhibitor Bafilomycin A1 (BafA1, an inhibitor of autophagosome-lysosome fusion and lysosomal degradation) leads to prominent accumulation of autophagosomes in *CALCOCO2* silenced cells, supporting an autophagy-mediated reduction of immature granules upon loss of *CALCOCO2*
- Inhibition of the ubiquitin-proteasome system through the inhibitor MG132 does not rescue the insulin phenotype in *CALCOCO2* silenced cells
- Islets from carriers of T2D associated variants at the *CALCOCO2* locus have defects in insulin secretion.

Details of these experiments and our findings are now included in our revised manuscript.

Specific points

Line 85 and supplemental Figure 1: Several antibodies and staining protocols were tested for human insulin staining but it is not clear which antibody and staining protocol was used in the CRISPR screen and subsequent validation study.

We thank the reviewer for highlighting that the description of the FACS staining used in the genome-wide CRISPR screen was not sufficiently clear and we have added the following sentences in the methods and Supplementary Fig S1, respectively.

'Upon testing of several INS targeting antibodies, a rabbit monoclonal anti-INS antibody from Cell Signaling (#3014) was used in the genome-wide screen.'

'The genome-wide CRISPR screen was performed using the anti-INS antibody from Cell Signaling and the commercial fixation and permeabilisation Cytofix/Cytoperm reagents and protocol from BD Sciences.'

The validation studies using anti-INS antibodies in Western Blot, Immunofluorescence or alphaLISA experiments have been performed with previously validated antibodies which are listed in Supplementary Table S4 in the context of each experimental approach.

Line 209: No change in mRNA, a reduction in proinsulin, and no change in proinsulin:insulin with CALCOCO2 siRNA suggests CALCOCO2 affects proinsulin translation or turnover. This should be explored.

We thank the reviewer for this suggestion and agree that insulin precursor translation, processing or degradation would be potential explanations for the observed reduction in insulin content in light of unchanged mRNA expression.

Firstly, before further exploring the underlying functional mechanism, we performed additional experiments to assess proinsulin and insulin content at a higher sensitivity. Previously, we only used western blot to assess the amount of proinsulin and albeit the proinsulin:insulin ratio showed a trend towards a reduction, the variation inherent to quantifications of western blots did not allow for a high enough sensitivity to detect any differences. Upon reassessment using a highly sensitive ELISA, we demonstrated that the proinsulin:insulin ratio was indeed significantly reduced (Fig 6e,f), We further observed a reduction in immature granules, confirming this observation and indicating an CALCOCO2 mediated effect on proinsulin/immature granules.

Secondly, we performed RNA-Seq on siCALCOCO2 treated EndoC-βH1 cells compared to non-targeting controls to assess potential expression level changes that would result in impaired processing of insulin precursors. The levels of enzymes processing and modifying insulin precursors in human beta cells such as *PC1/2/3* and *CPE* however were not significantly changed, further also confirmed on the protein level (Sup Fig S8). We have therefore no indication that the processing of insulin precursors would be affected due to affected levels of the processing enzymes.

Thirdly, there is no indication that general translation or protein synthesis is affected upon loss of CALCOCO2 as other assessed proteins with a short half-life such as GAPDH are unchanged as measured through western blot with a normalization to total loaded protein¹.

Fourthly, as we have no indication of translation or processing defects and based on its previously identified role as an autophagy receptor, we hypothesized that CALCOCO2 affects insulin levels through a degradation or autophagy-based mechanism. Further evidence was provided based on accumulation of vacuolated structures upon CALCOCO2 silencing and incubation with the lysosomal inhibitor BafA1

which led to prominent accumulation of autophagosomes, supporting an autophagy-mediated reduction of immature granules upon loss of *CALCOCO2*. We further excluded ubiquitin-proteasome mediated degradation as the proteasome inhibitor MG132 did not rescue the observed insulin phenotype in *CALCOCO2* silenced cells.

Line 220: Authors state that normalizing insulin secretion to insulin content reduced the difference between controls and si*CALCOCO2* from 39.3% to 30.2%, but do not state that this normalization causes a loss of significant differences in all conditions (which is what the figures show). With this normalization, it seems that *CALCOCO2* has no effect on insulin secretion.

We completely agree with the reviewer that it was not clearly stated that the difference between si*CALCOCO2* and control cells is non-significant upon normalization. While we wanted to highlight that the normalized insulin secretion in si*CALCOCO2* does not fully reach the baseline level observed in control cells, we did not want to infer any misleading and non-significant effects so we added the following sentence:

'Even though the level of insulin secretion did not reach baseline levels upon normalization, the difference was not statistically significant and highlighted the reduced level of insulin content as the primary underlying cause of the reduction in total insulin secretion across all stimulation conditions.'

Line 222: Should say Figure 4p,q.

This has now been corrected.

Line 236: "If the lower silencing efficiency in pseudoislets is taken into account..." Not clear as to what this is referring to. Citation needed comparing silencing efficiency in primary cells vs infinite cell lines?

We thank the reviewer for highlighting that this conclusion has not been expressed clearly. Here, we simply wanted to point out that the reduction in insulin content was proportional to the achieved silencing efficiency. Insulin content was reduced in EndoC- β H1 and pseudoislets by 0.31 vs 0.29 percentage points per 1% achieved silencing efficiency, respectively. We have therefore edited the statement:

'Although the decrease in intracellular insulin in pseudoislets was lower than in EndoC-βH1, it was reduced to the same extent in both, the experimental model and primary human tissue if normalized to the achieved silencing efficiency, supporting a relationship between CALCOCO2 levels and function.'

Line 238: Would be nice to see data on raw insulin secretion in pseudoislets before normalization to insulin content to see if that is reduced (as was done with EndoC-BH1 analysis).

In light of this comment by the reviewer, we have carefully revisited the experimental approach and presentation of our insulin secretion data in pseudoislets. We agree that assessing the data before normalisation as it has been done for EndoC-βH1 allows for a much better side-by-side comparison and the opportunity to interpret the effects on insulin secretion without normalizing for the differences in insulin content.

For our experiments in EndoC-βH1, it is possible to show the data in a pre-normalised state as the results were obtained from a homogenous beta cell line suspension which has been simultaneously plated into separate wells to achieve equal cell numbers throughout. Therefore, normalisation is not necessarily required to interpret the results as they have been obtained from comparable wells.

Pseudoislets however are a more heterogenous population of islets in different sizes, cell type proportions, viability etc (Fig 5c). The cells are being hand-picked and counted for secretion experiments to select morphologically healthy looking pseudoislets and to approximately achieve the same number of pseudoislets or cells per well. Although this allows for assaying the cells within a close phenotypic range, it does not reach the same accuracy as the plating of homogenous EndoC-βH1. We therefore believe that presenting the pseudoislet data pre-normalisation would not be technically accurate and it would not be possible to make any meaningful conclusions.

Instead of presenting data before normalisation in pseudoislets, we showed gDNA normalised insulin secretion data to provide a normalisation method independent of the observed differences in insulin content.

Line 240: Why did shCALCOCO2 islets increase secretion with IBMX but not control islets?

We thank the reviewer for this interesting question to elaborate further on the secretory response in shCALCOCO2 pseudoislets compared to control islets. While we appreciate that a tendency towards higher insulin secretion can be observed in shCALCOCO2 cells treated with IBMX and high glucose, we would like to point out that this difference is not statistically significant ($p=0.6$) and therefore has no

statistical support. In addition, this difference is only observed upon normalisation to insulin content while normalisation to genomic DNA (gDNA) did not introduce that tendency. Upon careful revision of the data, we can therefore conclude that the amplification pathway which is being targeted with IBMX is not significantly affected in shCALCOCO2 treated cells as this tendency should otherwise also be apparent upon gDNA normalisation. We therefore conclude that the observed differences reflect the impact of changes in insulin content on secretion which might have been further enlarged due to the large variation observed in shCALCOCO2 cells compared to minimal variation in control cells treated with IBMX and high glucose.

Line 248: This section is a weak closing to the paper. Validating non-hits of the screen should be in supplementary information, if included at all.

We thank the reviewer for pointing out that the end of the manuscript did not focus on the main message of the paper and left room for a stronger closing paragraph. We have therefore moved figure 6 into the supplement (Supplementary Fig S6) and removed the final standalone paragraph on the non-hit investigation into the causal T2D genes *QSER1* and *PLCB3*.

We do however also believe that there is value in keeping the focussed experiments into those genes in the main text of the manuscript. Using a complementary strategy beyond the standard hit validation, based on genes that are causal in T2D but have not been identified as screening hits increases the confidence into the screening results to also use the data as a tool to exclude an effect on insulin content in previously identified causal genes based on genetic data with so far unknown tissue of action or disease mechanism.

In addition, previous investigations into *QSER1* and *PLCB3* in the context of beta cell function have been limited or not been pursued at all, so these experiments add an additional layer of functional insights to the established genetic data and will lead to a better understanding into the mechanism underlying their causality.

We therefore inserted a shorter embedded paragraph on the validation of *QSER1* and *PLCB3* as causal but non-hit genes into the section '*Loss of the T2D associated gene CALCOCO2 reduces insulin content*':

'To confirm that this screen correctly identified T2D associated genes which were likely to exert their effect on disease risk through modulating beta cell function, we examined additional genes which have been predicted to be causal for T2D but were not identified as screening hits (Supplementary Fig S7). The prioritized genes QSER1 and PLCB3 both contain coding variants consistent with a causal role in T2D but their mechanism of effect or tissue of action has not yet been resolved. Consistent with our negative screening results, siRNA knockdown of either gene did not affect intracellular insulin content or insulin

secretion. Together, this genome-wide CRISPR screen was able to correctly identify the insulin regulatory function of CALCOCO2 whilst also distinguishing between cellular phenotypes likely related to islet-cell development (QSER1) or insulin action (PLCB3).'

Line 296: "...but also revealed a role in insulin secretion." No differences were observed when secretion was normalized to content, so this statement is a stretch.

As described in the response to the reviewer comment on Line 220, we agree and have amended the sentence in regard to CALCOCO2's independent effect on insulin secretion:

'Our focused functional studies using complementary siRNA-based knockdown confirmed its effect on insulin content and an indirect, insulin-content mediated reduction on insulin secretion.'

Line 862: I believe "DHSB" (an ETC protein) is a typo and legend should say "DSHB" (antibody company).

This has now been corrected in the Supplementary Table S4 and Supplementary Fig S1.

Reviewer 2

The idea for the experimental approach is remarkable and valuable. However, despite the brief mentions in the downstream validation experiments, this sophisticated screen entirely neglects the secretory branch of beta cell function. The indicated shortcomings of GSIS in cell lines are valid, but as the screen was drawn away from secretion, the relationship of the present findings with the defined hallmarks of beta cell dysfunction in T2D, namely an impaired first phase of insulin release with perturbed pulsatile secretion and increased proinsulin/ratio remain unclear. In the end, notwithstanding the complexity and the merits of this work, I am not convinced that its outcome substantially advances the identification of causality genes for beta cell deficits in T2D.

We thank the reviewer for acknowledging that our experimental approach is both remarkable and valuable. We agree that insulin secretion is an important cellular phenotype to measure when assessing beta cell function but this does not mean insulin content is not. We believe that the field requires assessment of multiple disease relevant cellular phenotypes, which include insulin content, in multiple disease models to acquire a complete picture of the consequence of transcript perturbation. No one screen will provide all disease relevant phenotypes and efforts will need to be made across the community to build on this first screen to ensure a complete map exists to relate cellular phenotypes to gene perturbation.

- The abstract is misleading since the CRISPR-loss-of function screen was performed in a human pancreatic beta cell line rather than in human pancreatic beta cells.

We apologize that the abstract was misleading and exchanged '*human pancreatic beta cells*' into '*human pancreatic beta cell line*' to more accurately represent our experimental approach.

- In the introduction there should be some information regarding the similarities, but also the differences between EndoC-βH1 cells and primary beta cells. The limitations of a screen which neglects insulin secretion as a read-out should be discussed more thoroughly.

We thank the reviewer for this suggestion and have included the following additional information about the similarities and differences between EndoC-βH1 and primary beta cells in the introduction:

'The immortalized EndoC-βH1 cell line displays a multi omic signature similar to primary human beta cells albeit with distinct characteristics highlighting the fetal and transformed origin of the cell line^{18,19}. While insulin content is lower compared to primary islets, EndoC-βH1 cells demonstrate similar

electrophysiological and secretory properties, making it a physiologically relevant model to study beta cell function in vitro^{17,20–22}.'

In addition, we have added the following paragraph in the discussion to further acknowledge the limitations of a screen focussing only on insulin content:

'While this genome-wide CRISPR screen has identified novel genes involved in beta cell function and provides a resource for future integration and in-depth studies, it only captures one of several possible disease relevant cellular phenotypes under one condition. The screen assessed changes in insulin content under basal glucose conditions and consequently, genes specifically affecting insulin secretion without modulating insulin content or changes in intracellular content that only occur upon glucose starvation or stimulation were not captured.'

- The introduction states that changes in insulin content are causally linked to T2D. What is the evidence for it? The cited reference is not sufficient to support this claim, as the expression of the T2D risk gene PAM is not significantly changed in most of the analyses of islets of T2D donors. In general, the claims of causality should be toned down, also considering the lack of data in islets of prediabetic subjects.

We thank the reviewer for this comment and have amended our manuscript to better describe that our previous evaluation of the PAM gene, an effector transcript at a T2D GWAS locus, demonstrates that the causal mechanism involves reduced insulin content. Furthermore, we were able to demonstrate that the T2D-risk alleles resulted in reduced protein function and expression through *in vitro* studies and effects on PAM expression in serum and primary human islets². The reviewer is correct that this has not been reported in eQTL studies but this most likely reflects the modest effect size which is why our analysis focused on allelic imbalance.

*'Unlike insulin secretion, insulin content however can be quantified using FACS. Changes in insulin content have also previously been causally associated with T2D-risk, such as at the PAM locus where T2D-risk alleles cause reduced expression and/or function and reduced PAM expression is associated with reduced insulin content*²³.

To further provide independent support for the causal link between *CALCOCO2* and T2D, we have now performed additional experiments in primary human islets of prediabetic individuals carrying a T2D risk allele, which have been added as Fig. 6m-p and demonstrate a specific effect of *CALCOCO2* risk alleles on insulin secretion. Although the effector transcript status of the GWAS variant remains, the independent coding variant signal provides evidence for a direct role of *CALCOCO2*.

- Does the DAKO A0564 anti-insulin antibody used in the screen to measure insulin content recognize proinsulin? If so, the interpretation of the data would be more challenging.

We thank the reviewer for this question and would like to clarify that the DAKO antibody has only been used for immunofluorescence staining and not in the genome-wide screen (as shown in detail in Supplementary Table S4). To further clarify which antibody has been used, we have added the following sentences in the methods and Supplementary Fig S1, respectively.

'Upon testing of several INS targeting antibodies, a rabbit monoclonal anti-INS antibody from Cell Signaling (#3014) was used in the genome-wide screen.'

'The genome-wide CRISPR screen was performed using the anti-INS antibody from Cell Signaling and the commercial fixation and permeabilisation Cytofix/Cytoperm reagents and protocol from BD Sciences.'

To provide the reviewer with further information to ensure we have adequately addressed their concern, the antibodies used in the screen and subsequent validation studies target the B chain of the protein which is present in both, insulin precursor and mature insulin as indicated below. As only the absence of the C-chain/peptide differentiates mature insulin from insulin precursors, antibodies targeting mature insulin also cross-react with insulin precursor.

Capturing both precursor and mature insulin however does not invalidate the interpretation of the screening hits. Although it is not possible to specifically pinpoint initially if mature, immature or all forms of insulin are changed, the interpretation of beta cell dysfunction and insulin content dysregulation within the cell upon gene knockout remains. Follow-up studies for individual genes, such as done here for *CALCOCO2* showing a specific effect on proinsulin can then be used to identify if the changes in total insulin content occur in all insulin forms or only apply to specific mature or immature forms.

- CALCOCO2 expression has not been reported to be changed in islets of T2D donors. Shouldn't this be the case for a causal gene?

We thank the reviewer for raising this point of confusion. There are two reasons why a difference in *CALCOCO2* expression may not have been reported in human islets from carriers of T2D-risk variants at this locus. First, there is an independent coding variant association at this locus and the coding variant is predicted to alter function of the protein rather than gene expression. Second, for regulatory variants power issues are well reported for detecting effects on gene expression. The absence of co-localisation of a cis-eQTL in islets with the T2D-GWAS signal at this locus is not unusual. Note that it was only in the recent combined analysis from the INSPIRE consortium that an eQTL at the *TCF7L2* locus was detected. The effect size at this locus (OR 1.7) is much larger than the effect size at the *CALCOCO2* locus.

- The specificity of the phenotypic changes observed in EndoC-βH1 cells should be corroborated by rescue experiments with siRNA-insensitive *CALCOCO2* constructs.

We thank the reviewer for the suggestion to further verify our findings that the reduction in insulin content is attributable to reduced *CALCOCO2* expression. We now demonstrate that we can rescue the effect on insulin content by overexpression of a siRNA-insensitive *CALCOCO2* construct. We have added the additional results as Supplementary Fig S5h.

Due to the very low transfection efficiency of EndoC-βH1, we transduced the cells with control virus or lentivirus inducing expression of *CALCOCO2* harboring silent mutations (*mutCALCOCO2*) to prevent siRNA silencing. Silencing with si*CALCOCO2* in control cells induced the expected significant reduction in insulin content while silencing in *mutCALCOCO2* expressing cells did not induce a significant change of insulin content. We have added the following paragraph in the manuscript: ‘

‘This effect could be rescued to baseline insulin levels upon overexpression of CALCOCO2 harboring silent mutations to prevent siRNA silencing (Supplementary Fig S5).’

- Validation of *CALCOCO2* functional importance in an alpha cell line would be beneficial to corroborate its expression to be crucial only for beta cell function.

We thank the reviewer for highlighting that it is crucial to also independently validate *CALCOCO2*s function in alpha cells as the complementary pancreatic secretory cell type involved in regulating blood glucose levels. However, no human alpha cell line has so far successfully been generated or cultured so it was not possible to perform those validation studies. We did not want to invalidate our results

generated in a human cell line through inappropriate validation in a rodent cell line having different species-specific cell characteristics.

As part of our *CALCOCO2* validation pipeline however, we also assessed the effects of *CALCOCO2* in primary human pseudoislets which allows the study of alpha cell function as shown in Figure 5h-j. We did not detect any differences between sh*CALCOCO2* and control cells in glucagon content or secretion, indicating no effect on alpha cell function in this experimental approach.

- The inclusion of mechanistic data addressing how *CALCOCO2* positively regulates insulin content would be highly desirable. For instance, is *CALCOCO2* downregulation correlated with an increased detection of insulin in autophagosomes? Are insulin granules reduced in number?

We thank the reviewer for this suggestion and agree that those follow-up experiments would be insightful to understand *CALCOCO2*'s functional effects on insulin content.

Using EM, we demonstrate that immature granules are reduced upon loss of *CALCOCO2* while the number of mature granules is unchanged, in line with our reduction in total proinsulin content. We observed no changes in insulin processing based on expression levels (RNA and protein) of processing enzymes. An accumulation of vacuolated structures (likely autophagosomes) seen in EM and a prominent accumulation of autophagosomes in *CALCOCO2* silenced cells based on LC3-staining upon incubation with the lysosomal inhibitor Bafilomycin A1 (BafA1, an inhibitor of autophagosome-lysosome fusion and lysosomal degradation) support an autophagy-mediated reduction of immature granules upon loss of *CALCOCO2*.

- The fact that downregulation of *QSER1* and *PLCB3* – two other T2D risk genes also not shown to be altered in T2D donor islets – did not change insulin content is not a sufficient proof for the power of the screen. This figure could be better placed in the supplementary material, also to avoid the manuscript to terminate with some difficult to interpret negative results. It is rather surprising instead that neither *PAM* nor *STARD10*, but also none of the other genes required for proinsulin processing, e.g., *PCSK1*, *PCSK2* and *CPE*, are among the 580 genes found to alter insulin content. Conversely, this list includes *ARAP1*, which the cited studies (ref. 55-56) confuted being the mediator of the T2D risk allele rs140130268. Further, in human islets the T2D risk allele rs11603334 correlated with increased, not reduced, *ARAP1* expression (ref. 57).

We thank the reviewer for their comments. We have moved the figure to the supplementary information as requested. We further agree that it is interesting that we did not detect an effect of *PAM*

knockout on insulin content. In a separate study using gRNAs from a genome-wide library to generate *PAM* knockout cells, we detected residual gene expression and a protein isoform which had not been targeted with the sgRNAs in the library. It is therefore possible that some genes, including *PAM*, are not detected due to incomplete gene knockout. This phenomenon has actually been reported by others and is why groups are now exploring complementary or additional screens with CRISPRi libraries.

We agree with the reviewer that our finding of *ARAP1* as a gene influencing insulin content adds to the complexity at the *ARAP1/STARD10* locus. We have indeed previously reported co-localisation of cis eQTL at *STARD10* and a T2D-GWAS variant at this locus^{3,4} and worked with others to examine the impact of gene perturbation on islet function in mice⁵. Our observation of an effect of *ARAP1* knockout on insulin secretion does not invalidate these other data. There is increasing evidence that GWAS variants could influence multiple transcripts. Regarding the direction of effect, the earlier small study by Kulzer and colleagues in 2014⁶ which only investigated expression in 3 to 6 islet samples could be misleading. Reporting our findings in the context of other literature provides the field with the opportunity to evaluate all findings.

Reviewer 3

Grotz et al. present here a genome-wide CRISPR screen for insulin content, using the human EndoC-bH1 beta cell line. The authors developed an assay based on insulin immunofluorescence to FACS-sort cells with elevated or diminished insulin protein after individual gene knockout, and found 580 genes influencing this phenotype. 20 of the genes overlap with candidate effector transcripts, based on previous GWAS studies and computational predictions. They subsequently evaluate gene *CALCOCO2*, which encodes an autophagy receptor, and show that insulin protein is decreased in human islets, and evaluate the effects on glucose-stimulated insulin secretion.

This work is significant and novel in being the first genome-wide screen performed in human beta cells. Only a handful of studies have performed such screens in rodent cell lines (e.g., MIN6, RIN) so this work is a strong technical advance. The quality of the data and presentation is strong.

I have a few questions about 1) the clarity surrounding the analytical choices made in the screen, and 2) the significance of the findings with *CALCOCO2*. These issues are enumerated as follows:

We thank the reviewer for their evaluation of our manuscript and their suggestions for clarification.

1. Some clarification should be made around the criteria for selecting cell populations after FACS sorting. The methods say that the 10% of highest insulin expression were chosen, but there is no similar discussion of the low end. Figure S2 shows the screen results - first, they should probably go in the main text (at least panels c-e), and second, the control gating shows that 13.2% of the cells are on the low side of expression. This value goes up to 43.2% with the sgRNA library. Why was this gating chosen?

We thank the reviewer for the suggestion to move the flow cytometry results of the screen into the main text of the manuscript and have added them as panel a of Figure 2 beside the histogram overlay.

As we were not able to predict the effects on insulin content or determine the gating strategy of the CRISPR screen prior to performing it, we initially decided against a fixed gating strategy such as the bottom and top 10 % of the cells irrespective of their distribution. Instead, we wanted to first assess the distribution of the cells and base our decision on the spread of the screening cells. The tail of the cell population towards higher insulin in the screening cells compared to control cells was not as wide as towards lower insulin and there was no predetermined control representing a maximum level of insulin

which we could have used as a high insulin gate (such as an isotype control for low insulin). We therefore decided to sort the top 10 % of cells with the highest insulin level as INS^{HIGH} .

The effect towards lower insulin however was much stronger as visible by the wide tail of the cells towards lower insulin and exceeded 10 % of the cells. To avoid being too restrictive in the gating and potentially lose real hits, we decided not to apply a gating based on only the bottom 10 %. Instead, we used the minimal gate defining the insulin positive population, the isotype control. As well-known from previous studies and our own experience, EndoC- β H1 are a heterogeneous population so it is not surprising that 13.2% of control cells also fall into the category of low insulin⁷⁻⁹. Instead of potentially losing real hits through setting a more restrictive FACS gate based on the control cells, we were confident that a combination of gating on the isotype control together with stringent downstream analysis criteria excluding genes without consistent sgRNA effects would result in the most extensive but at the same time specific hit list.

To further clarify this gating strategy, we have added the following paragraph in the methods:

'Samples were gated based on live and single cells prior to INS level assessment. Gates for cells with reduced INS level (INS^{LOW}) were set based on the isotype control and the highest 10 % of INS-expressing cells were sorted as INS^{HIGH} .'

2. Figure 1: in panels b and e, the scales should be indicated. It's hard to determine by eye that the HEK293T histogram is much lower than the EndoC histograms.

We thank the reviewer for highlighting that the scales have not been indicated in our flow cytometry figures and have corrected this throughout the manuscript. Additionally, to be able to better differentiate the individual samples within experiments such as in panel g, we have also added the individual FACS pseudocolor plots besides the histogram overlays. The additional individual plots also clarify specific gating strategies such as in panel i based on the isotype control.

3. Figure 1: in panel g, the "y-axis" is presumably viability or similar, but it should be indicated.

We apologize that we have not labeled all flow cytometry axis clearly and have corrected this throughout the manuscript in all figures with both the protein of interest (e.g. INS) and its according fluorophore (e.g. AF488) or accordingly marker-independent values such as count. This specific y-axis label was indicated as SSC-A, side scatter area, a measure of cell complexity as a secondary variable to plot the results of INS-AF488 against and obtain two-parameter scatter plots.

4. Figure 2b: the traces for individual sgRNAs are shown, but it would be helpful to quantify this effect. Related question: were only the INS-low and INS-high populations sequenced, or was the center of the distribution sequenced as well?

We thank the reviewer for pointing out that adding a quantification to the individual sgRNA traces would aid in assessing the effect and improve the comparability. Therefore, we extracted the log₂ fold change (FC) values for the depicted individual sgRNAs (*INS*, *NKX2.2* and *CALCOCO2*) from the MAGeCK analysis results and included this information in the detailed respective figure legends:

Figure 2c: 'Changes in sgRNA count from low to high insulin content screening sample with each colour representing the same sgRNA across the two screen replicates for INS and NKX2-2 (Log₂(FC) 1.06 and 1.21, respectively).'

Figure 4b: 'Changes in sgRNA count from low to high insulin content sample with each color representing the same sgRNA across the two screen replicates. The black dashed line represents the median sgRNA count for this gene (Log₂(FC): 1.08).'

To be able to pick up the strongest sgRNA enrichment or depletion, we have only sequenced and compared INS-low and INS-high populations.

5. Figure 2e: the colors are difficult to distinguish by eye to see where each gene falls in the distribution.

We completely agree with the reviewer that the colors chosen in Figure 2e were not easily distinguishable and have changed this in the updated figure.

6. Figure 2f (and associated text): are any of these genes known to be involved in insulin secretion?

We thank the reviewer for the question and would like to add that we have highlighted several screening hits in Figure 2f that have previously been identified to play a role in monogenic diabetes, T2D, beta cell function or in the regulation of insulin. Specifically involved in regulating insulin secretion however are the following genes:

While the beta cell transcription factor *NKX2.2* is essential in the development of beta cells, it has also shown to play a crucial role in mature beta cells where loss of *Nkx2.2* in islets of transgenic adult mice led to reduced insulin expression, insulin content and impaired insulin secretion^{10,11}.

Homozygous LoF mutations in *SLC2A2*, which encodes the glucose transporter GLUT2 have been identified as a rare cause of neonatal diabetes likely mediated through impaired insulin secretion¹².

NIPBL has been shown to regulate *Ins2* expression and insulin secretion in follow-up studies of a mouse MIN6 CRISPR insulin content screen¹³.

RFWD2/COP1 has been associated with insulin granule docking defects and diabetes in beta cell specific KO mice and *HUWE1* has been shown to impair insulin exocytosis in pancreas-specific KO mice^{14,15}.

The GPCR receptor *FFAR1 (GPR40)* is well established to enhance insulin secretion upon activation and *TIAM1* has been shown to potentiate insulin secretion upon KD in a rat beta cell line^{16,17}.

Another large cluster among the screening hits was the MAPK pathway centered around MAPK1/ERK2 which is involved in a wide range of cellular processes including proliferation and transcriptional regulation. *MAPK1* itself has been shown to be regulated by glucose and KD in rodent beta cells has been linked to reduced insulin secretion¹⁸. The MAPK pathway plays an essential downstream role in the insulin signaling pathway and the associated protein cluster (Fig 3c) contains crucial components of the insulin/IGF1 signal transduction network such as *IGF2*, *IRS2* and *PTPN1*. Both, overexpression and KO of *IGF2* have been associated with reduced beta cell function and demonstrated inhibited glucose stimulated insulin secretion in beta cell specific overexpression and female KO mice, respectively^{19,20}. In addition, *IGF2* has also been identified as a regulator of insulin secretion in EndoC- β H1 in a previous arrayed siRNA screen²¹. KO of the insulin receptor substrate *Irs2* in pancreatic KO mice demonstrated impaired insulin secretion while upregulation of *Irs2* on the other hand increased insulin content and secretion in mice with beta cell specific overexpression^{22,23}. Further, *PTPN1* acts as a negative regulator of insulin receptor signaling and has been linked to T2D risk and increased insulin secretion in KO mice^{24,25}.

7. Figure S3d: what data was PCA performed on?

The Principal Component Analysis (PCA) of the samples from each screen replicate was performed on median-normalized sgRNA read counts. This normalization as part of the MAGeCK algorithm allows to compare read count values between independent samples and replicates. We have further clarified this data in the according figure legend in S3d:

'Principal Component Analysis on normalized sgRNA read counts for the first two components of low (grey) and high (blue) insulin content and reference samples (purple) from both replicates.'

8. Page 9: last line, should be figure 4p,q (listed as fig. 5).

We thank the reviewer for careful reading of the manuscript. This has now been corrected to Fig. 4 p,q.

9. Regarding the CALCOCO2 results, it appears that the gene knockout decreases insulin content, but does not really affect insulin secretion (when normalized for genome DNA). I think the last sentence of page 10 ("Together, our findings show that CALCOCO2 regulates human islet beta cell function.") over interprets the results, and should be moderated to reflect the data in these figures.

We thank the reviewer for pointing out that we interpreted the observed effects of *CALCOCO2* on insulin content as general beta cell function although we did not observe an insulin content independent effect on insulin secretion. To phrase our conclusions more moderately, we made the following changes:

First, we adapted the conclusion earlier in the same results paragraph and while we wanted to highlight that the normalized insulin secretion in si*CALCOCO2* does not fully reach the baseline level observed in control cells, we did not want to infer any misleading and non-significant effects so we added the following sentence:

'Even though the level of insulin secretion did not reach baseline levels upon normalization, the difference was not statistically significant and highlighted the reduced level of insulin content as the primary underlying cause of the reduction in total insulin secretion across all stimulation conditions.'

Second, we took onboard the reviewer's suggestion and modified the last sentence of the paragraph as well to more accurately reflect its effect only on insulin content rather than directly on insulin secretion:

'Together, our findings show that CALCOCO2 plays an important role in regulating insulin content in human islet beta cells.'

10. The imaging data in figure 5a are not all that convincing, and would be made much stronger if higher magnification images were shown, even as insets to this panel.

We agree with the reviewer that the immunofluorescence staining of *CALCOCO2* in primary human pancreas sections would provide more insights about the intracellular localisation of *CALCOCO2* at a higher magnification.

In Fig 5a, we initially only wanted to demonstrate that *CALCOCO2* was expressed in primary human islets which can be unambiguously identified from the ubiquitous expression across the tissue.

To provide additional information about the intracellular localisation and higher confidence into the specificity of the staining, we performed additional stainings in EndoC- β H1, which have been added as Supplementary Figure S7. The absence of a signal for CALCOCO2 in siCALCOCO2 cells demonstrated the specificity of the antibody and staining. The CALCOCO2 staining exhibited a cytoplasmic localisation of the protein, consistent with the experiment in pancreas sections. Co-staining with INS did not indicate colocalization, highlighting that CALCOCO2 does not localize to secretory granules.

These points to address do not deter from the overall excitement of the approach and results. The diabetes community will welcome this study, and I think that with these edits, the manuscript can be made suitable for publication.

1. Mansur, N. R., Meyer-Siegler, K., Wurzer, J. C. & Sirover, M. A. Cell cycle regulation of the glyceraldehyde-3-phosphate dehydrogenase/uracil DNA glycosylase gene in normal human cells. *Nucleic Acids Res.* **21**, 993–998 (1993).
2. Thomsen, S. K. *et al.* Type 2 diabetes risk alleles in PAM impact insulin release from human pancreatic β -cells. *Nat. Genet.* **50**, 1122–1131 (2018).
3. van de Bunt, M. *et al.* Transcript Expression Data from Human Islets Links Regulatory Signals from Genome- Wide Association Studies for Type 2 Diabetes and Glycemic Traits to Their Downstream Effectors. *PLoS Genet.* **11**, 1–21 (2015).
4. Viñuela, A. *et al.* Genetic variant effects on gene expression in human pancreatic islets and their implications for T2D. *Nat. Commun.* **11**, 4912 (2020).
5. Carrat, G. R. *et al.* Decreased STARD10 Expression Is Associated with Defective Insulin Secretion in Humans and Mice. *Am J Hum Genet* **100**, 238–256 (2017).
6. Kulzer, J. R. *et al.* A Common Functional Regulatory Variant at a Type 2 Diabetes Locus Upregulates ARAP1 Expression in the Pancreatic Beta Cell. *Am J Hum Genet* **94**, 186–197 (2014).
7. Weir, G. C. & Bonner-Weir, S. Finally! A human pancreatic β cell line. *J. Clin. Invest.* **121**, 3395–3397 (2011).
8. Hastoy, B. *et al.* Electrophysiological properties of human beta-cell lines EndoC- β H1 and - β H2 conform with human beta-cells. *Sci. Rep.* **8**, 1–16 (2018).
9. Balboa, D., Prasad, R. B., Groop, L. & Otonkoski, T. Genome editing of human pancreatic beta cell

- models: problems, possibilities and outlook. *Diabetologia* **62**, 1329–1336 (2019).
10. Sussel, L. *et al.* Mice lacking the homeodomain transcription factor Nkx2.2 have diabetes due to arrested differentiation of pancreatic beta cells. *Development* **125**, 2213–2221 (1998).
 11. Doyle, M. J. & Sussel, L. Nkx2.2 regulates beta-cell function in the mature islet. *Diabetes* **56**, 1999–2007 (2007).
 12. Sansbury, F. H. *et al.* SLC2A2 mutations can cause neonatal diabetes, suggesting GLUT2 may have a role in human insulin secretion. *Diabetologia* **55**, 2381–2385 (2012).
 13. Fang, Z. *et al.* Single-Cell Heterogeneity Analysis and CRISPR Screen Identify Key β -Cell-Specific Disease Genes. *Cell Rep.* **26**, 3132–3144.e7 (2019).
 14. Suriben, R. *et al.* β -Cell Insulin Secretion Requires the Ubiquitin Ligase COP1. *Cell* **163**, 1457–1467 (2015).
 15. Wang, L. *et al.* Dichotomous role of pancreatic HUWE1/MULE/ARF-BP1 in modulating beta cell apoptosis in mice under physiological and genotoxic conditions. *Diabetologia* **57**, 1889–1898 (2014).
 16. Tsuda, N. *et al.* A novel free fatty acid receptor 1 (GPR40/FFAR1) agonist, MR1704, enhances glucose-dependent insulin secretion and improves glucose homeostasis in rats. *Pharmacol. Res. Perspect.* **5**, e00340 (2017).
 17. Veluthakal, R., Madathilparambil, S. V., McDonald, P., Olson, L. K. & Kowluru, A. Regulatory roles for Tiam1, a guanine nucleotide exchange factor for Rac1, in glucose-stimulated insulin secretion in pancreatic beta-cells. *Biochem. Pharmacol.* **77**, 101–113 (2009).
 18. Longuet, C. *et al.* Extracellularly Regulated Kinases 1/2 (p44/42 Mitogen-Activated Protein Kinases) Phosphorylate Synapsin I and Regulate Insulin Secretion in the MIN6 β -Cell Line and Islets of Langerhans. *Endocrinology* **146**, 643–654 (2005).
 19. Casellas, A. *et al.* Insulin-like growth factor 2 overexpression induces β -Cell dysfunction and increases beta-cell susceptibility to damage. *J. Biol. Chem.* **290**, 16772–16785 (2015).
 20. Modi, H. *et al.* Autocrine Action of IGF2 Regulates Adult β -Cell Mass and Function. *Diabetes* **64**, 4148–4157 (2015).
 21. Thomsen, S. K. *et al.* Systematic Functional Characterization of Candidate Causal Genes for Type 2 Diabetes Risk Variants. *Diabetes* **65**, 3805–11 (2016).
 22. Hennige, A. M. *et al.* Upregulation of insulin receptor substrate-2 in pancreatic β cells prevents

- diabetes. *J. Clin. Invest.* **112**, 1521–1532 (2003).
23. Cantley, J. *et al.* Pancreatic deletion of insulin receptor substrate 2 reduces beta and alpha cell mass and impairs glucose homeostasis in mice. *Diabetologia* **50**, 1248–1256 (2007).
 24. Bento, J. L. *et al.* Association of protein tyrosine phosphatase 1B gene polymorphisms with type 2 diabetes. *Diabetes* **53**, 3007–3012 (2004).
 25. Fernandez-Ruiz, R., Vieira, E., Garcia-Roves, P. M. & Gomis, R. Protein Tyrosine Phosphatase-1B Modulates Pancreatic β -cell Mass. *PLoS One* **9**, e90344 (2014).

Decision Letter, first revision:

Dear Anna,

Your Article, "A genome-wide CRISPR screen identifies CALCOCO2 as a regulator of beta cell function involved in type 2 diabetes risk" has now been seen by 3 referees. You will see from their comments below that while they find your work of interest, some important points are raised. We are interested in the possibility of publishing your study in Nature Genetics, but would like to consider your response to these concerns in the form of a revised manuscript before we make a final decision on publication.

In brief, Reviewers #1 and #3 - previously positive - are now satisfied and supportive of publication.

Reviewer #2, however, remains negative. They offer a range of comments on the revision and added data, but their fundamental point seems to remain the same as in the first round of review: they do not agree that there is a clear, causal link between the screen readout (insulin content) and T2D. Their new requests are, overall, minor, rather than for substantial additional work.

We note that the other two referees did not have the same objection, but it is unclear to us what can be done to persuade the reviewer of this "causal" link. We think the specific comments on the new results should be fully addressed if possible. It may be the case that toning down the claim for this direct insulin content to T2D causal link will be sufficient to persuade the referee to be more supportive, and we would recommend considering this option.

To guide the scope of the revisions, the editors discuss the referee reports in detail within the team, including with the chief editor, with a view to identifying key priorities that should be addressed in revision and sometimes overruling referee requests that are deemed beyond the scope of the current study. We hope that you will find the prioritized set of referee points to be useful when revising your study. Please do not hesitate to get in touch if you would like to discuss these issues further.

We therefore invite you to revise your manuscript taking into account all reviewer and editor comments. Please highlight all changes in the manuscript text file. At this stage we will need you to upload a copy of the manuscript in MS Word .docx or similar editable format.

We are committed to providing a fair and constructive peer-review process. Do not hesitate to contact

us if there are specific requests from the reviewers that you believe are technically impossible or unlikely to yield a meaningful outcome.

*2) If you have not done so already please begin to revise your manuscript so that it conforms to our Article format instructions, available [here](http://www.nature.com/ng/authors/article_types/index.html). Refer also to any guidelines provided in this letter.

[redacted]

We hope to receive your revised manuscript within four to eight weeks. If you cannot send it within this time, please let us know.

Sincerely,

Michael Fletcher, PhD
Senior Editor, Nature Genetics

ORCID: 0000-0003-1589-7087

Reviewers' Comments:

Reviewer #1:

Remarks to the Author:

The authors made significant improving the manuscript, especially about the mechanism of action of CALCOCO2. I think their revision to my comments is satisfying, therefore I have no further comments and recommend the acceptance of this manuscript. Thank you.

Reviewer #2:

Remarks to the Author:

The authors have diligently performed additional work and tried to address my previous concerns. Yet, I remain skeptical about the outcome of a screen for the identification of T2D "causal" genes using as read-out insulin content, rather than insulin secretion.

In the rebuttal to my point 3 the authors do not actually indicate what is the evidence for changes in insulin content being causally linked to T2D, except for referring to their own previous studies on PAM, which was neither found in the present screen nor to be among the consistently downregulated genes in islets of donors with T2D.

In response to point 3, the authors have included data about insulin secretion of primary human islets of donors carrying variants in the CALCOCO2 locus. However, is there clinical laboratory evidence for these donors being prediabetic? In the PDF the Supplementary Table 6 about the characteristics of these islet donors is missing.

What is also puzzling is that two variants in the CALCOCO2 locus do not reduce insulin content in primary islet cells (Fig. 6m and 6o), which seems in contrast with the outcome of the screen that led in the first place to the identification of CALCOCO2 as a candidate causal gene of T2D. Of note, the data presented in Fig. 6m-p do not point to significant changes in insulin content or secretion in basal glucose condition, albeit in the screen was performed in EndoC-beta H1 kept in basal conditions. The reason for exposing primary human islets to 1 mM glucose, which is very low, is unclear, while in panel 6n the significantly greater glucose stimulated insulin secretion of controls is likely attributable to the values of 1-2 donors only.

The interpretation of the EM data is difficult. Why should an increased mature/immature insulin granule ratio upon CALCOCO2 silencing reduce total insulin content, a phenotype not observed in carriers of CALCOCO2 variants? And why insulin secretion?

The authors state that reduction of proinsulin in CALCOCO2-depleted EndoC-beta H1 cells cannot be attributed to reduced expression of PCSK1, CPE and PCSK2. Apart from the impression that PCSK1 is reduced in CALCOCO2-silenced cells (Supplementary Figure 8o), the activity of these enzymes may be altered due to factors other than their expression levels.

Reviewer #3:

Remarks to the Author:

Grotz et al have submitted a revision with a more complete evaluation of the role of CALCOCO2 in insulin content. Their experiments confirm that loss of CALCOCO2 is a direct cause of the loss of insulin in cell culture, which they postulate is due to the degradation of immature insulin granules. Although the human carriers of T2D-associated variants had alterations in insulin secretion, not content, there are sufficient mechanisms to explain that to not consider it a weakness. Finally, the responses to comments regarding assay development and the CRISPR screen itself were thorough and appropriate. I would recommend publication.

Author Rebuttal, first revision:

Response to referees

Grotz, AK et al. - A genome-wide CRISPR screen identifies CALCOCO2 as a regulator of beta cell function involved in type 2 diabetes risk

We thank the reviewers, and the editor, for their careful evaluation of our manuscript and their feedback to further improve it. We were pleased to learn that our revisions to strengthen our conclusions and to provide mechanistic insight into the role of CALCOCO2 in human beta cell function satisfied most of the expressed concerns and comments. Below we outline our response to the remaining comments and detail how we have adapted the manuscript to address them.

Reviewer 1

The authors made significant improving the manuscript, especially about the mechanism of action of CALCOCO2. I think their revision to my comments is satisfying, therefore I have no further comments and recommend the acceptance of this manuscript. Thank you.

We thank the reviewer for their evaluation of our manuscript, their suggestions for revisions that further strengthened our manuscript and are pleased to hear that they now recommend acceptance.

Reviewer 2

The authors have diligently performed additional work and tried to address my previous concerns. Yet, I remain skeptical about the outcome of a screen for the identification of T2D “causal” genes using as read-out insulin content, rather than insulin secretion.

We thank the reviewer for their evaluation of our manuscript, their previous suggestions for revisions which have already strengthened our manuscript and we would now like to further address their remaining concerns.

In the rebuttal to my point 3 the authors do not actually indicate what is the evidence for changes in insulin content being causally linked to T2D, except for referring to their own previous studies on PAM, which was neither found in the present screen nor to be among the consistently downregulated genes in islets of donors with T2D.

We thank the reviewer for giving us the opportunity to further explain why we think that the evidence points towards a causal link between insulin content and T2D risk.

First, throughout the manuscript, we have further rephrased the statements addressing the causality of insulin content to not just highlight that the screen can be used as an approach to identify causal genes for T2D but also its value to identify regulators of beta-cell function irrespective of their causal association with T2D.

Second, although we initially only highlighted the *PAM* locus as one example to primarily focus on the novel results, we would like to point out that this is only one of many T2D loci where insulin content has been identified as an underlying contribution to the molecular mechanism. As outlined by Dame Professor Frances Ashcroft in her Banting Lecture at the American Diabetes Association scientific sessions earlier this month there is compelling evidence that heterozygous activating mutations in *KCNJ11* which cause neonatal diabetes, reduce insulin content highlighting the importance of this cellular phenotype in diabetes pathogenesis. Furthermore, the GWAS locus with the strongest association with T2D risk to date, *TCF7L2*, has been shown to regulate insulin production and processing while the T2D- risk allele at rs7903146¹ has been shown to be associated with reduced insulin content. Taken together these data provide important support for the importance of insulin content in

contributing to mechanisms of beta-cell dysfunction in diabetes. We have updated the manuscript to provide additional evidence to support the value of this phenotype.

“Changes in insulin content have also previously been mechanistically associated with T2D-associated risk genes, such as at the PAM locus where T2D-risk alleles cause reduced expression and/or function and reduced PAM expression is associated with reduced insulin content or for TCF7L2, the locus mediating the strongest risk for T2D which has been identified as a master regulator of insulin production with decreased insulin content in carriers of the risk allele²³⁻²⁵.”

Third, as we also pointed out in our last revisions, we fully agree based on our previous studies demonstrating reduced insulin content upon loss of PAM, that we would have expected to detect PAM as one of the screening hits. However, as we have already demonstrated in our proof-of-concept study, KO of PAM using one of the sgRNAs from the TKOv3 library induced a consistent but only very small effect in our FACS based insulin assay. In a separate study using gRNAs from a genome-wide library to generate PAM knockout cells, we further detected residual gene expression and a protein isoform which had not been targeted with the sgRNAs in the library. It is therefore possible that some genes, including PAM, are not detected due to incomplete gene knockout and a resulting small effect size.

Forth, we have clearly stated in the discussion that this is a first cellular phenotype and ultimately further screens under different perturbations and with additional phenotype including insulin secretion are warranted.

“While this genome-wide CRISPR screen has identified novel genes involved in beta cell function and provides a resource for future integration and in-depth studies, it only captures one of several possible disease relevant cellular phenotypes under one condition. The screen assessed changes in insulin content under basal glucose conditions and consequently, genes specifically affecting insulin secretion without modulating insulin content or changes in intracellular content that only occur upon glucose starvation or stimulation were not captured.”

Finally, the reviewer expresses concern that islets from people with Type 2 Diabetes do not show reduced PAM expression. We do not share this concern. There are many reasons why differentially expressed genes identified in studies comparing individuals with and without diabetes will not overlap with genes involved in type 2 diabetes risk or eQTLs. These include but are not limited to statistical power, study design and the molecular mechanism which may not involve effects on gene expression.

In response to point 3, the authors have included data about insulin secretion of primary human islets of donors carrying variants in the CALCOCO2 locus. However, is there clinical laboratory evidence for these donors being prediabetic? In the PDF the Supplementary Table 6 about the characteristics of these islet donors is missing.

We thank the reviewer for raising this point of confusion and apologize that Supplementary Table 6 was not part of the merged pdf document as it was submitted as a separate excel file. In our analysis of primary human islet donors, we only included non-diabetic donors with an HbA1C < 6 which we included in this Supp Table 6. We have provided an extract of the table here for clarity and can confirm that the entire excel table is available to reviewers:

Record ID	rs10278_CALCOCO2	ID	rs35895680_NA	Age	Sex	Body mass	Glycated hemog	Insulin Content (pg/ml)			Secretion at 1mM glucose (pg/ml)			Secretion at 16.7 mM glucose (pg/ml)			
R051	R051	CC	R051	CA	73	Male	23.7	5.7	647519	615096	854760	9735	8186	18612	34088	31768	33109
R059	R059	CG	R059	CA	79	Female	23.5	5.8	2193695	1366404	1415162	616	917	1730	5626	7591	6544
R060	R060	CC	R060	CA	54	Female	21.10721	5.9	134141	194305	105915	573	336	357	6750	3205	4562
R067	R067	CC	R067	CC	60	Male	26	5.5	1113470	1423477	1462383	903	1003	2712	28785	31783	30690
R073	R073	CC	R073	CC	74	Female	28.3	5.4	1423300	955442	1054195	2473	616	1172	24321	16069	23746
R074	R074	CC	R074	CC	55	Male	26.4	5.5	18267	321683	224001	232	520	464	5735	8029	3928
R075	R075	CC	R075	CC	27	Male	26.23457	5.4	296774	265190	648531	1358	1062	1108	12488	9327	10224
R077	R077	CC	R077	CC	54	Female	24.7	5.2	436743	157815	224757	10410	213	116	3711	1664	1828
R081	R081	CG	R081	CA	68	Male	23.7	5.9	1030091	818286	711709	1356	1732	1058	8937	11727	12891
R082	R082	CG	R082	CA	65	Female	24.9	5.4	1241763	1807287	2187223	4023	1591	3875	16948	12150	17245

We have also added additional details to the methods to clarify the clinical stratification of the donors:

“Human islets for T2D risk allele studies were obtained from 173 non-diabetic organ donors (Haemoglobin A1C (HbA1C) < 6) isolated and assessed at the Alberta Diabetes Institute IsletCore as previously described and cultured in DMEM with 10% FCS and 100 U/ml penicillin/streptomycin (all Thermo Fisher) at 37°C and 5% CO2 4,5.”

What is also puzzling is that two variants in the CALCOCO2 locus do not reduce insulin content in primary islet cells (Fig. 6m and 6o), which seems in contrast with the outcome of the screen that led in the first place to the identification of CALCOCO2 as a candidate causal gene of T2D. Of note, the data presented in Fig. 6m-p do not point to significant changes in insulin content or secretion in basal glucose condition, albeit in the screen was performed in EndoC-beta H1 kept in basal conditions (1). The reason for exposing primary human islets to 1 mM glucose, which is very low, is unclear (2), while in panel 6n the significantly greater glucose stimulated insulin secretion of controls is likely attributable to the values of 1-2 donors only (3).

- (1) We thank the reviewer for pointing out this interesting question and giving us the opportunity to elaborate further on the differences between our initial screen in the human beta-cell model and the experiments using primary human islets. We fully agree with the reviewer that work beyond the scope of this manuscript remains to be done to better understand the observed reduced insulin content in a model of human beta cells upon loss of *CALCOCO2* whilst human carriers of T2D-associated variants demonstrate an effect on insulin secretion. However, we would like to point out that many potential mechanisms could translate *in vitro* changes in insulin content into *in vivo* insulin secretion changes. One potential explanation which we have alluded to in the discussion are long-term compensatory effects of adapted insulin expression and granule biogenesis, leading to unaffected total insulin content but simultaneously an affected pool of mature and secretion-ready granules. This mechanism would also be supported by our observation that we only see an effect upon incubation with high glucose.

- (2) We completely agree with the reviewer that a glucose concentration of 1mM is at the low end of the assay range. However, we would like to point out that this concentration has been widely used in the literature and we were able to obtain the largest number of donors that had been assayed under those conditions^{6,7}. In addition, the significant changes can be observed upon incubation with high glucose and not under low glucose so even increasing that concentration to a more physiological lower range would not affect our high glucose results.

- (3) We thank the reviewer for this question, to ensure that the observed effect is not driven by donors outside of the expected range, we performed ROUT outlier testing and removed any outlier prior to our analysis. Due to a large sample size of more than 73 donors in each group, it is unlikely that a single donor that has not been removed through outlier analysis and therefore falls within a natural distribution of the insulin secretory response would affect the overall response as drastically. Additionally, to support this claim, we reperformed the analysis without the donors with the strongest response and we do not see a loss of significance between carriers and controls.

The interpretation of the EM data is difficult. Why should an increased mature/immature insulin granule ratio upon *CALCOCO2* silencing reduce total insulin content, a phenotype not observed in carriers of *CALCOCO2* variants? And why insulin secretion?

We thank the reviewer for this comment. While we do observe an increased proportion of mature granules upon *CALCOCO2* silencing in our EM analysis (Fig 6c), we primarily observed a **reduced overall number** of insulin granules/ μm^2 in *CALCOCO2* silenced cells as shown in Fig 6b. This reduction of immature granules and a concomitant reduction of proinsulin was further independently verified through pooled ELISA based measurements, highlighting the sensitivity and validity of our image-based EM-analysis. Our further analysis of *CALCOCO2*'s effect on autophagy revealed a prominent accumulation of autophagosomes, pointing towards an effect of insulin content whereby loss of *CALCOCO2* affects autophagic degradation of proinsulin containing immature granules leading to reduced overall total insulin content. We state those mechanistic insights and predictions in both, the results and discussion section:

“Our results indicate that CALCOCO2 is a regulator of insulin granule homeostasis and mediates its observed effect on total insulin content through autophagy-based reduction in proinsulin and immature granules.”

“Instead, our results support an effect on insulin content through degradation of proinsulin containing immature insulin granules.”

As alluded to in our response to the reviewer's previous comment, many mechanisms could lead to the observed difference between our *in vitro* insulin content phenotype and the insulin secretion phenotype observed in primary human islets and should therefore not invalidate either of those results.

The authors state that reduction of proinsulin in *CALCOCO2*-depleted EndoC-beta H1 cells cannot be attributed to reduced expression of PCSK1, CPE and PCSK2. Apart from the impression that PCSK1 is reduced in *CALCOCO2*-silenced cells (Supplementary Figure 8o), the activity of these enzymes may be altered due to factors other than their expression levels.

We thank the reviewer for highlighting that our chosen western blot images for PCSK1 and the absence of a figure showing our quantification of the western blot data resulted in a lack of evidence to support our claims of an unchanged expression level. To rectify this, we have now included a representative western blot image that better mimics the effect observed across replicates and added the accompanying quantification blots as Supplementary Fig S8p/q demonstrating no significant changes. We also completely agree that changes beyond the expression levels of the enzymes affecting their

activity were not assessed and we therefore modified the statement in the manuscript to address that limitation:

'This reduction in proinsulin was likely not mediated by altered insulin maturation as expression levels of genes involved in insulin processing (PCSK1, CPE and PCSK2) were unchanged upon CALCOCO2 silencing while enzyme activity was not assessed (Supplementary Fig S8).'

Reviewer 3

Grotz et al have submitted a revision with a more complete evaluation of the role of CALCOCO2 in insulin content. Their experiments confirm that loss of CALCOCO2 is a direct cause of the loss of insulin in cell culture, which they postulate is due to the degradation of immature insulin granules. Although the human carriers of T2D-associated variants had alterations in insulin secretion, not content, there are sufficient mechanisms to explain that to not consider it a weakness. Finally, the responses to comments regarding assay development and the CRISPR screen itself were thorough and appropriate. I would recommend publication.

We thank the reviewer for their evaluation of our manuscript, their suggestions for revisions that strengthened our manuscript and were pleased to hear about the recommendation of publication.

Decision Letter, second revision:

Dear Anna,

Thank you for submitting your revised manuscript "A genome-wide CRISPR screen identifies CALCOCO2 as a regulator of beta cell function involved in type 2 diabetes risk" (NG-A57652R1). I would like to apologise for the time it took for this decision - but I am happy that I now have some news.

As previously discussed, we asked Reviewer #3 to comment on your responses to Reviewer #2 (who was not able to review this revision).

The reviewer is satisfied with your responses, and therefore we'll be happy in principle to publish it in Nature Genetics, pending minor revisions to satisfy the referees' final requests and to comply with our editorial and formatting guidelines.

We are now performing detailed checks on your paper and will send you a checklist detailing our

editorial and formatting requirements soon. Please do not upload the final materials and make any revisions until you receive this additional information from us.

Sincerely,

Michael Fletcher, PhD
Senior Editor, Nature Genetics

ORCID: 0000-0003-1589-7087

Reviewer #3 (Remarks to the Author):

I am satisfied with the responses of the authors to the next round of comments and questions. This work represents an excellent step in the scientific process toward understanding the genetics of the beta cell and, potentially, T2D. The expanded text describing rationale and caveats is appropriate, and there is additional data presented to provide further evidence for their conclusions.

Further, I am not concerned about some of the issues raised with islet studies. It is very difficult to acquire islets from T2D individuals, much less pre-diabetic individuals, and the experiments described seem to conform to standard practices in the field.

Final Decision Letter:

Dear Anna,

I am delighted to say that your manuscript "A genome-wide CRISPR screen identifies CALCOCO2 as a regulator of beta cell function influencing type 2 diabetes risk" has been accepted for publication in an upcoming issue of Nature Genetics.

Due to the importance of these deadlines, we ask that you please let us know now whether you will be difficult to contact over the next month. If this is the case, we ask you provide us with the contact information (email, phone and fax) of someone who will be able to check the proofs on your behalf,

and who will be available to address any last-minute problems.

Your paper will be published online after we receive your corrections and will appear in print in the next available issue. You can find out your date of online publication by contacting the Nature Press Office (press@nature.com) after sending your e-proof corrections. Now is the time to inform your Public Relations or Press Office about your paper, as they might be interested in promoting its publication. This will allow them time to prepare an accurate and satisfactory press release. Include your manuscript tracking number (NG-A57652R2) and the name of the journal, which they will need when they contact our Press Office.

Please note that *Nature Genetics* is a Transformative Journal (TJ). Authors may publish their research with us through the traditional subscription access route or make their paper immediately open access through payment of an article-processing charge (APC). Authors will not be required to make a final decision about access to their article until it has been accepted. [Find out more about Transformative Journals](https://www.springernature.com/gp/open-research/transformative-journals)

Authors may need to take specific actions to achieve [compliance with funder and institutional open access mandates](https://www.springernature.com/gp/open-research/funding/policy-compliance-faqs). If your research is supported by a funder that requires immediate open access (e.g. according to [Plan S principles](https://www.springernature.com/gp/open-research/plan-s-compliance)) then you should select the gold OA route, and we will direct you to the compliant route where possible. For authors selecting the subscription publication route, the journal's standard licensing terms will need to be accepted, including [self-archiving-and-license-to-publish](https://www.nature.com/nature-portfolio/editorial-policies/self-archiving-and-license-to-publish). Those licensing terms will supersede any other terms that the author or any third party may assert apply to any version of the manuscript.

Please note that Nature Portfolio offers an immediate open access option only for papers that were first submitted after 1 January, 2021.

If you have not already done so, we invite you to upload the step-by-step protocols used in this manuscript to the Protocols Exchange, part of our on-line web resource, natureprotocols.com. If you complete the upload by the time you receive your manuscript proofs, we can insert links in your article that lead directly to the protocol details. Your protocol will be made freely available upon publication of your paper. By participating in natureprotocols.com, you are enabling researchers to more readily reproduce or adapt the methodology you use. [Natureprotocols.com](http://natureprotocols.com) is fully searchable, providing your protocols and paper with increased utility and visibility. Please submit your protocol to <https://protocolexchange.researchsquare.com/>. After entering your [nature.com](http://www.nature.com) username and password you will need to enter your manuscript number (NG-A57652R2). Further information can be found at <https://www.nature.com/nature-portfolio/editorial-policies/reporting-standards#protocols>

Sincerely,

Michael Fletcher, PhD
Senior Editor, Nature Genetics

ORCID: 0000-0003-1589-7087

Click here if you would like to recommend Nature Genetics to your librarian
<http://www.nature.com/subscriptions/recommend.html#forms>

** Visit the Springer Nature Editorial and Publishing website at http://editorial-jobs.springernature.com?utm_source=ejp_NGen_email&utm_medium=ejp_NGen_email&utm_campaign=ejp_NGen for more information about our career opportunities. If you have any questions please click [here](mailto:editorial.publishing.jobs@springernature.com).**